# A versatile platform for sequential glyco-, phospho-, and proteomics with multi-PTMs integration

Xuefang Dong[1,2,7], Fangfang Xiong[1,2,7], Guangzhu Du[1,3], Yunfei Yang[4,5], Cheng Chen[1,2], Yun Cui[2], Xinlian Ding[1,6], Xiuling Li [1,2] ✉, Yidong Shen [4,5] ✉ & Xinmiao Liang [1,2] ✉

Serial multi-omic analysis of proteome, phosphoproteome, and glycoproteome is pivotal for elucidating drug mechanisms, discovering biomarkers, and identifying therapeutic targets. However, simultaneous multi-level post-translational modifications (PTMs) analysis via parallel processing is hampered by laborious, time-consuming procedures and inconsistent reproducibility. We present an integrated Multi-level PTMs-Proteomic Enrichment platform (MuPPE), enabling sequential glycoproteome, phosphoproteome, and proteome analysis from single biological samples. It combines protein aggregation capture with on-bead digestion and tandem enrichment, achieving superior reproducibility (CV 12.3% vs 17.6% conventional methods) while reducing processing time by 87.5% (4 hours vs 32 hours). MuPPE also enhances coverage, identifying more serum glycopeptides and brain phosphopeptides than other platforms. Applied to aging mouse cohorts, the platform uncovers tissue-specific PTMs remodeling and brain barrier dysfunction. For arsenic mechanisms of action, MuPPE reveals drug-induced PTMs crosstalk between glycosylation and phosphorylation-driven pathway regulation. MuPPE offers a transformative tool for advancing multi-omics insights across precision medicine and disease research.

With the achievement of the Human Genome Project[1] and the Human Proteome Project[2], the exploration of physiology mechanism and disease biomarkers on the genetic and proteomic levels has become possible. Yet the correlation between the proteome and the genome is not always straightforward due to the complexity of biological system[3,4]. One of the most significant factors is the widespread cooc-currence of post-translational modifications (PTMs) of proteins, such as phosphorylation, glycosylation, and acetylation[5]. While these PTMs cannot be directly predictable and controlled from the genome, they constantly modulate the dynamic changes and functional states of proteins in response to various cellular conditions and external stimuli, alter protein function and stability[6,7]. Decrypting proteomic PTMs can not only characterize mechanisms of action (MoA) for drugs, generate drug-specific PTM signatures[8], but also reveal potential tumor-specific biomarkers or drug targets[9,10]. Thus, elucidating underlying biological mechanisms by integrated multi-level PTMs proteomics holds great

[1]State Key Laboratory of Phytochemistry and Natural Medicines, Dalian Institute of Chemical Physics, Chinese Academy of Sciences, Dalian, P. R. China. [2]Ganjiang Chinese Medicine Innovation Center, Nanchang, P. R. China. [3]Department of Materials Science and Engineering, Dalian Maritime University, Dalian, P. R. China. [4]Department of Geriatrics, The First Affiliated Hospital of Chongqing Medical University, Chongqing, P. R. China. [5]State Key Laboratory of Cell Biology, Shanghai Institute of Biochemistry and Cell Biology, Center for Excellence in Molecular Cell Science, Chinese Academy of Sciences, Shanghai, P. R. China. [6]University of Chinese Academy of Sciences, Beijing, P. R. China. [7]These authors contributed equally: Xuefang Dong, Fangfang Xiong. ✉e-mail: lixiuling@dicp.ac.cn; yidong.shen@sibcb.ac.cn; liangxm@dicp.ac.cn

promise and provides a deeper insight of functional proteome[8]. However, the diversity of PTMs typically increases the complexity of multi-level PTMs proteomic research and then proposes great challenges to sample preparation before Mass spectrometry (MS) analysis. Therefore, establishing an integrated and serial enrichment approach that enables synchronous analysis of multi-level PTMs is inevitable with the single aliquot of sample input. Besides, serial processing of the same proteomic sample in contrast to the parallel samples processing is expected to yield a stronger correlation between multi-level PTMs proteomic analysis. This would generate more comprehensive proteomic datasets with different PTMs layers, significantly promoting subsequent biological information analysis.

Recently, a series of integrated proteomic sample preparation methodologies, e.g., practical SISPROT, in-StageTip (iST), pressure cycling technology-assisted, and single-pot solid-phase-enhanced sample preparation (SP3) methods, have been developed[11-14]. Although these exciting methodologies have achieved significant advancements, they are primarily tailored for the proteome, yet they do not adequately account for the complexity of multi-PTMs. Parallel methods facilitate simultaneous analysis of different PTMs, while sequential methods permit stepwise enrichment process. However, these approaches are not integrated, face challenges such as low throughput, sample loss during multiple-step processes and the reproducibility concerns. These limitations can significantly impact the comprehensiveness and reliability of PTMs-proteomics, potentially leading to compromised results. Steven A. Carr et al. presented the MONTE enrichment workflow, enabling effectively serial, deep-scale analysis of immunopeptidome, ubiquitylome, proteome, phosphoproteome, and acetylome from the same tissue sample and offering an extensive view of disease biology[15]. While powerful, this workflow was tailored for tissue samples, not for biological liquid samples and does not introduce other crucial PTMs, such as glycosylation. Therefore, it is essential to develop a universal sample preparation workflow that enables comprehensive, simultaneous analysis of the proteome and multi-level PTMs from the same biological sample, even in minor sample volumes or low concentrations of proteins. Meanwhile, this type of workflow should not only ensure the consistency during sample analysis for both the proteome and multi-level PTMs but also enhance the utilization efficiency of precious biological samples.

To address this issue, an integrated multi-level proteomic platform, termed MuPPE (Multi-level PTM-Proteomic Enrichment), was introduced and exemplified with the profiling of serial glycoproteome and phosphoproteome and proteome. MuPPE can achieve serial sample preparation in one single tube and exhibit potentials for high-throughput sample preparation. And this platform supports customized combinations of various multi-level proteomics according to individual research requirements. MuPPE was systematically evaluated through leveraging different types of biological samples (cell lysates, body fluids and tissues) and also displays its advantages for the samples of small volume or in low protein concentration such as cerebrospinal fluid (CSF). And its versatility was also demonstrated through interpreting the mechanisms of aging in multiple samples from mice cohorts of different ages and investigating MoA of arsenic trioxide with the NB4 cell line at multi-level proteomics. This concise and integrated platform brings the convenience to proteomic sample preparation, improving efficiency in multi-level proteomic analysis.

## Results
### Development of MuPPE: an integrated platform for streamlined multi-level proteome and PTMs analysis
General PTMs-proteomics workflows rely on fragmented, multi-step processing for the analysis of the proteome, glycoproteome, and phosphoproteome. These workflows introduce challenges such as sample loss, variability, and inefficiency[16] (Fig. 1a). To overcome these limitations, we have developed the MuPPE platform-a unified pipeline that integrates protein aggregation capture (PAC), on-beads digestion, and sequential PTMs enrichment. As a proof-of-principle, this platform enables the simultaneous interrogation of the proteome, glycoproteome, and phosphoproteome (Fig. 1b). The MuPPE workflow commences with protein denaturation, reduction, and alkylation, followed by the introduction of a modified and scalable PAC method for efficient protein isolation[17]. Proteins aggregate on hydrophilic beads, and on-beads digestion is employed to minimize sample transfers. Glycopeptides are enriched directly from these microspheres, while the flow-through is reserved for phosphopeptides enrichment and global proteome analysis. This design simplifies the workflow to just two tubes, eliminating intermediate transfers and enabling the potential for high-throughput automation. Overall, the MuPPE platform simplifies the workflow (requiring only two tubes), eliminating the need for multiple sample transfers and enabling the simultaneous exploration of the proteome, glycoproteome, and phosphoproteome. The streamlined workflow also endows MuPPE with the potential for easy automation in high-throughput applications. Furthermore, this platform facilitates the tailored integration of diverse multi-level proteomic approaches, in alignment with individualized research requirements. Meanwhile, the incorporation of PAC significantly enhances the applicability of MuPPE for samples with low protein abundance or small volume. The application of this platform to different biological samples will underscore its robustness and efficiency, leading to a more holistic understanding of biological mechanisms.

In-solution digestion is widely employed to enzymatically cleave proteins into peptides, facilitating subsequent MS analysis. This process is conducted in a homogeneous liquid phase without the need for immobilized substrates, making it suitable for a broad range of protein samples. Contrastively, PAC is an advanced technique and leverages the inherent tendency of proteins to form aggregates under organic solvent conditions, which can then be captured and isolated from the sample[17]. We systematically compared MuPPE with the gold-standard in-solution digestion method using human serum and mouse brain tissue samples. In terms of MS pattern compatibility and minimal bias, the base peak chromatograms (Fig. 1c) showed that the MS patterns generated by the two methods were nearly identical, confirming the compatibility of MuPPE with downstream MS analysis. For global human serum proteome analysis, MuPPE equipped with Click-Maltose (Click-Mal) hydrophilic beads identified 313 protein groups, while in-solution digestion method identified 298 protein groups, with 297 shared ones (Fig. 1d). The overlap of identified proteins and the correlation of protein identification coverage ($R^2 = 0.88$, Fig. 1e) demonstrated that MuPPE does not introduce significant bias, validating its reliability. Coefficient of variation (CV%) analysis (Fig. 1f) revealed that MuPPE yields more precise quantifications (mean CV = 12.3% vs. 17.6% for in-solution digestion), attributed to reduced sample handling. For glycoproteome/phosphoproteome analysis in human serum, MuPPE identified 2640 glycopeptides (compared to 1520 identified by in-solution digestion) and 137 glycoproteins (compared to 121 identified by in-solution digestion); in mouse brain, MuPPE identified 4144 phosphopeptides (compared to 3778 identified by in-solution digestion) and 1586 phosphoproteins (compared to 1445 identified by in-solution digestion, Fig. 1g). Notably, the enrichment workflow of MuPPE and the in-solution method were both performed with Click-Mal[18-20] beads and IMAC-Ti[4+] beads for glycopeptide and phosphopeptide enrichment. These results clearly demonstrate that MuPPE outperforms the conventional in-solution method in PTMs and proteome identification efficiency. Additionally, we compared the compatibility of the in-solution method and the MuPPE method with TMT labeling experiments, and confirmed that MuPPE is compatible with TMT-based multiplexed quantification (Supplementary Fig. 1).

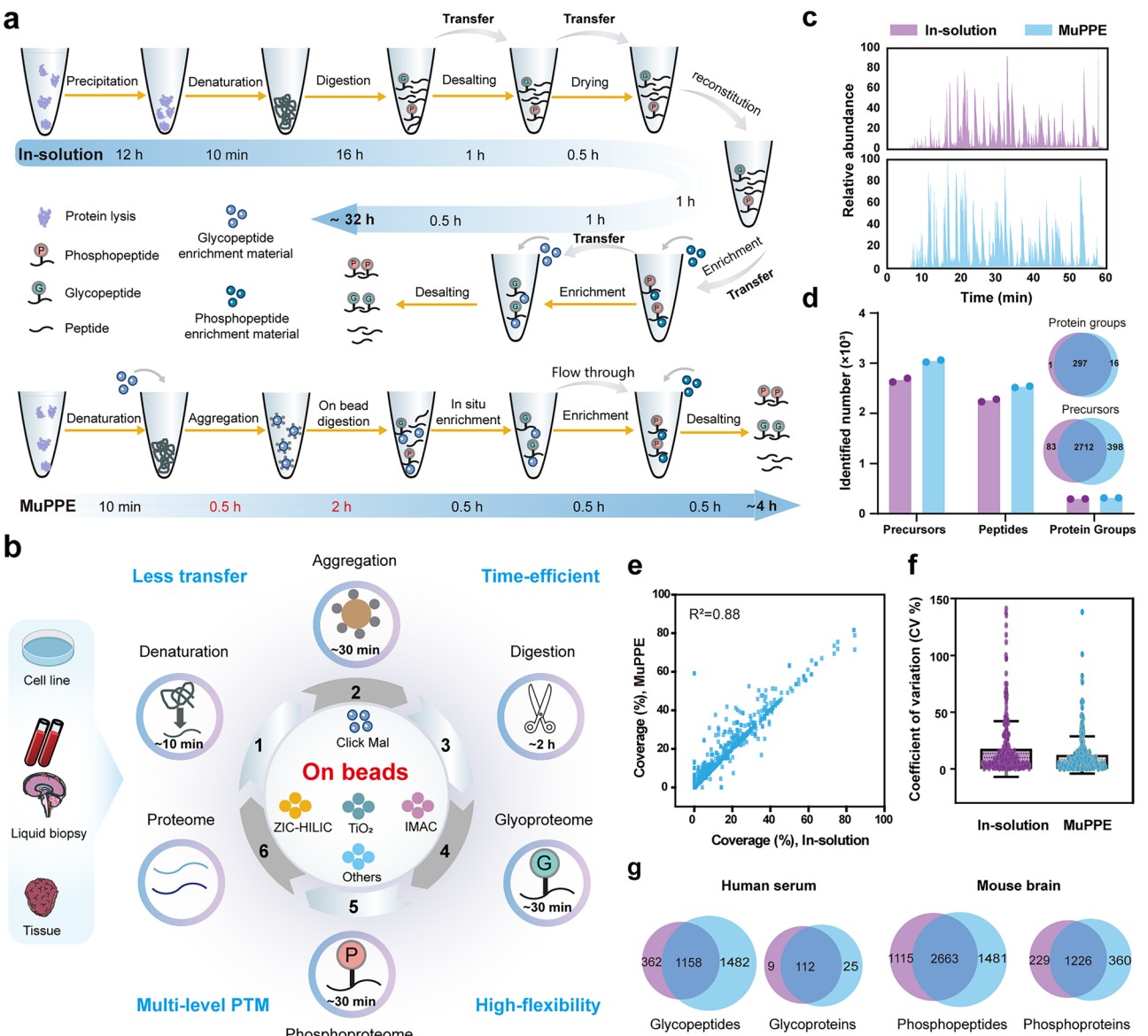

**Fig. 1 | The MuPPE platform enables integrated multi-level proteomics with enhanced efficiency and coverage. a** Schematic workflow of the MuPPE platform. Protein aggregation capture (PAC), on-bead digestion, and serial enrichment of glycopeptides, phosphopeptides, and proteome are integrated into a single pipeline. Key steps (denaturation, aggregation, digestion, enrichment) and their approximate timeframes are depicted. **b** The design rationale of MuPPE. Graphical summary of MuPPE's core design principles, centered on on-bead multi-function integration to address proteomic workflow challenges. Image(s) provided by Servier Medical Art (https://smart.servier.com), licensed under CC BY 4.0 (https://creativecommons.org/licenses/by/4.0/). **c** Base peak chromatograms of digested human serum samples prepared with in-solution method (top) and MuPPE (bottom), showing comparable MS pattern profiles. **d** Comparative identification of precursors, peptides, and protein groups in human serum by in-solution (purple)

and MuPPE (blue) methods, demonstrating increased identifications by MuPPE. **e** Correlation of protein identification coverage between in-solution and MuPPE methods ($R^2 = 0.88$), indicating minimal bias in MuPPE. **f** Coefficient of variation (CV %) distribution of protein quantifications by in-solution (purple) and MuPPE (blue) methods, illustrating higher precision of MuPPE. Each data point in this figure (**f**) represents the CV% of one individual protein quantified by either the in-solution method (purple) or MuPPE method (blue). Centre line indicates the median; box limits denote the 25th and 75th percentiles; whiskers represent 1.5 × the interquartile range; points show individual quantified features. **g** Venn diagrams showing overlap of identified glycopeptides/glycoproteins (human serum, left) and phosphopeptides/phosphoproteins (mouse brain, right) by in-solution and MuPPE methods, highlighting expanded coverage by MuPPE. For Figure (**c**–**g**), $n = 2$ technical replicates per method, error bars presented as mean ± SD.

## MuPPE enables streamlined and robust glycoproteomic analysis across diverse biological matrices

Glycoproteomics links proteome and glycosylation analysis, and enables the identification of the individual molecular features of all glycoproteins in different biological samples[21,22]. In such analysis, the proteome and glycoproteome are typically evaluated independently using independent biological samples[9,23]. To facilitate sequential identification of the proteome and glycoproteome, we optimized a glycoproteomic workflow within MuPPE that allows for subsequent

enrichment of glycopeptides. PTMs enrichment efficiency hinges critically on enzymatic digestion-a pivotal step bridging protein capture and PTMs interrogation. To optimize this, we systematically varied tryptic digestion durations (1–12 h) within the MuPPE workflow using human serum, comparing the performance via protein and glycopeptide identifications (Fig. 2a–c). The comparative analysis indicated that a digestion time of under 3 h is sufficient to achieve nearly consistent identification outcomes (Fig. 2a) and glycopeptides (Fig. 2b), with glycopeptide selectivity remaining stable across these timepoints

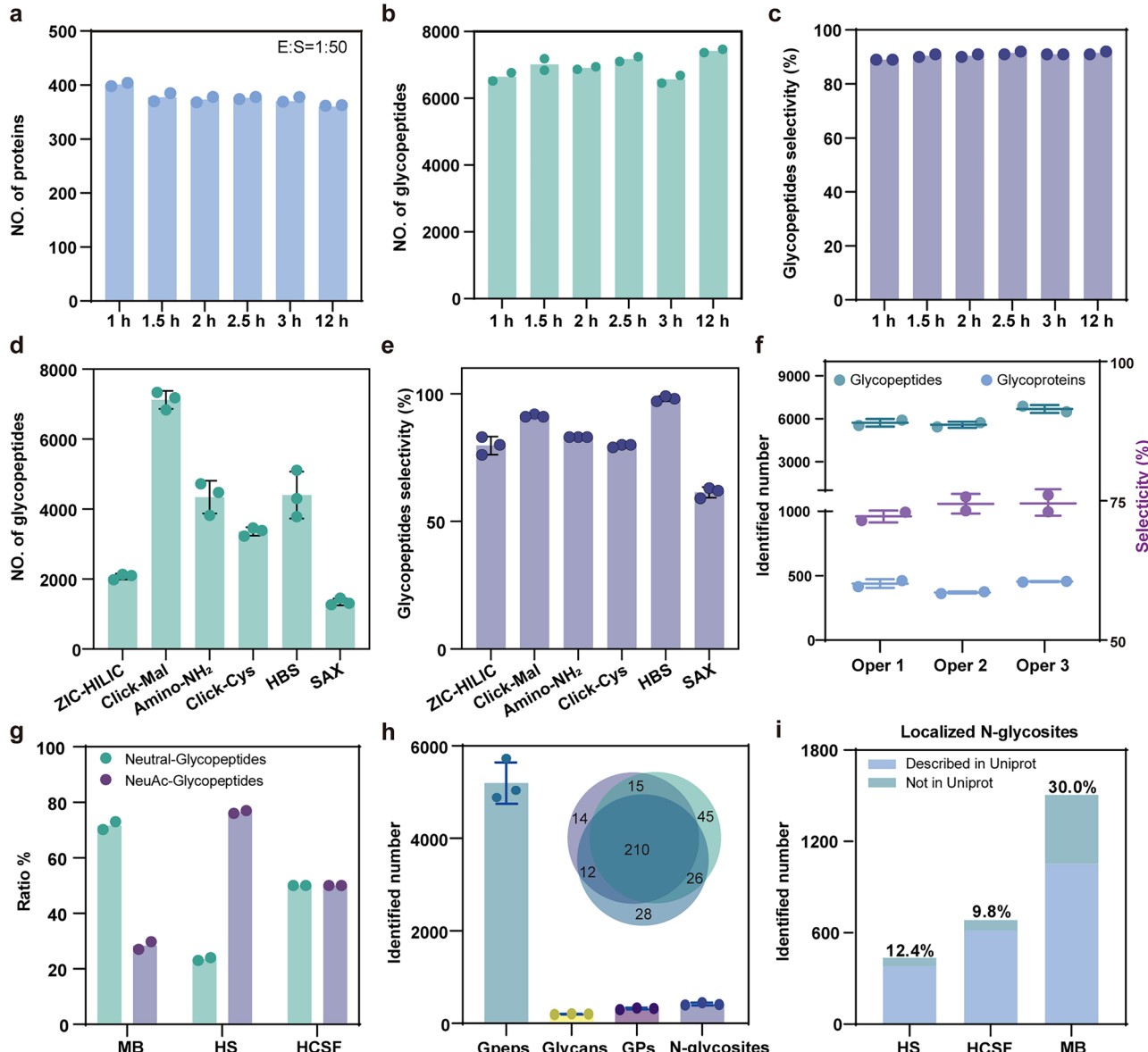

**Fig. 2 | The feasibility of incorporating glycosylation workflow into MuPPE.** The optimized enzymatic digestion parameters of the MuPPE workflow. **a–c** Comparison of identified human serum protein groups (**a**), glycopeptides (**b**), and the glycopeptides selectivity (**c**) when digested with E/S ratio of 1 : 50 in digestion time from 1 to 12 h ($n = 2$). Comparative analysis of the numbers of glycopeptides (**d**) and the corresponding selectivity of glycopeptides (**e**) identified using different hydrophilic beads in human serum ($n = 3$). **f** The glycosylation identification results of glycopeptides, glycoproteins, and the selectivity of glycopeptides from different operators with mouse brain lysates ($n = 2$). Oper: Operator. **g** Sialylated and neutral

glycosylation levels comparison across various biological samples ($n = 2$; MB: mouse brain, HS: human serum, HCSF: human CSF). **h** The glycosylation identification of HCSF. Gpeps, GPs, and N-glcosites are the abbreviations for glycopeptides, glycoproteins, and N-glycosylation sites ($n = 3$). **i** The N-glycosylation sites of MB, HS, and HCSF identified with MuPPE relative to those reported in UniProt, as summarized in the GlyGen dataset (Version 2.6.1, released on 08/07/2024), and [https://data.glygen.org/GLY_000039] for mouse N-glycosylation sites dataset and [https://data.glygen.org/GLY_000038] for human N-glycosylation sites dataset. Error bars presented as mean ± SD.

(Fig. 2c). This underscores that short digestion windows (≤ 3 h) suffice to maximize proteomic coverage while streamlining workflow timelines, aligning with MuPPE's efficiency-focused design. Then we evaluated the enrichment performance of various hydrophilic beads, including both commercially available (ZIC-HILIC, Amino-NH₂ bonded, SAX beads) and home-made beads (Click-Mal[18–20], HBS[24], Click-Cys[25] beads) for N-linked glycopeptide enrichment. Click-Mal beads, reported widely applicable in glycopeptides enrichment, demonstrated the relative superior enrichment performance based on the numbers of identified N-linked glycopeptides and glycoproteins indeed (Fig. 2d, e). Considering the potential effects of beads surface charge properties and hydrophilicity, we hypothesize that the superior performance of Click-Mal may be associated with the relatively neutral surface charge

and strong hydrophilicity of the beads. Consequently, Click-Mal beads were employed for glycopeptide capture in all subsequent experiments. We then conducted glycopeptides enrichment using Click-Mal beads and observed distinct glycosylation profiles across various biological samples for the following work. The robustness of MuPPE was tested by examining the intra-laboratory reproducibility of glycosylation results with mouse brain lysates when MuPPE was processed by different operators (Fig. 2f). Remarkably, different operators can achieve almost consistent identification results and selectivity.

The MuPPE also exhibits no bias towards glycan types and provides valuable insights into glycosylation patterns (Fig. 2g). Specifically, sialylation levels exceeded neutral glycosylation levels in human serum but opposite in mouse brain, whereas they were comparable in

human CSF consistent with previous reports[26–28]. Collectively, MuPPE can achieve a precise and robust qualification at the glycoproteomic level. In contrast to a protein-rich biofluid serum, the low protein concentrations of CSF (0.15–0.45 mg/mL) pose additional challenges for glycosylation identification[29].

CSF is a biological sample with low protein concentration, posing significant challenges for efficient glycoproteome identification due to limited protein availability. Applying the MuPPE platform, we successfully identified an average of 5200 glycopeptides from just 50 µg of CSF protein (equivalent to ~142 µL of CSF) (Fig. 2h). Additionally, a significant overlap (~77%) of identified N-linked glycoproteins in human CSF across triplicates underscores the repeatability of the MuPPE workflow (Fig. 2h, inset). These results illustrated the improved glycopeptide enrichment capacity and glycosylation identification efficiency in CSF samples compared to previous benchmarks[29,30] (Supplementary Fig. 2). The CV distribution across different glycoproteins for N-linked glycan type quantification also demonstrated the robustness (Supplementary Fig. 3). The reliable quantification of various glycoproteins and their associated glycans further implies the effectiveness of MuPPE in glycosylation studies (Supplementary Fig. 3, inset). By comparing our identified N-glycosylation sites with UniProt annotations, we observed that 30.0% (MB), 12.4% (HS), and 9.8% (HCSF) of sites were not annotated in UniProt, expanding the annotated N-glycosylation site landscape (Fig. 2i). We attribute this enhanced performance to the optimized PAC step, where the addition of organic solvents promotes efficient protein aggregation and immobilization, thereby minimizing loss and significantly improving protein recovery prior to enzymatic digestion. Collectively, these results support the feasibility of incorporating glycosylation enrichment workflow into MuPPE.

## MuPPE benchmarking: validating simultaneous profiling of the proteome, glycoproteome, and phosphoproteome

Building on our previous findings that highlight the advantages of enriching glycopeptides prior to phosphopeptides with hydrophilic beads and commercial IMAC-Ti$^{4+}$ beads, respectively[31], we adopted this sequential order in all subsequent MuPPE enrichment. We also assessed the efficacy of integrating sequential phosphopeptides enrichment into the MuPPE workflow, utilizing mouse brain lysate as a model sample. The flow-through from the initial glycoproteome enrichment was reserved for subsequent phosphopeptides enrichment, as mentioned before. The comprehensive MuPPE workflow at three proteomic levels is represented in Fig. 3a. We investigated the effect of sample input amounts on identification at PTMs levels and a summary of the glycoproteome, and phosphoproteome results for different sample loading amounts was depicted (Fig. 3b). The identified numbers of glycopeptides, glycoproteins, phosphopeptides, and phosphoproteins showed a gradual increase as the sample amount increased from 5 to 75 µg; however, when the sample amount further increased from 75 to 125 µg, the identification numbers plateaued. Notably, even at 25 µg of MB lysate, the method achieved approximately 60% of the phosphopeptides, 66% of the phosphoproteins, 53% of the glycopeptides, and 57% of the glycoproteins identified at the 75 µg level.

To elucidate the distinct characteristics of various enrichment materials and the method's robustness following the integration of phosphopeptide enrichment into MuPPE workflow, we conducted a comparative analysis of enrichment properties for commonly used beads in phosphoproteomics, as well as the robustness of MuPPE. The study involved comparing different enrichment materials, such as TiO$_2$ and IMAC with distinct metal ions (Ti$^{4+}$ and Fe$^{3+}$, Supplementary Fig. 4). The findings revealed that the IMAC-Ti$^{4+}$ method yielded the highest number of identified phosphoproteins, highlighting its superior enrichment efficacy. Additionally, the workflow's robustness was demonstrated when executed by two operators using TiO$_2$, ensuring

reliable and consistent results. Besides the mouse brain, we also applied this MuPPE workflow to identify the glycoproteome, phosphoproteome, and proteome of the mouse liver, heart, and kidney, respectively. Notably, MuPPE demonstrated broad adaptability to diverse organ-derived samples, with each tissue type exhibiting distinct identification results across multiple proteomic levels (Supplementary Fig. 5).

We initially attempted to incorporate individual phosphopeptide enrichment materials from the protein aggregation in the workflow, but the outcomes were disappointing. The number of identified phosphopeptides remained low, and the selectivity for phosphopeptides was not satisfied. We infer that this phenomenon, which aligns with observations reported in other studies[32,33], demonstrated this protein aggregation instability, likely due to the charge properties of the phosphopeptide enrichment materials, adversely affected the subsequent enrichment results (Supplementary Fig. 6). Consequently, the optimal workflow is to utilize phosphopeptide enrichment materials exclusively during the phosphopeptide enrichment, rather than as initial protein aggregation materials. Collectively, the MuPPE workflow facilitates multi-level proteomic analysis by enabling sequential analysis of the glycoproteome, followed by the phosphoproteome from just single biological sample, previously only feasible through parallel processing of multiple sample aliquots. In multi-PTMs research, workflow duration is a critical parameter that directly determines overall efficiency. MuPPE excelled in time efficiency (total workflow <4 h vs. more than 32 h for others, Fig. 3c) and sample transfers (2 transfers vs. 5–10 transfers for others), positioning it as a leading method. These comparisons fully demonstrate the efficiency and advantages of MuPPE in multi-levels proteomic research. A radar plot (Fig. 3d) benchmarked MuPPE against five state-of-the-art methods[15,31,34–36], evaluating parameters such as on-bead digestion, trace sample applicability, sample compatibility, time efficiency, and minimized transfers numbers.

We compared MuPPE with conventional in-solution digestion protocols at the proteome level (Fig. 1), demonstrating its superiority in precursor, peptide, and protein identifications, as well as quantitative precision in serum samples. However, these comparisons primarily addressed proteomic profiling rather than systematic multi-PTMs benchmarking against dedicated enrichment strategies. To fill this gap, we evaluated MuPPE alongside four representative studies that reported simultaneous enrichment and identification of glycosylation and phosphorylation, including one work[15] that enabled broader multi-PTMs capture. We summarized and visualized the workflows of all platforms and presented a side-by-side schematic to highlight methodological distinctions (Supplementary Fig. 7). To ensure methodological comparability, we benchmarked MuPPE against the conventional in-solution approach and two previously reported strategies[31,34] under strictly matched conditions, including identical sample input, mass spectrometric acquisition parameters, database search settings, and data processing workflows. MuPPE yielded a substantially higher number of glycopeptide-spectrum matches than all three reference methods (Fig. 3e). Notably, the number of identified proteins and phosphopeptides were comparable between MuPPE and the other methods, indicating that the integrated multi-omics enrichment design did not compromise proteome or phosphoproteome coverage (Fig. 3f, g). The primary advantage of MuPPE in glycoproteomics lies in its enhanced glycan coverage and structural diversity, as well as its superior capacity to capture a greater proportion of medium-length and structurally complex glycopeptides (Fig. 3h, i). Peptide length distribution analysis (Fig. 3j) revealed that MuPPE achieved a more uniform distribution of identified peptide lengths and a higher overall number of peptides compared to the other three methods. This reflects MuPPE's superiority in capturing peptides across a broader size range, enabling more comprehensive proteomic coverage. Additionally, by comparing the overall

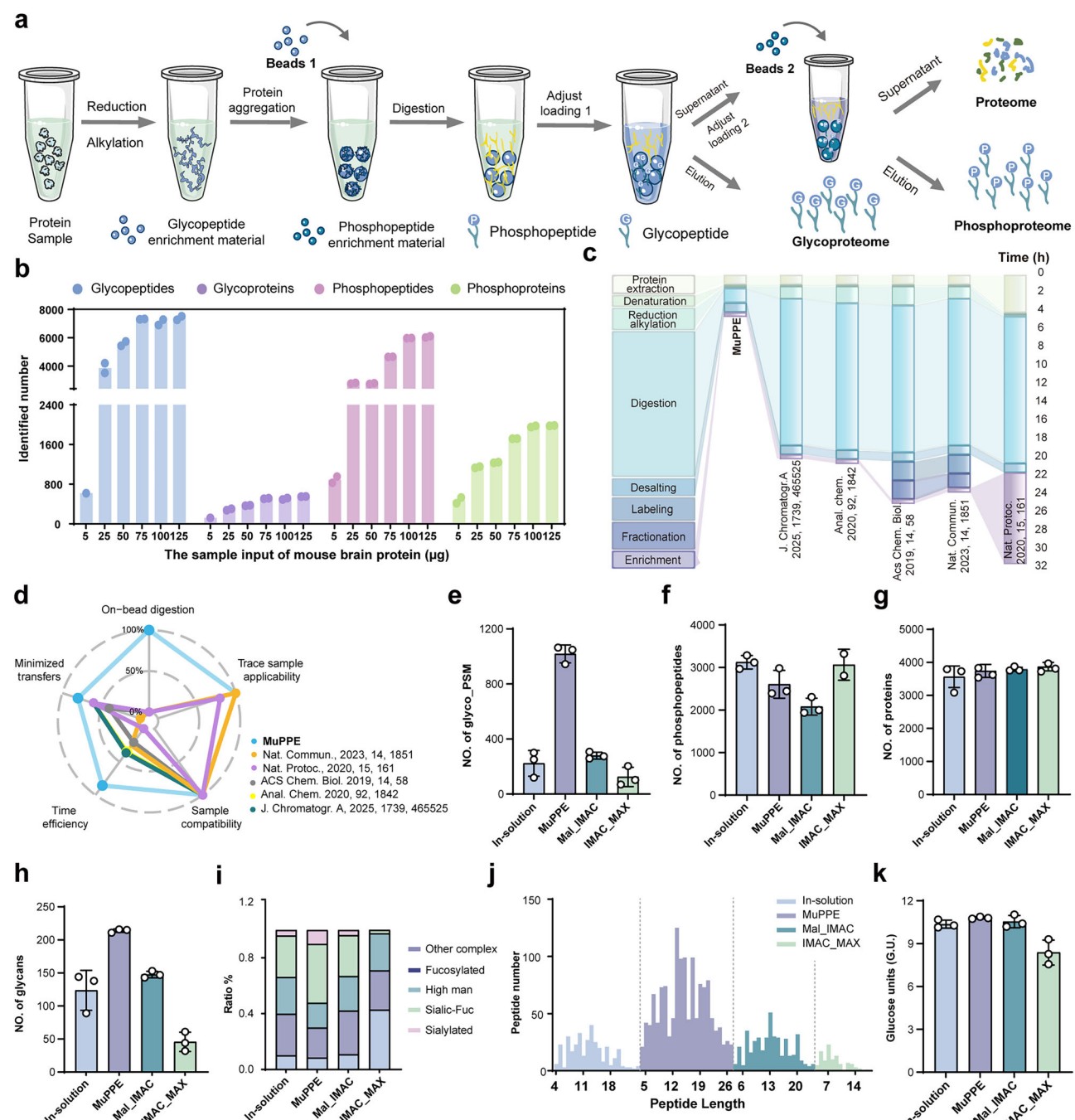

**Fig. 3 | Serialized analysis of glycoproteome, phosphoproteome and proteome using MuPPE. a** The detailed MuPPE workflow at three proteomic levels. **b** A summary of the proteome, glycoproteome, and phosphoproteome results for different mouse brain sample input amounts using Click-Mal and IMAC-Ti$^{4+}$ beads ($n = 2$, only one valid replicate at 5 μg due to low glycopeptide signal). **c** Step-by-step time breakdown of MuPPE workflow from protein extraction to enrichment, showcasing reduced sample transfers and time efficiency. **d** Radar plot comparing MuPPE with other multi-level proteomics methods across five parameters demonstrating MuPPE's superior performance. **e–h** Quantitative comparisons across enrichment methods with glycopeptide spectrum matches (PSMs) (**e**),

phosphopeptides (**f**), proteins (**g**), and glycans (**h**). **i** Distribution of glycan types identified across four methods, showing relative proportions of high-mannose, fucosylated, sialylated, and complex glycans. **j** Peptide length distribution for glycopeptides captured by MuPPE versus reported platforms. **k** Comparison of glycan structural features, including glycan unit number across enrichment strategies. For figure (**e–k**), $n = 3$ technical replicates per method, error bars presented as mean ± SD. For (**f**), only two valid technical replicates were obtained for the IMAC_MAX method due to one LC-MS/MS run failing quality control; therefore, the mean and SD were calculated based on these two replicates.

composition of glycan units identified by the four methods (Fig. 3k), MuPPE enabled improved characterization of glycan structural features. These combined results highlight MuPPE's enhanced capability to dissect the complexity of glycoproteomes at both peptide- and glycan-structural levels, providing deeper insights into glycosylation biology.

## MuPPE enables simultaneous profiling of proteome, glycoproteome, and phosphoproteome in the aging mice cohort

Aging is a natural biological process characterized by the gradual decline in physiological functions and an increased susceptibility to various age-related diseases[37,38]. Despite extensive research into the molecular mechanisms of aging based on genomics, transcriptomics,

and proteomics[39–42], there remains a significant gap in our understanding of how PTMs, such as phosphorylation and glycosylation, concurrently influence cellular processes in aging. This gap hinders our comprehensive understanding of the molecular mechanism of aging. In studies of physiological aging, certain mammalian brain regions yield minute tissue mass and limited peptide amounts, posing practical challenges for conventional proteomics input and statistical stability. To address this gap, we utilized the established MuPPE platform to perform a comprehensive and integrated analysis of phosphoproteome, glycoproteome, and proteome across different life stages, hoping to uncover new insights into aging-related mechanisms.

The choroid plexus (CP), responsible for CSF production and forming a selective barrier between the central nervous system and bloodstream, is affected by aging, which impact blood perfusion and microstructural integrity[43–45]. Yet a systematic view of age-dependent PTM changes in the CP remains lacking, owing to the limited input and analytical challenges posed by this minute tissue when using conventional enrichment workflows. To address this, we used the MuPPE platform to profile mouse CP (MCP), CSF (MCSF), and serum (Mserum) from 2-, 12-, and 24-month (M)-old cohorts (14 females and 14 males per group), generating matched phosphoproteome, glycoproteome, and proteome datasets (Fig. 4a). In the MCP samples, a total of 3845 N-glycosites were identified, corresponding to 30,153 unique site-specific glycan structures derived from 1,096 glycoproteins (Fig. 4b). In the MCSF and Mserum samples, we identified 3934 and 5442 unique site-specific glycan structures, respectively, associated with 511 and 510 unique N-glycosites from 200 and 197 N-glycoproteins (Supplementary Fig. 8). Approximately 45% of the identified glycosites in MCP samples were previously unreported in the UniProt database. Despite the previously extensive glycoproteomic analysis in serum samples[46], we still identified 14.9% of the glycosites as new. These results further demonstrate that MuPPE platform significantly expand the known glycosylation repertoire, offering valuable new data to advance understanding in this field (Supplementary Fig. 9). Figure 4c shows the distribution of high-mannose, fucosylated, and sialylated glycans on identified glycosites, which aligns with the patterns observed in previous study[26]. Supplementary Fig. 10 presents the percentages of proteins with identified single to multiple glycosites (left panel) and the distribution of glycans on these glycosites (right panel), emphasizing the macro- and micro-heterogeneity inherent in protein glycosylation. Notably, the levels of macro-heterogeneity in glycoproteins from MCSF and Mserum are roughly similar, though Mserum glycoproteins exhibit greater micro-heterogeneity, reflecting broader glycosylation diversity of different biological samples.

Moreover, the analysis of digalactosylation (the addition of two galactose residues) revealed distinct sex- and age-related differences (Fig. 4d). Galactosylation levels decreased with advancing age[47], with higher levels observed in females compared to males. Notably, females exhibited a pronounced decline in galactosylation during middle age, whereas males showed a more gradual reduction over time (Supplementary Fig. 11)[48]. We then performed differential expression (DE) analysis on the glycopeptides identified in any two groups of the three age groups. Supplementary Fig. 12 showed that after FDR correction, there was no significant difference in the expression of glycopeptides between the 24 M and 12 M groups, and the comparison between the other two groups showed that the upregulated glycopeptides mainly decorated with fucosylated and neutral unmodified complex/hybrid glycans. We assessed potential protein-level confounding by pairing each quantified PTM site with its parent protein and plotting site $\log_2$ fold-change versus protein $\log_2$ fold-change for each contrast (Supplementary Fig. 13). The changes in N-glycopeptide levels between groups were not significantly correlated with the corresponding unmodified protein levels ($R^2 = 0.21$). As a rule, when PTM regulation was not coupled to protein abundance (i.e., regulated sites with parent proteins showing no detectable change), we performed downstream

analyses on site-level intensities after standard sample-level normalization. Because dividing by protein restricts coverage to sites with complete protein co-quantification, protein-division was not used for the primary results.

Additionally, Kyoto Encyclopedia of Genes and Genomes (KEGG) analysis revealed that pathways such as extracellular matrix-receptor interaction, focal adhesion, and PI3K-AKT signaling were enriched among positively regulated glycoproteins. Conversely, the pathways related to autophagy and lysosomes were enriched among negatively regulated glycoproteins. (Supplementary Fig. 14). Multiple identical upregulated glycopeptides in both Mserum and MCSF is associated with the selective permeability of the blood-brain barrier allowing their bidirectional exchange between the two[49]. Then an upset plot was generated to illustrate the distribution of upregulated glycopeptides across MCP, MCSF, and Mserum samples (Fig. 4e). *Ighm* represents the initial antibody response from B cells upon encountering the infection[50,51] and four *Ighm* glycopeptides (*Ighm*-280-H2N5; *Ighm*-280-H6N3F1; *Ighm*-280-H4N4F1; *Ighm*-280-H3N3F1 and *Ighm*-280-H3N4F1) significantly upregulated across all three sample groups. Notably, they are characterized by the presence of bisecting GlcNAc in their core-fucosylated glycans. This finding suggests that the core-fucosylated glycans attached to immune-related proteins are pivotal in the context of aging, acting as potential significant biomarkers for the study of aging and age-related diseases. Therefore, alterations in the glycosylation patterns of *Ighm* could be indicative of shifts in immune function, shedding light on immune deficiencies associated with aging or alterations in immune responses[52,53].

To gain systematic insights into PTMs in the mouse brain and their changes related to aging, we performed PTMs coregulation network analysis by using the weighted correlation network analysis (WGCNA). Then we performed module-trait association analysis to assess the correlation between each module PTM and aging-relevant phenotypic traits and other characteristics of MCP samples. For glycans, the eigengene dendrogram (Fig. 4f, bottom) summarizes inter-module relatedness and guided module merging (mergeCutHeight = 0.25). Trait associations were then tested by correlating module eigengenes with Age and Sex (Fig. 4g). The network consists of one positively correlated glycan module (MEblue) and one negatively correlated glycan module (MEyellow) with significant association to aging status but no association with sex genotype (Fig. 4f, g). MEblue contained 30 glycans, including undecorated complex/hybrid glycans, pauci-mannose glycans with relatively simple, neutral structures (H2N2 and H3N2, H4N2, H3N3, H3N4, and H3N6) and bisecting GlcNAc-containing core-fucosylated glycans (H3N4F1, H3N5F1, and H3N6F1)[54,55]. In contrast, the MEyellow module with a strong negative correlation to aging ($P < 0.01$) contained 23 glycans (Fig. 4h), which primarily consisted of sialofucosylated glycans (H9N8F1Sa3, H8N7F1Sa2Sg1, and H9N6F1Sa1Sg1), and highly branched and/or elongated glycans (H4N7, H6N6Sa2, and H6N6Sa1). The above results suggest that decreased sialylation and N-glycan branching or elongation is involved in aging.

In our analysis of the MCP samples, we identified 1823 phosphoproteins with 3779 phosphosites (Fig. 4b). A comparative DE analysis of the elderly and young groups showed that phosphosites upregulation peaked at 12 M (Supplementary Fig. 15). The upset map plot revealed a dynamic pattern in phosphosite expression levels, with some initially downregulated then upregulated, and others the opposite (Fig. 5a). The changes in phosphosite relative abundance are mainly due to alterations in phosphorylation modification levels, rather than changes in protein expression (Fig. 5b). As for phosphosites, WGCNA analysis uncovered 22 co-regulated phosphosite modules (Supplementary Fig. 16). Among these, 4 phosphosite modules significantly associated with age, including 1 positively correlated module and 3 negatively correlated modules ($P < 0.01$). Only one module showed a strong association with sex. We selected the modules with the strongest correlation with age (MEpurple and MEsalmon)

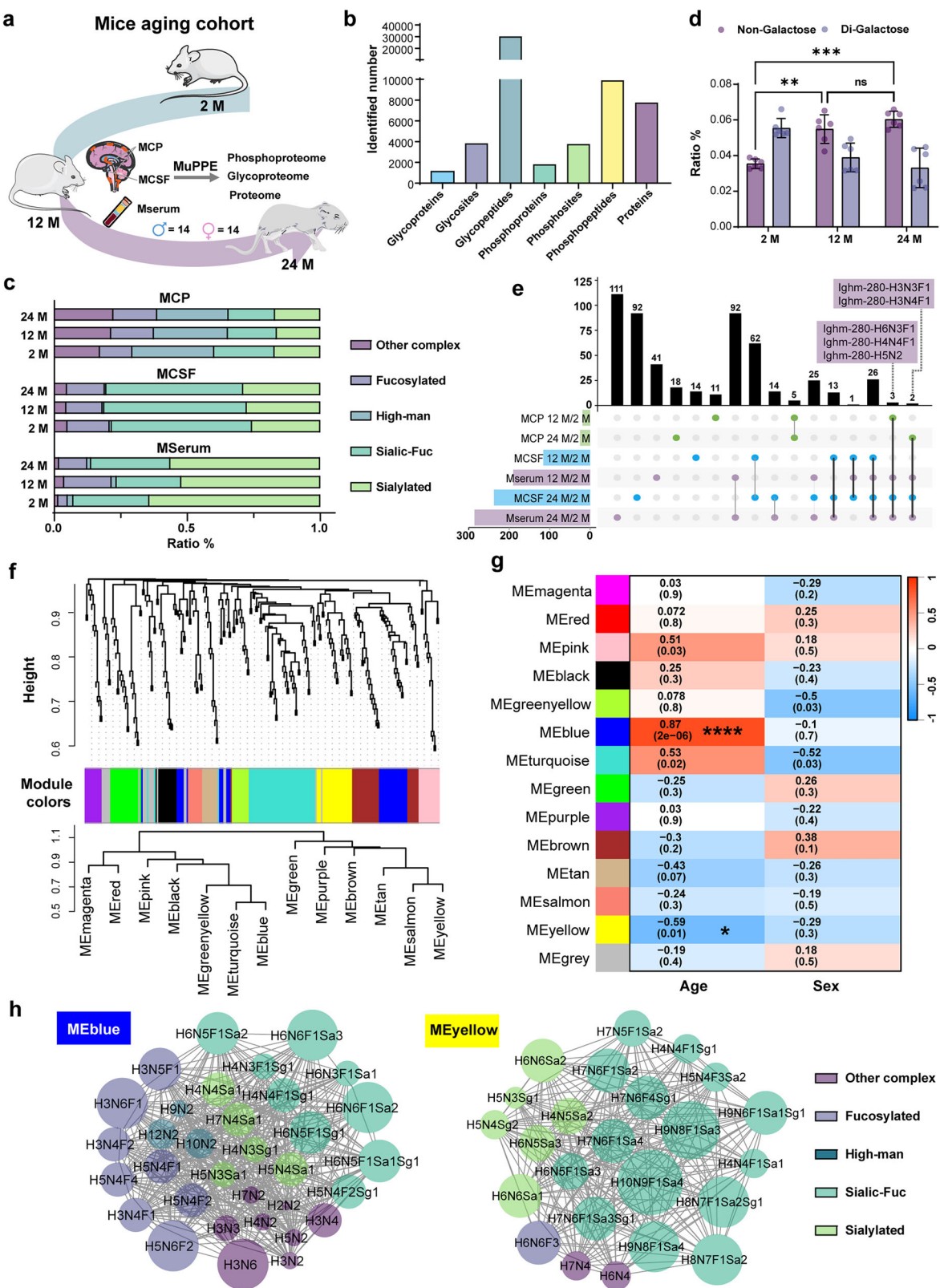

and the module with the strongest correlation with sex (MEred) for Gene Ontology (GO) analysis in Supplementary Fig. 17. The results showed that the modules positively correlated with age were enriched in the functions related to mRNA, RNA processing and splicing, while the negatively correlated modules were most enriched in actomyosin structure organization, negative regulation of stress fiber assembly and neuronal cell homeostasis. Interestingly, the modules related to

sex showed differences in learning, memory and synaptic function. We further conducted site-level pathway enrichment using PTM-SEA, which infers PTM regulators from substrate-specific signatures[56]. As shown in Supplementary Fig. 18, compared with 2 M controls, 24 M samples displayed positive enrichment of the PI3K-AKT pathway and kinase-centric sets (*AKT1, CDK2, SIK2, MAPKAPK5/RSK2*), with higher normalized enrichment scores and significance than 12 M, indicating

**Fig. 4 | Intact glycopeptide-based glycoproteome analysis of mice aging cohort. a** Schematic overview of MuPPE workflows used to investigate the aging mice cohort. The study cohort included 24, 12, and 2 M mice (*n* = 28 and 14 males and 14 females per group) with the paired Mserum, MCP, and MCSF. Image(s) provided by Servier Medical Art (https://smart.servier.com), licensed under CC BY 4.0 (https://creativecommons.org/licenses/by/4.0/). **b** Overview of identified glycoproteome, phosphoproteome, and proteome obtained from MCP. **c** Percentage distribution of glycan structures observed across the cohort. **d** Comparison of relative abundances of galactosylation in MCP samples, comparison by ANOVA followed by Tukey's two-sided multiple comparisons test (*P* < 0.05, **P* < 0.01), horizontal bars indicate median values. Error bars presented as mean ± SD, *n* = 6. **e** Upset plot illustrating significantly increased glycopeptides in Mserum, MCP, and MCSF, showing both unique glycopeptides for specific age windows and overlap among sample types. **f** WGCNA cluster dendrogram of N-glycan modifications, revealing thirteen color-coded modules and their inter-modular connectivity. **g** Heatmap showing correlations between each glycan module and age or sex, represented by Biweight midcorrelation coefficients (bicor *r* values). Numbers indicate the bicor correlation coefficients with corresponding two-sided *P* values (Student's *t*-test) in parentheses. Significant positive (red) and negative (blue) correlations are denoted by asterisks (*P* < 0.01; ****P* < 1 × 10$^{-5}$). *P* values were adjusted for multiple comparisons using the Benjamini-Hochberg method. **h** Glycan co-regulation network plots of aging-associated modules MEblue and MEyellow, where nodes represent individual glycan compositions. Each color represents a different type of modification. The size of a node indicates its degree of connectivity within the network. H: hexose, N: N-acetylhexosamine, F: fucose, Sa: N-Acetylneuraminic acid, Sg: N-Glycolylneuraminic acid.

progressive, age-dependent activation. Conversely, *CK2/CSNK2A2, NLK*, and *ERK5/MAPK7* signatures were consistently negatively enriched at 12 M and 24 M. Together, these results indicate a shift toward enhanced AKT/cell-cycle kinase activity with a concomitant attenuation of CK2-linked regulation during physiological aging[57]. Consistent with GO results, PTM-SEA indicates age-progressive activation of AKT/ RSK-cellcycle signaling linked to RNA/spliceosomal regulation, with concomitant attenuation of *CK2/ERK5/NLK* pathways that support actomyosin architecture and neuronal/synaptic homeostasis.

Alterations in glycosylation enzymes and regulators are associated with various pathological processes, including the aging. However, the role of these changes in driving N-glycosylation and their broader implications for aging-related decline in brain function remain poorly understood. Here, we assessed the protein expression patterns of glycosylation enzymes and regulators in the MCP samples based on the proteome (Fig. 5c). In the juvenile stage, accelerated development is supported by *Ctbs* and *Gba2*, which facilitate extracellular matrix (ECM) remodeling through glycosaminoglycan and glycolipid degradation[58], while *Fut11* refine glycan complexity to optimize cell adhesion and signaling[59]. Glycosidases *B3galt5, B3galt6, St6gal1*, and *St6galnac2*, which are responsible for elongation and sialylation, are highly expressed during middle age and may enhance ECM stability and support tissue repair[60]. *Alg1, Alg2*, and *Alg5* enzymes involved in the synthesis of precursor oligosaccharides for N-glycosylation, play a critical role in proper protein folding and endoplasmic reticulum (ER) quality control. Their high expression in the elderly stage indicates increased ER stress[61]. Concurrently, the elevated expression of *Erlec1*, a key protein in the ER-associated degradation pathway, suggests an enhanced mechanism to alleviate ER burden by targeting misfolded glycoproteins for degradation[62]. This coordinated response underscores the cellular effort to maintain proteostasis and address the challenges of ER stress associated with aging.

Furthermore, our findings indicate that certain glycosidases exhibit abnormal phosphorylation regulation as age progressing (Fig. 5d). For instance, glycogen synthase (*Gys1*), which transfers glycosyl residues from UDP-Glc to the non-reducing end of α-1,4-glycans, shows increased phosphorylation levels with aging, resulting in reduced activity[63]. Calnexin, an ER lectin chaperone that binds to monoglycosylated glycans from glucosidase II to maintain ER quality control by retaining misfolded proteins and promoting protein folding, shows a decline in phosphorylation levels with age[64]. The interplay between distinct PTMs, such as phosphorylation, glycosylation[65], and acetylation[66], is a confirmed phenomenon that profoundly influences protein function. Here, we identified 205 instances of co-occurring N-glycosylation and phosphorylation events on the same protein in CP and investigated their correlation. In 24 M mice, 20 modification pairs demonstrated a significant positive correlation (FDR < 0.05), underscoring potential age-dependent regulatory correlation between these modifications (Fig. 5e). The most statistically significant example (FDR = 4.29E-4) involves phosphorylation at S794 and N-glycosylation

at N264 on *Adam17*, a disintegrin-metalloprotease-domain sheddase. *Adam17* drives inflammatory signaling by converting pro-TNF-α to its soluble form and by shedding EGFR ligands, and sustained activation of this axis contributes to chronic inflammation-a hallmark of aging (Supplementary Fig. 19)[67].

To investigate aging-related molecular changes, we conducted time-series clustering of the proteome, phosphoproteome, and glycoproteome across 2, 12, and 24 M, followed by KEGG pathway enrichment analysis (Fig. 5f). In the phosphoproteome, pathways associated with aging, including oxidative phosphorylation, Alzheimer's disease, and Parkinson's disease, were enriched early in young mice and remained detectable throughout all age groups. The proteome exhibited significant enrichment in pathways such as long-term depression and peroxisome function, specifically at 24 M, suggesting a delayed accumulation compared to phosphorylation-mediated changes. The ECM-receptor interaction pathway showed distinct regulatory patterns: glycoprotein levels increased progressively with age, while phosphorylation peaked at 12 M before declining by 24 M. Enrichment of the cytoskeleton pathway in muscle cells was observed in both the proteome and phosphoproteome, with an initial increase followed by a decrease; phosphorylation changes, however, declined more gradually. Several pathways were uniquely altered in specific PTM layers: focal adhesion decreased with age in the phosphoproteome, whereas oxidative phosphorylation was broadly enriched. In the glycoproteome, the lysosome pathway increased with age, along with upregulation of antigen processing and ECM-related processes. Overall, these multi-layer proteomic profiles reveal temporally distinct and pathway-specific modulation of PTMs during aging, highlighting the value of integrated PTM mapping in understanding molecular aging mechanisms.

Multiple proteomic comparisons among MCP, MCSF, and MSerum (Fig. 5g) reveal that CP aging contributes to impaired integrity of the blood-CSF barrier (B-CSFB) and neuroinflammation, characterized by glial activation (high *Gfap* expression)[68], tissue remodeling (elevated *IL11*[38] in CP), and enhanced systemic and central inflammation regulation (elevated *IL18BP* in serum and CSF). Although the B-CSFB is primarily composed of epithelial cells in the CP, the observed changes in endothelial markers (e.g., *Vwf*[69] and *CD31*[70]) may reflect the influence of vascular-associated processes on the microenvironment of the CP. These alterations could indirectly impact barrier integrity, highlighting the interplay between epithelial and vascular components in maintaining the functional stability of the B-CSFB. Tight junction changes show clear time-dependent patterns. Claudin-1 (*Cldn1*) and Claudin-5 (*Cldn5*) are significantly upregulated at 12 M but decline at 24 M, indicating mid-stage barrier strengthening followed by late-stage dysfunction. Claudin-2 (*Cldn2*) exhibits dynamic regulation, while Claudin-12 (*Cldn12*) shows sustained age-related upregulation, potentially compensating for barrier alterations (Supplementary Fig. 20). These findings highlight the complex regulation of tight junction proteins in the CP during aging, providing key insights into B-CSFB dysfunction[71].

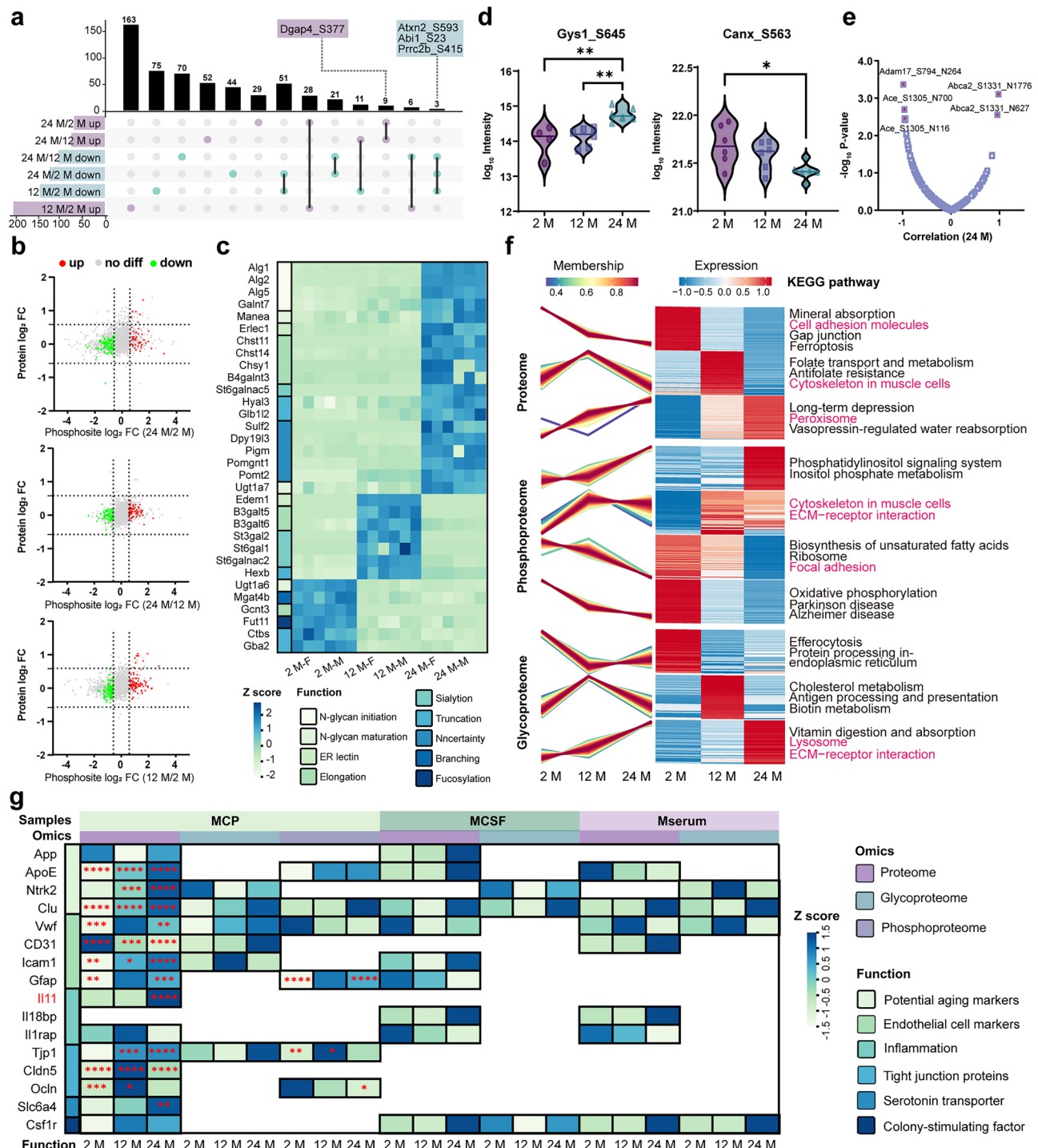

**Fig. 5 | Phosphoproteome, glycoproteome and proteome integrative analysis of mice aging cohort. a** Upset plot depicting significant phosphosites. **b** Median phosphosite fold change compared to the protein fold change in each compared group. **c** Heatmap of protein-level expression of MCP glycosidases across the different age groups. **d** Phosphorylation expression profiles of glycosidases *Gys1* and *Canx* in each age group of MCP, comparison by ANOVA followed by Tukey's two-sided multiple comparisons test, horizontal bars indicate median values (*P < 0.05, **P < 0.01). **e** Glycosylation/phosphorylation sites correlation in 24 M group.

**f** Integrated KEGG enrichment analyses of proteome, phosphoproteome, and glycoproteome datasets. **g** The heatmap of changes in proteins, their glycosylation, and phosphorylation during aging in MCP, MCSF, and Mserum. Each group was analyzed using analysis of variance (ANOVA) followed by Tukey's two-sided post-hoc multiple comparison test. Statistical significance between the groups (12 M/2 M, 24 M/12 M, and 24 M/2 M) is indicated by asterisks as follows: *P < 0.05, **P < 0.01, ***P < 0.001, ****P < 0.0001.

Totally, our findings reveal that aging is associated with significant remodeling of glycosylation and phosphorylation patterns, which lead to ECM instability, ER stress, and chronic inflammation. The proteomic changes also impact B-CSFB integrity, increasing permeability and promoting neuroinflammation. By linking specific molecular changes to aging-related dysfunctions, this study advances our understanding of the complex regulatory networks underlying brain aging and highlights potential biomarkers and targets for therapeutic intervention.

These findings, derived from our MuPPE platform, highlight its potential for advancing research in neurodegenerative diseases and therapeutic interventions.

## Multi-dimensional insights of arsenic MoA revealed by MuPPE

Arsenic therapy has shown remarkable efficacy in treating acute promyelocytic leukemia (APL) by inducing apoptosis and promoting the differentiation of malignant cells[72,73]. However, the specific mechanisms behind these effects, particularly regarding PTMs, have not yet been clearly reported. Further investigation at different molecular layers is essential to elucidate these mechanisms and advance our understanding of arsenic's therapeutic potential in leukemia treatment. Based on the feasibility of MuPPE, we subsequently investigated the MoA of the arsenic on NB4 cell line. The NB4 cells were incubated with the arsenic of different concentrations (0 - 10 μM) for 6 h before cell lysis and then the samples were transferred to the MuPPE workflow for further multi-level proteomic profiling (Fig. 6a).

This whole MuPPE workflow was achieved within 4 h and totally ~5600 protein groups were identified. Further quantitative analysis revealed that 105 protein groups exhibited differential expression. Among these proteins, 15 differentially expressed protein groups were consistent with previous report, including the thioredoxin-like protein 1 emphasized in previous study (Fig. 6b)[66]. We also observed a distinct molecular landscape at the PTMs-level compared to the global proteome. Consistently, further analysis revealed that glycosylated and phosphorylated sites exhibit more abundant differential expression, thereby providing valuable insights at the PTMs-level (Fig. 6c, d). These findings align with the expectation that PTMs-based proteomic drug profiling is more informative than conventional single proteome analyses. Over 30% of glycoproteins with differentially expressed glycopeptides were categorized as enzymes (Fig. 6c, insets), while kinases accounted for a higher proportion of proteins associated with upregulated phosphorylated peptides (Fig. 6d, insets). These observations implicate potential functional variations in enzymes. It is worth mentioning that as the drug concentrations increase, the number of glycopeptides and phosphopeptides changes exhibits an opposite trend, while the total number of protein groups remains almost unchanged (Fig. 6e middle). Among the total glycopeptides and phosphopeptides identified, the number of differentially regulated ones tends to increase in a drug concentration-dependent manner. (Fig. 6e left and right). We also focused on glycopeptides and phosphopeptides that emerged or vanished upon drug treatment. As the arsenic concentration increased, the number of vanished glycopeptides exceeded that of emerged ones, whereas phosphopeptides showed the opposite trend. This indicates that these two PTMs are differentially regulated by the arsenic (Fig. 6f). Additionally, high-mannose glycans (Man5 to Man9) exhibited significant structural alterations (Fig. 6g). Kinase-substrate enrichment analysis (KSEA) revealed that arsenic treatment regulated a higher proportion of kinases from the AGC and CMGC families (Fig. 6h).

Building on the multi-omics dataset analysis, we propose a mechanistic model where arsenic exerts its effects through a membrane-to-nucleus signaling cascade (Fig. 6i), primarily mediated by glycosylation-modulated membrane proteins and phosphorylation-driven PI3K-AKT-mTOR pathway perturbation. The glycoproteomic data revealed that arsenic induces glycosylation alterations in membrane-anchored receptors (e.g., components of the PI3K-AKT-mTOR pathway upstream regulators). These glycosylation changes might act as molecular switches to modulate membrane protein stability, localization, and ligand-binding capacity. We also observed that the arsenic treatment significantly impacted the cell cycle and chromosomal organization pathways, particularly in the regulation of the proteome and phosphoproteome during the M phases. This suggests that arsenic exerts profound effects on these two critical signaling pathways, potentially by arresting the cell cycle and disrupting

chromosomal functions, thereby promoting apoptosis. Flow cytometry analysis (Fig. 6j and Supplementary Figs. 21 and 22) revealed that arsenic-treated NB4 cells accumulate in the G2/M phase, which is consistent with the phosphorylation changes in cell cycle regulators (e.g., CDK, CCNB proteins in Fig. 6i), directly supporting this G2/phase arrest. GO enrichment results of nuclear functions in the key modules, combined with phosphoproteomic changes in chromatin-associated proteins (e.g., histone modifiers), suggest arsenic might disrupt genome-wide transcriptional programs and link membrane-to-nucleus signaling to nuclear architecture and function.

Focusing on *IGF2R*, a key membrane protein identified in the PI3K-AKT-mTOR pathway, we interrogated its individual proteomic, glycoproteomic, and phosphoproteomic alterations (Fig. 6k). At the global protein level, *IGF2R* abundance showed no significant fold-change between arsenic-treated and control groups. However, its glycoproteomic and phosphoproteomic data revealed notably PTMs-specific alterations. We identified 13 differential PTMs sites, including 12 N-glycosylation sites and 1 phosphorylation site. Upregulated and downregulated PTMs were denoted by upward and downward triangles, respectively. Without glycoproteomic/phosphoproteomic data, *IGF2R* would appear unchanged at the protein level, masking its potential role as a PTMs-driven hub. Thus, arsenic exerts its effects not by altering protein abundance broadly but by rewiring PTM-mediated signaling networks-a mechanism only detectable via multi-omics profiling.

## Discussion

MuPPE represents a transformative advancement in proteomic workflows, addressing long-standing challenges in multi-PTMs proteomic analysis. By enabling sequential glycoproteome, phosphoproteome, and proteome profiling from a single biological sample, it overcomes the limitations of traditional approaches that suffer from sample loss and inability to capture inter-PTMs regulatory relationships. The MuPPE workflow, with its 4 hours processing time, is a significant improvement over conventional methods that often take more than 32 hours. This efficiency is not achieved at the cost of sensitivity. For example, in low-input samples such as CSF, the protein aggregation step allows for the concentration of proteins from as little as 50 μg of CSF protein. This is crucial for applications where sample availability is limited, like in clinical settings with pediatric CSF or needle biopsies. In benchmarking against state-of-the-art techniques, MuPPE demonstrates superior performance in glycoproteome, phosphoproteome, and proteome identifications. By first capturing glycoproteomes via Click-Mal beads and then phosphoproteomes via IMAC-Ti⁴⁺ from a single sample aliquot, it minimizes sample loss. This is a stark contrast to parallel PTMs enrichment methods that often require multiple sample aliquots, leading to increased variability and reduced data integration. The ability to profile different PTMs from the same starting material also allows for the exploration of inter-PTMs regulatory networks, which was previously difficult to achieve.

When applied to aging mouse tissues, MuPPE has revealed temporal and tissue-specific PTM changes that are associated with the aging process. The identification of age-related alterations in glycoprotein glycosylation patterns and phosphoprotein phosphorylation events has shed light on the molecular mechanisms underlying aging. For example, changes in specific glycoprotein glycosylation in different tissues may be related to tissue-specific functional decline during aging. These insights into PTMs changes during aging also open up new avenues for anti-aging interventions. By identifying key PTMs that are dysregulated with age, it may be possible to develop therapeutic strategies to reverse or slow down these processes. MuPPE can be used to evaluate the efficacy of potential anti-aging compounds by monitoring their effects on PTMs profiles in aging tissues.

In the study of arsenic-treated NB4 cells (APL model), MuPPE has refined our understanding of the drug's mechanism of action.

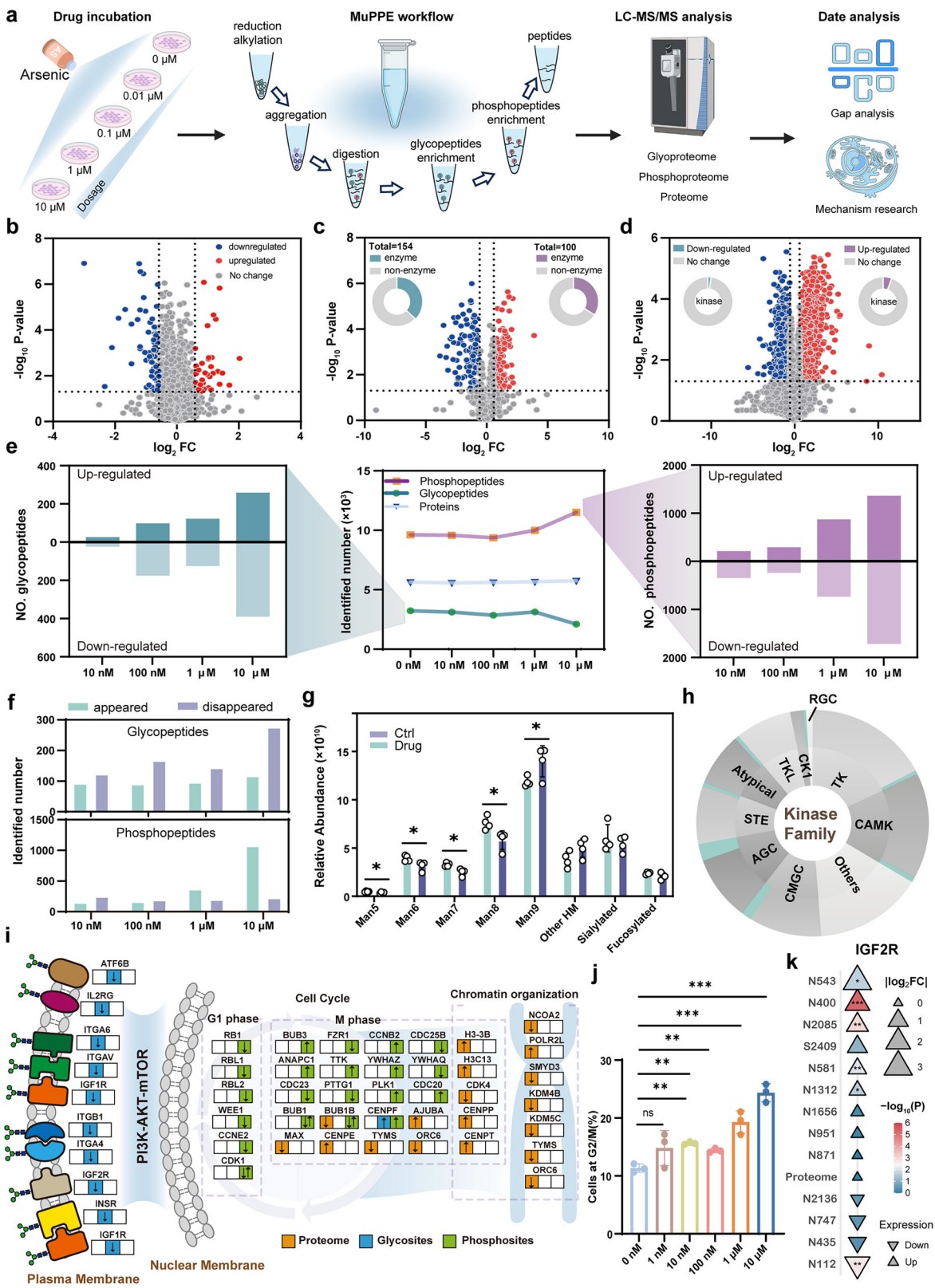

The identification of glycoproteomic and phosphoproteomic changes has revealed previously uncharacterized regulatory pathways. For instance, changes in glycosylation and phosphorylation events in key signaling molecules have been uncovered. These findings go beyond what traditional single-PTMs or global proteome analyses could achieve, as they link different PTMs to specific cellular responses to arsenic. The ability of MuPPE to dissect the complex PTMs-driven regulatory networks in drug-treated cells has significant implications for drug development. It can help in identifying new drug targets, understanding drug resistance mechanisms, and optimizing drug dosing. By providing a more comprehensive view of how drugs affect the proteome at the PTMs level, MuPPE can accelerate the development of more effective and targeted therapies.

**Fig. 6 | Global proteomic, glycoproteomic, and phosphoproteomic analyses reveal the differentially regulated molecular features of arsenic MoA. a** The overview of MuPPE workflow for the dose-dependent treatment of NB4 cells with arsenic. Created in BioRender. Dong. (2025) [https://BioRender.com/b218mw3]. **b** Volcano plot depicting differentially expressed proteins upon arsenic treatment (10 μM vs 0 μM). **c** Volcano plot showing differentially regulated glycosylation sites following arsenic exposure (10 μM vs 0 μM). Inset: relative proportion of enzymes versus non-enzymes among glycoproteins with differential glycopeptide expression. **d** Volcano plot illustrating differentially regulated phosphorylation sites after arsenic treatment (10 μM vs 0 μM). Inset: percentage of kinases within proteins exhibiting differential phosphorylation. Downregulated entities are depicted in blue, upregulated in red, and unchanged in grey. **e** The counts of differentially regulated glycopeptides (left panel) and phosphopeptides (right panel), and total numbers of identified proteins, glycopeptides, and phosphopeptides across varying drug concentrations (middle panel). **f** Number of newly emerged or vanished glycopeptides and phosphopeptides compared with control group (0 μM).

**g** Alterations in glycan structural profiles following arsenic treatment (10 μM vs 0 μM); HM = high-mannose glycans; comparison by two-sided $P$ values (Student's $t$-test), $*P < 0.05$, $**P < 0.01$. Error bars presented as mean ± SD, $n = 4$. **h** Differential regulation of kinases by family, as identified through KSEA. Cyan-shaded regions denote the proportion of regulated kinases within each family. **i** Schematic of PI3K-AKT-mTOR pathway perturbation by arsenic, with downstream effects on cell cycle and chromatin organization. Proteome (green), glycoproteome (orange), and phosphoproteome (blue) alterations are mapped. Created using ChemDraw (version 14.0). **j** Flow cytometry analysis of NB4 cells treated with increasing arsenic concentrations, showing dose-dependent G2/M phase accumulation comparison by two-sided $P$ values (Student's $t$-test), $*P < 0.05$, $**P < 0.01$. Error bars presented as mean ± SD, $n = 3$. **k** Differential expression of protein relative abundance, glycosylation sites, and phosphorylation sites of IGF2R protein with the $\log_2$ fold change values. Different symbols representing upregulation (upward triangles), downregulation (downward triangles), and the color intensity indicating the significance of the change.

Despite its many advantages, MuPPE has some limitations. Currently, it is mainly focused on N-glycosylation and phosphorylation. Extending it to other PTMs such as acetylation and ubiquitylation will require further optimization of enrichment processes. In the future, we aim to expand the scope of MuPPE to include other PTMs. This will involve developing and optimizing new enrichment strategies that can be integrated into the existing workflow without compromising its efficiency and sensitivity. Additionally, we plan to couple MuPPE with advanced data analytics tools to better handle the complex multi-PTMs data and to identify more subtle inter-PTMs regulatory relationships. Another direction is to adapt MuPPE for single-cell proteomics, which will require further miniaturization and optimization of the workflow to handle the low protein content of single cells.

In conclusion, MuPPE is a powerful tool for multi-PTMs proteomic analysis that has already provided valuable insights into drug mechanisms and aging biology. With further development and refinement, it has the potential to make even greater contributions to our understanding of complex biological processes and to the development of novel therapeutic strategies.

## Methods

### Cell lines, mouse samples, and human samples source

NB4 cell line: NB4 acute promyelocytic leukaemia cells was purchased from the American Type Culture Collection (Manassas, VA) and cultured in DMEM (Gibco) containing 10% FBS (Gibco), 100 units/mL penicillin, and 50 units/mL streptomycin at 37 °C in a $CO_2$ incubator (95% relative humidity, 5% $CO_2$). All reagents were purchased from Sigma unless otherwise indicated.

The study adhered to the guidelines set by the committee. For all of our animal experiments, we used mixed sex groups in the same ratio of male and female C57BL/6J mice in the range of 20-30 × $g$ of body weight up to 24 M of age. Animals were kept under standard conditions in a specific pathogen-free facility at 20-24 °C and 45-65% humidity on a 12 hours light/dark cycle and had access to food and water ad libitum. Mouse brain, heart, liver, and kidney tissues (utilized for MuPPE performance validation), along with mice CP, CSF, and serum (used for MuPPE practical applications), were carefully dissected and stored at −80 °C. For detailed implementation steps, please refer to the Supporting Information (SI).

### Processing of different biological samples

To obtain cell lysate from untreated and treated cells with the arsenic (0 or 10 μM), cell suspension was centrifuged at 1000 × $g$ for 5 min at 4 °C, washed with cold PBS (phosphate-buffered saline, without calcium or magnesium) and pelleted before resuspension in lysis buffer (20 mM Tris, 150 mM KCl, 10 mM $CaCl_2$, pH 8, 1% cocktail). Subsequently, the cells were lysed using lysis buffer through three freeze-thaw cycles in liquid nitrogen. The lysis buffer was snap-

frozen, followed by thawing at 37 °C using a metal bath until about half of the suspension was thawed. Then the samples were transferred onto ice until the entire content was thawed, followed by sonication using a probe. Cell debris was then removed by centrifuging at 14,000 $g$ for 5 min at 4 °C and the protein concentration was tested by using a BCA protein assay kit (Thermo Fisher Scientific, San Jose, CA, USA).

Frozen mouse tissues were resuspended in ice-cold 20 mM HEPES, 250 mM KCl, 250 mM NaCl, 10 mM $CaCl_2$, pH 7.4 buffer, and subsequently homogenization was achieved using ceramic beads and an automated homogenizer with 2 × 15 s pulses, followed by sonication using a probe. Homogenates were clarified by centrifugation for 10 min at 14,000 $g$ at 4 °C and protein concentration was determined by the BCA protein assay kit.

The human or mice CSF and serum samples were diluted fourfold and 50-fold, respectively, with 1% SDS in 100 mM ammonium bicarbonate. Protein concentration was determined using a Nanodrop spectrophotometer, and 50 μg of protein from each sample was prepared for subsequent analysis on the MuPPE platform.

### MuPPE process

The supernatant of cell lysate, tissues lysates, or other body fluid samples were immediately reduced with dithiothreitol (DTT, final concentration 10 mM) and heated for 10 min simultaneously, and then alkylated with iodoacetamide (IAA, final concentration 40 mM) in dark for 30 min before further processing. Aggregation was induced by the addition of acetonitrile and different beads were added to solution followed by mixing the beads solution uniformly. The mixed solution was allowed to settle for 10 min and then the beads were separated through centrifugation. After washed with 80% ($v/v$) ethanol, the beads were retained and the supernatant was removed. For eluting the beads bound protein aggregates, the 8 M urea solution was added and the samples were subjected to ultrasonic treatment for better solubilization. Then, the mixture was diluted with 5 mM $NH_4HCO_3$ to reduce the urea concentration below 2 M. Trypsin (enzyme-to-protein ratio, 1:50, $w/w$) was added for digestion at 37 °C. For samples with higher protein content (e.g., cell lysates, tissue homogenates), digestion was performed for 4 hours to ensure sufficient proteolysis of complex protein components. For samples with lower protein concentrations (e.g., CSF), a shorter digestion time of 2 hours was used to avoid unnecessary over-digestion. And then the obtained samples were used for subsequent enrichment experiments based on individual experimental purposes. The detailed enrichment procedures were provided in Supplementary Information.

### Proteome identification

The peptides of flow-through from phosphopeptides or glycopeptides enrichment procedure for proteomic profiling were desalted using the

C18 cartridge and dried down, and reconstituted in 0.1% FA prior to LC-MS/MS analysis.

## In-solution digestion of human serum

One microliter of human serum was diluted 10 times with 8 M urea in 50 mM $NH_4HCO_3$ and denatured by reduction with 20 mM (final concentration) DTT for 45 min and alkylation with 40 mM (final concentration) IAA for 30 min at room temperature. Then, the mixture was diluted with 50 mM $NH_4HCO_3$ to reduce the urea concentration below 2 M. Then the sample was digested with trypsin at an enzyme-to-protein ratio of 1:50 (*w/w*) at 37 °C overnight. Samples were acidified with formic acid at 1% (*v/v*) final concentration and the tryptic peptides were subsequently desalted using the C18 cartridge and then lyophilized.

## MS data analysis

For proteomic data processing, DIA spectra were analyzed with Spectronaut (v19.0.240606.62635) using the default settings. Briefly, digestion enzyme specificity was set to Trypsin/P and specific. Up to 2 missed cleavages were allowed. Search criteria included carbamidomethylation of cysteine as a fixed modification, as well as oxidation of methionine and acetylation (protein N-terminus) as variable modifications. The false discovery rate (FDR) was set to 1% at the PSM, peptide, and protein level. Precursor and protein Q value cutoff was set to 1%. DIA files were analyzed in Direct-DIA mode, thus without spectral library, using the same settings for quantitative analysis. Direct-DIA was performed by DirectDIA⁺ Deep.

Glycoproteomic data were processed using Byonic software (v5.4.10, Protein Metrics) with a built-in database of 309 human N-glycans. The parameters for Byonic were set as the following: precursor mass tolerance was 6 ppm; fragment mass tolerance was 20 ppm; The digestion specificity was tryptic fully specific. The missed cleavages were set 2. Carbamidomethylation at C was set as fixed modification. Oxidation at M and deamidated at N were set as variable modifications. The total common max and total rare max were both set as 1. Quantification of intact N-glycopeptides was performed by using Byologic software (5.4.10, Protein Metrics), which uses inputs from both MS1 raw data and Byonic search results to determine glycopeptide intensities. The Byologic quantification results were filtered to score >150. For quantitative analysis of N-glycan modifications, glycopeptides were grouped by their glycan composition or glycan type, and the relative abundance of glycan modification for each glycan category was determined by the summed intensities of glycopeptides for the glycan category normalized by total intensities of glycopeptides for all N-glycan modifications in each sample.

For phosphoproteomic data processing, the raw data were processed using Spectronaut (v19.0.240606.62635) via the direct DIA⁺ workflow with mostly default settings against different sequence database (*Mus musculus* or *Homo sapiens*) from UniProt. Phosphorylation (STY) was set as a variable modification with PTMs localization activated, and the site confidence score cutoff was set to 0.75. Stringent data filtering was applied with a precursor *q*-value, precursor PEP, protein *q*-value cutoff (experiment), protein *q*-value cutoff (run), and protein PEP cutoff all set to 0.01.

## Phosphosite/Glycan modification coregulation network analysis

A data-driven network analysis of phosphosite and glycan modifications was conducted using the WGCNA algorithm, facilitating the construction of a phosphosite/glycan network in the MCP samples from modification abundance profiles. We used WGCNA package in R to calculate a correlation matrix for all pairwise correlations of phoshossites/glycan modification abundances across all samples and then transformed it into a weighted adjacency matrix with a soft threshold power of 9 for phosphosites and 3 for glycan according to the scale-free topology criterion. The weighted adjacency matrix was used to generate a topological overlap (TO) matrix, and hierarchical clustering of phosphosites/glycans was performed using 1-TO as the distance measure. Modules of coregulated phosphosite/glycan modifications were identified using dynamic tree-cutting with the following parameters: minimal module size = 50/10, deepSplit = 2, merge cut height = 0.25, and a reassignment threshold of $P < 0.01$. Phosphosite or glycan coregulation module networks were graphically depicted by using the Cytoscape v.3.10.2 software. For each module in the phosphosite/glycan coregulation network, we calculated the module eigenphosphosite or eigenglycan as the first principal component of the coregulated phosphosites or glycans. Pearson correlations between each phosphosite/glycan and its module eigenphosphosite or eigenglycan determined module membership (kME), indicating intramodular connectivity. Module-trait associations with aging and sex were analyzed using biweight midcorrelations and corresponding $P$-values.

## Pathway temporal clustering and visualization

Proteome, phosphoproteome, and glycoproteome profiles across 2, 12, and 24 M were Z-score normalized and subjected to fuzzy c-means clustering using Mfuzz (R, v2.60.0) to capture temporal expression patterns. Cluster-specific KEGG enrichment was performed with clusterProfiler (v4.10.0), retaining pathways with FDR < 0.05, FC > 1.5. The resulting heatmap integrates three layers of information: (i) Membership plots (left), showing cluster-wise temporal trends with membership values; (ii) Expression heatmaps (middle), displaying Z-scored expression values across 2, 12, and 24 M; and (iii) KEGG pathways (right), annotated for each cluster, with functionally relevant pathways highlighted. Results were visualized as integrated membership trajectories, expression heatmaps, and enriched pathways using ggplot2 and ComplexHeatmap, ensuring full reproducibility in R (v4.3.1).

## PTM-SEA analysis

For PTM-SEA analysis, phosphosites were ranked based on signed effect sizes derived from moderated *t*-tests. Enrichment analyses were then performed against PTMsigDB, supplemented with in-house curated glycosite datasets, to retain directional regulatory information (upregulation vs. downregulation). Normalized enrichment scores (NES) and *P*-values were reported for the results. The entire workflow was implemented in R, utilizing limma for statistical calculations and ssGSEA2.0 for enrichment analyses.

## Statistical information

Significance of difference in group means between each group was assessed by unpaired two-tailed Student's *t* test, limma moderated *t* test, or Kruskal-Wallis test as indicated. For data visualization, heatmaps with *Z*-score values of normalized abundances were generated using the Heatmapper tool. Phosphosite or glycan coregulation network analysis was performed using the WGCNA algorithm to define network modules. Module-trait association was assessed by biweight midcorrelation between each module eigenphosphosite or eigenglycan and each trait with $P < 0.01$ as the confidence threshold. Enrichment analyses were performed with one-sided Fisher's exact test to calculate $P$ values. Correction for multiple comparisons was performed using the $q$ value or Benjamini-Hochberg FDR adjustment method as indicated.

## Ethics statement

All experiments involving cell lines, animals, and human samples complied with institutional and national guidelines. NB4 cells were obtained from ATCC and cultured under standard conditions. Animal studies were approved by the Ethics Committee of the Center for Excellence in Molecular Cell Science (protocol 2025-001), using mixed-sex C57BL/6J mice housed under specific pathogen-free conditions. Mouse tissues and biofluids were collected following approved

procedures and stored at −80 °C. Human plasma and CSF studies were approved by the Ethics Committee of the Gan Jiang Chinese Medicine Innovation Center (protocol GJCMIC2025-008). Written informed consent was obtained from all participants involved in this study. All samples were anonymized and stored at −80 °C until analysis.

## Reporting summary

Further information on research design is available in the Nature Portfolio Reporting Summary linked to this article.

## Data availability

Source data are provided with this paper. All data supporting the findings of this study are available in the Supplementary Information and from the figshare data repository [https://figshare.com/s/5f422ef846184ea9df83]. The mass spectrometry proteomics data generated in this study have been deposited to the ProteomeXchange Consortium via the iProX partner repository[74,75] with the dataset identifier PXD059701 and IPX0009561000. The N-glycosylation sites of MB, HS, and HCSF identified with MuPPE relative to those reported in UniProt, as summarized in the GlyGen dataset (Version 2.6.1, released on 08/07/2024), and [https://data.glygen.org/GLY_000039] for mouse N-glycosylation sites dataset and [https://data.glygen.org/GLY_000038] for human N-glycosylation sites dataset. Unless otherwise stated, all data supporting the results of this study can be found in the article, Supplementary and source data files. Source data are provided with this paper.

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

## Acknowledgements

The authors thank financial support by the National Natural Science Foundation of China (22274155, X.F.D.), Jiangxi Province Ganpo Talents Program (S2024CQKJ1797, X.L.L.), National Natural Science Foundation of Jiangxi Province (20232ACB203017 X.L.L.; 20224BAB206003 X.F.D.), and Dalian Institute of Chemical Physics Innovation Funding (DICP I202325, X.F.D.). We gratefully acknowledge Jia, Y.L. for her valuable contribution to sample collection. We would like to express our grati-tude to Servier Medical Art [https://smart.servier.com] for providing the medical image(s) used in this manuscript. The image(s) are licensed under the Creative Commons Attribution 4.0 International License (CC BY 4.0), with the official license terms available at https://creativecommons.org/licenses/by/4.0/.

## Author contributions

X.F.D. and F.F.X. contributed to data processing and drafting the manuscript. C.C. and G.Z.D. performed the primary experiments, with Y.C. and X.L.D. providing assistance for specific mouse studies. Y.F.Y. was responsible for the collection of aging mouse cohort samples. X.L.L., Y.D.S., and X.M.L. supervised the overall experimental design and critically revised the manuscript.

## Competing interests

The authors declare no competing interests.
