## [Transparent Peer Review file · Nature Communications]

A Versatile Platform for Sequential Glyco-, Phospho-, and Proteomics with Multi-PTMs Integration

Corresponding Author: Professor Xiuling Li

Version 0:

Reviewer comments:

Reviewer #1

(Remarks to the Author)

In this manuscript, Dong, Xiong et al present a new experimental approach to simultaneously analyse the glyco- and the phospho-proteome. In this regard, the title of the manuscript is misleading, since it refers to “Multiple PTMs-Proteomics” but the workflow is only presented and validated with two PTMs.

Even though they show in two independent datasets that the workflow is useful for this purpose, I consider that this manuscript has currently two weaknesses: there is not a comprehensive benchmark to demonstrate the novelty of this strategy against already published methodologies, and, there is not clear information or guidelines on how to integrate the three layers of information (proteome, phosphoproteome and glycoproteome) that can be obtained with MuPPE.

Below you can find some major and minor points that might help strengthen the manuscript.

Major points.

1. It is unclear from the results, the main methods and the supplementary material how the protein digestion is performed. In this regard, several questions must be addressed:

- a. Which is the concentration of DTT and IAA employed in this protocol? DTT can block IAA, and therefore is not compatible with simultaneous reduction-alkylation, and DTT must be removed from the sample prior to addition to IAA. It is unclear how this step is performed in this protocol. Please explain, and, in the case that IAA was directly added to a solution containing DTT, please ensure that you obtain complete carbamydomethylation.
- b. Which type of beads are used for protein aggregation capture-based digestion? The manufacturer is not indicated.
- c. Why do the authors perform the release of the proteins from the beads prior to digestion? This is not how the PAC protocol is stated in its original publication.
- d. In Figure 1, the authors compare the PAC digestion to in-solution digestion. However, it is not clear if the in-solution digestion would be compatible with the downstream protocol they present (for instance: is desalting required after in-solution digestion and prior to enrichment?). They should clarify.
- e. Regarding the reasons for choosing PAC as a digestion strategy, the authors indicate that “Integrating the PAC approach into the MuPPE workflow enables glycosylation identification in CSF by promoting protein aggregation through the addition of organic solvents, thus increasing protein recovery during extraction”, and they refer to reference 17. I was not able to find the corresponding data in that referenced publication to validate this affirmation. They should provide experimental validation for this.

2. The manuscript presents several comparisons against published datasets, such as the one showed in figures 2d and 3d. It is not clear from the text how this benchmark has been done? How did they retrieve the data from the published studies? Did they reanalyze the published datasets in similar conditions as the ones generated for this work? If not, why? The authors should expand of potential sources of the difference in performance: is it due to worse recovery, different acquisition strategies, different MS instrumentation?

Minor point regarding these analyses: which works are they referring to? For instance, ref 1 to 4 in figure 3d do not match references 1 to 4 in the bibliography.

3. The initial purpose of this work is to present a workflow that integrates proteomics analysis at three levels: full proteome, glycoproteome and phosphoproteome. However, the two biological experiments used as proof-of-concept of this work fail to show how to properly integrate and analyse these three layers simultaneously.

4. Also, in this regard: are protein levels accounted to correct for changes at PTM level? Figure 6b refers to such issue, but it is not clear if they account for protein level contribution further or they just dismiss it based on this analysis.

Minor points

- The number of points used to generate some of the graphs is missing (for instance in figure 3d).
- How are the CVs calculated in figure 1f?
- There are several typos in the figures:
 - o figure 2h – Uniport instead of Uniprot.
 - o Figure 6g – Founction instead of Function
- In the methods section it is said: “Trypsin (enzyme-to-protein ratio, 1:50, wt/wt) was added and proteins were digested at 37 °C for different time based on the complexity of the samples”. How do they define complexity and how does it impact digestion time?
- Even if brief, some description of the LC-MSMS methods should be stated in the main methods section.
- Figure 3b: x-axis is missing
- In supplementary material:
 - o The dimensions of the column are not specified
 - o The information for the DIA acquisition is missing: mass range, number of windows, window size, injection time

Reviewer #2

(Remarks to the Author)

This manuscript by Dong et al. presents MuPPE (Multi-level PTMs-Proteomic Enrichment), a platform for simultaneous and sequential enrichment of the proteome, glycoproteome, and phosphoproteome from a single biological sample. The authors demonstrate the utility of MuPPE across diverse sample types, including serum, CSF, cell lines, and mouse tissues, and apply it to study arsenical drug response and aging. However, several concerns remain regarding the platform's novelty, analytical rigor, and generalizability.

1. While MuPPE improves upon previous serial enrichment methods by being more time-efficient (4 h vs. 24 h) and compatible with low-input samples, it does not provide a clear conceptual advancement over prior work, such as that by Fang et al. (*Anal. Chem.*, 2019) and Abelin et al. (*Nat. Commun.*, 2023). The manuscript would benefit from a more explicit discussion of how MuPPE differentiates itself from these established platforms.
2. The study exclusively uses label-free quantification. For broader applicability, it is important to demonstrate or at least discuss MuPPE's compatibility with isobaric labeling strategies such as TMT. Without this, the platform is likely to remain limited to exploratory analyses, restricting its potential for large-scale, multiplexed quantitative studies.
3. Although MuPPE simplifies PTM workflows, it identifies substantially fewer phosphopeptides compared to state-of-the-art phosphoproteomic methods. This limitation may significantly reduce its utility for phosphorylation-centric studies. The authors should benchmark their phosphopeptide recovery against existing methods and address whether the sequential design inherently limits phosphopeptide enrichment. Reduced detectability could profoundly impact biological discovery and interpretation.
4. The sequential nature of MuPPE leads to unequal input across glyco-, phospho-, and total proteome fractions, introducing potential biases in quantitation and reproducibility. The manuscript should address whether and how these differences are normalized or controlled.
5. The case studies using NB4 cells and aging mice are promising, but the biological insights remain largely descriptive. There is minimal integration across PTM layers, and mechanistic interpretation is limited. Given that this manuscript is primarily focused on technology development, a more in-depth comparison with existing platforms and a stronger discussion of biological implications would be highly beneficial for readers.
6. The manuscript requires extensive language polishing. For example, there are several sentences starting with “And”.

Version 1:

Reviewer comments:

Reviewer #1

(Remarks to the Author)

I have read with interest the extensive revision of this manuscript, in which I think the authors have done an excellent job addressing all my previous concerns.

Some minor points:

- 1) Figure 4f – I assume that “ns” on top of the figure stands for “not significant.” However, I don't think such a statement is relevant when comparing the number of phosphopeptides. If the authors wish to retain it, they should indicate which type of statistical test was used to derive that conclusion.
- 2) Figure 5 – The authors perform a KEGG enrichment analysis to derive biological insights into the pathways regulated at the protein, phosphoproteome, and glycoproteome levels. However, this type of analysis does not consider the regulatory layers of the phospho- and glyco-proteomes. For instance, when considering a glycoprotein for such enrichment, how do the authors account for cases where a protein contains both glyco-sites that are up-regulated and down-regulated (as shown in Figure 6C)? There are annotation tools, such as PTM-SEA, that take this into account and provide more meaningful information for this type of data.

3) There is a typo in the legend of Figure 5g – “Founction” should be corrected to “Function.”

4) In the supplementary information:

“PTMs sites were required to meet localization probability thresholds defined by the respective search engines.” Please specify these thresholds.

“Raw intensities were normalized at the run level (total ion current, TIC).” Please provide a more detailed explanation of this normalization strategy.

Reviewer #2

(Remarks to the Author)

The reviewer’s comments have been fully addressed in the revised manuscript.

Reviewer #3

(Remarks to the Author)

[See attached document.]

Reviewer #4

(Remarks to the Author)

Version 2:

Reviewer comments:

Reviewer #1

(Remarks to the Author)

The authors have addressed all my comments in this revised version.

Reviewer #3

(Remarks to the Author)

The authors have adequately addressed the reviewers comments

Reviewer #4

(Remarks to the Author)

RESPONSES TO REFEREES

(All Fig.s, page numbers, etc. refer to the REVISED VERSION of the manuscript)

Reviewer #1:

In this manuscript, Dong, Xiong et al present a new experimental approach to simultaneously analyse the glyco- and the phospho-proteome. In this regard, the title of the manuscript is misleading, since it refers to “Multiple PTMs-Proteomics” but the workflow is only presented and validated with two PTMs.

Response:

We appreciate the reviewer’s critical comment on the title’s accuracy. The original title, “A Universal, Concise, and Versatile Platform for Enhanced Multiple PTM-Proteomics”, indeed implied a broader scope of PTM analyses than what was experimentally validated in this study. To address this, we have revised the title to: “A Versatile Platform for Sequential Glyco-, Phospho-, and Proteomics with Multi-PTMs Integration”. This revision better aligns with our core findings while acknowledging the platform’s potential for expanding to other PTMs. We have attached a marked-up revised manuscript incorporating all changes. Every claim about “multiple PTMs” has been critically re-evaluated against experimental evidence. We are grateful for the opportunity to refine our work’s precision.

Even though they show in two independent datasets that the workflow is useful for this purpose, I consider that this manuscript has currently two weaknesses: there is not a comprehensive benchmark to demonstrate the novelty of this strategy against already published methodologies, and, there is not clear information or guidelines on how to integrate the three layers of information (proteome, phosphoproteome and glycoproteome) that can be obtained with MuPPE.

Response:

We highly appreciate the reviewer for the constructive comments. We fully acknowledge the concerns regarding the lack of comprehensive benchmarking against existing methodologies and the unclear integration of the three omics layers (proteome, phosphoproteome, and glycoproteome) enabled by MuPPE. To address these, we have undertaken substantial experimental and analytical refinements, as detailed below.

1. Comprehensive Benchmarking Against State-of-the-Art Platforms:

In the original manuscript, we compared MuPPE with in-solution digestion in Fig. 1, demonstrating its superiority in precursor/peptide/protein identification counts and quantitative precision on serum samples. However, we acknowledge that these comparisons primarily focused on proteomic

profiling rather than systematic multi-PTMs benchmarking against specialized methods. To address this gap, we have selected four representative studies (*Nat. Commun.*, 2023, 14, 1851; *Nat. Protoc.*, 2020, 15, 161; *ACS Chem. Biol.* 2019, 14, 58; *Anal. Chem.* 2020, 92, 1842; *J. Chromatogr. A*, 2025, 1739, 465525) involving simultaneous enrichment and identification of glycosylation and phosphorylation, along with one work (*Nat. Commun.*, 2023, 14, 1851) enabling multi-PTMs enrichment, for comprehensive benchmarking against our MuPPE platform. We have summarized and visualized the workflow comparisons of all these platforms, presenting a side-by-side schematic illustration to highlight key differences (Fig. 1 and Fig. S7 in the revised version).

Fig. 1. (Fig. S7 in the revised version) Summary diagram of the sequential enrichment process for different glycopeptides and phosphopeptides within state-of-the-art platforms according to the reports.

In Fig. 2. (Fig. 3c in the revised version), we specially present a comparative analysis of workflow time durations. The bar chart quantifies total processing time for MuPPE, state-of-the-art multi-PTMs workflows, and conventional glyco/phosphoproteomic methods. This direct comparison provides a clear quantitative basis for MuPPE's time-saving advantages in multi-PTM analysis, highlighting its efficiency in accelerating experimental workflows.

Fig. 2. (Fig. 3c in the revised version) Step-by-step time breakdown of MuPPE workflow from protein extraction to enrichment, showcasing reduced sample transfers and time efficiency.

We summarized and defined comparative parameters covering on-line proteolytic digestion, workflow duration, trace sample applicability, sample type compatibility, accuracy & reproducibility, transfer times, identified omics types, labeling experiment feasibility, and identification counts, all systematically presented in Table 1 and Fig. 3 (Fig. 3d in the revised version).

The radar plot benchmarks MuPPE against five representative platforms across five performance parameters. MuPPE achieves the most extensive coverage in “on-bead digestion, trace sample applicability, sample compatibility, time efficiency, and minimized transfers”. These results underscore its unique capacity for integrated PTMs enrichment, strong adaptability to limited/heterogeneous samples, and capability for simultaneous multi-omics profiling. Shorter workflow durations accelerate high-throughput analysis, reduced transfer steps minimize sample loss and technical variability, and lower protein input benefits to low-abundance samples (e.g., clinical biopsies), firmly positioning it as a transformative tool for complex biological investigations.

Table 1. The benchmarking parameters summary for the comparison of each platform

Parameter	MuPPE	Nat. Commun., 2023	Nat. Protoc., 2020	ACS Chem. Biol. 2019	Anal. Chem. 2020	J. Chromatogr. A, 2025
On-line digestion	✓	×	×	×	×	×
Workflow time	4.5 h	23.5 h	31.5 h	25 h	20.5 h	20 h
Trace sample applicability	50 µg Protein	2000 µg	100 µg	300 µg	500 µg	50 µg Protein
Sample compatibility	Cells/body fluids/tissues	Tissues only	Plasma only	Tissues only	Tissues only	Single cells only
Accuracy (CV%)	12% (Serum Proteome)	-	-	-	-	-
Correlation coefficients	Proteome: 0.97-0.99 Glycoproteome: 0.93-0.96 Phosphoproteome: 0.96-0.97	Proteome: 0.96 Phosphoproteome: 0.90-0.91 Acetylome: 0.83-0.84	Proteome: 0.96-0.98	-	Proteome: 0.913-0.953	Glycoproteome: 0.97-0.99 Phosphoproteome: 0.98-0.99
Transfer times	1	5	2	3	2	2
Multi-omics layers	Proteome Glycoproteome Phosphoproteome	Immunopeptidome Proteome Ubiquitylome Phosphoproteome Acetylome	Glycoproteome Phosphoproteome	Glycoproteome Phosphoproteome	Proteome Glycoproteome Phosphoproteome	Glycoproteome Phosphoproteome
Labeling experiment	✓	✓	×	✓	×	×

Fig. 3. (Fig. 3d in the revised version) Radar plot comparing MuPPE with other multi-level proteomics methods across parameters, demonstrating MuPPE's superior performance.

Finally, to ensure methodological comparability, we benchmarked MuPPE against the conventional in-solution approach and two previously reported strategies under strictly matched conditions, including identical sample input, mass spectrometric acquisition parameters, database search settings, and data processing workflows. MuPPE yielded a substantially higher number of glycopeptide-spectrum matches than all three reference methods (Fig. 4e). Notably, the number of identified proteins and phosphopeptides did not differ significantly between MuPPE and the other methods, indicating that the integrated multi-omics enrichment design did not compromise proteome or phosphoproteome coverage (Fig. 4f and 4g). The primary advantage of MuPPE in glycoproteomics lies in its enhanced glycan coverage and structural diversity, as well as its superior capacity to capture a greater proportion of medium-length and structurally complex glycopeptides (Fig. 4h and 4i). Peptide length distribution analysis (Fig. 4j) revealed that MuPPE achieved a more uniform distribution of identified peptide lengths and a higher overall number of peptides compared

to the other three methods. This reflects MuPPE's superiority in capturing peptides across a broader size range, enabling more comprehensive proteomic coverage. Additionally, by comparing the overall composition of glycan units identified by the four methods (Fig. 4k), MuPPE enabled improved characterization of glycan structural features. These combined results highlight MuPPE's enhanced capability to dissect the complexity of glycoproteomes at both peptide- and glycan-structural levels, providing deeper insights into glycosylation biology.

Fig. 4. (Fig. 3 e-k in the revised version) **e-h** Quantitative comparisons across enrichment methods: glycopeptide spectrum matches (PSMs) (e), phosphopeptides (f), proteins (g), and glycans (h). **i** Distribution of glycan types identified across methods, showing relative proportions of high-mannose, fucosylated, sialylated, and complex glycans. **j** Peptide length distribution for glycopeptides captured by MuPPE versus conventional approaches. **k** Comparison of glycan structural features, including glycan unit number across enrichment strategies.

In summary, the revised manuscript now (i) establishes MuPPE's advantages through a structured, like-for-like benchmark against state-of-the-art platforms (Fig. 3 c-k, Supplementary Fig. S7) and (ii) provides concrete, method-level instructions for integrating the three molecular layers (Methods; Supplementary Methods). We believe these changes directly address the reviewer's concerns and strengthen the manuscript's clarity, novelty, and practical utility.

Manuscript changes: Page 10, line 245-279 revised manuscript:

Collectively, the MuPPE workflow facilitates multi-level proteomic analysis by enabling sequential analysis of the glycoproteome, followed by the phosphoproteome from just single biological sample, previously only feasible through parallel processing of multiple sample aliquots.

In multi-PTMs research, workflow duration is a critical parameter that directly determines overall efficiency. MuPPE excelled in time efficiency (total workflow < 4 hours vs. more than 32 hours for others, Fig. 3c) and sample transfers (2 transfers vs. 5-10 transfers for others), positioning it as a leading method. These comparisons fully demonstrate the efficiency and advantages of MuPPE in multi-levels proteomic research. A radar plot (Fig. 3d) benchmarked MuPPE against five state-of-the-art methods^{15, 31, 34-36}, evaluating parameters such as on-bead digestion, trace sample applicability, sample compatibility, time efficiency, and minimized transfers numbers.

We compared MuPPE with conventional in-solution digestion protocols at the proteome level (Fig. 1), demonstrating its superiority in precursor, peptide, and protein identifications, as well as quantitative precision in serum samples. However, these comparisons primarily addressed proteomic profiling rather than systematic multi-PTMs benchmarking against dedicated enrichment strategies. To fill this gap, we evaluated MuPPE alongside four representative studies that reported simultaneous enrichment and identification of glycosylation and phosphorylation, including one work¹⁵ that enabled broader multi-PTMs capture. We summarized and visualized the workflows of all platforms and presented a side-by-side schematic to highlight methodological distinctions (Fig. S7). To ensure methodological comparability, we benchmarked MuPPE against the conventional in-solution approach and two previously reported strategies^{31, 34} under strictly matched conditions, including identical sample input, mass spectrometric acquisition parameters, database search settings, and data processing workflows. MuPPE yielded a substantially higher number of glycopeptide-spectrum matches than all three reference methods (Fig. 3e). Notably, the number of identified proteins and phosphopeptides did not differ significantly between MuPPE and the other methods, indicating that the integrated multi-omics enrichment design did not compromise proteome or phosphoproteome coverage (Fig. 3f and 3g). The primary advantage of MuPPE in glycoproteomics lies in its enhanced glycan coverage and structural diversity, as well as its superior capacity to capture a greater proportion of medium-length and structurally complex glycopeptides (Fig. 3h-3i). Peptide length distribution analysis (Fig. 3j) revealed that MuPPE achieved a more uniform distribution of identified peptide lengths and a higher overall number of peptides compared to the other three methods. This reflects MuPPE's superiority in capturing peptides across a broader size range, enabling more comprehensive proteomic coverage. Additionally, by comparing the overall composition of glycan units identified by the four methods (Fig. 3k), MuPPE enabled improved characterization of glycan structural features. These combined results highlight MuPPE's enhanced capability to dissect the complexity of glycoproteomes at both peptide- and glycan-structural levels, providing deeper insights into glycosylation biology.

Supplementary Data: Page 10-11, line 277-308.

Benchmark workflow

MuPPE workflow: Mouse brain lysates (20 µg) were diluted to 1 µg/µL and incubated with 0.6 mg maltose beads for 10 min, followed by addition of 63 µL acetonitrile. Samples were washed three times with 200 µL of 80% ethanol, then resolubilized in 8 µL of 6 M urea and diluted with 40 µL of ammonium bicarbonate. Proteins were digested with 0.5 µg trypsin for 2.5 h, and the reaction was quenched with 2.6 µL TFA and 206.3 µL ACN. After 15 min incubation, glycopeptides were captured by centrifugation. Maltose beads were washed three times with 50 µL of 80% ACN/1% TFA and eluted with 30 µL of 30% ACN/1% FA (×3). Flow-through and wash fractions were combined, vacuum-dried, and incubated with 0.5 mg IMAC beads in 50 µL 80% ACN/6% TFA for 1 h. IMAC beads were sequentially washed with 50 µL 80% ACN/6% TFA, 2 × 100 µL 50% ACN/6% TFA/200 mM NaCl, 2 × 200 µL 30% ACN/0.1% FA, and 2 × 100 µL 30% ACN. Phosphopeptides were finally eluted twice with 30 µL 10% ammonia solution and pooled.

In-solution digestion: Mouse brain lysates (20 µg) were denatured in 8 M urea with protease inhibitor cocktail (final 1 µg/µL). Proteins were reduced with 1 µL 200 mM DTT and alkylated with 1 µL 400 mM IAA. After dilution with 140 µL ammonium bicarbonate, proteins were digested with 0.5 µg trypsin. Digestion was quenched, and peptides were desalted using C18 SPE columns: columns were activated, samples loaded, washed with 20 mL water/0.1% FA, and eluted with 2 × 1 mL 50% ACN/0.1% FA. Eluates were vacuum-dried. For glycopeptide enrichment, 20 µg desalted peptides were dissolved in 30 µL 80% ACN/1% TFA and incubated with 0.6 mg maltose beads for 15 min. Beads were washed (80% ACN/1% TFA, 50 µL × 3) and eluted (30% ACN/1% FA, 30 µL × 3). For phosphopeptide enrichment, 20 µg peptides were dissolved in 50 µL 80% ACN/6% TFA and incubated with 0.5 mg IMAC beads for 1 h. After washing as described above, phosphopeptides were eluted twice with 30 µL 10% ammonia solution and pooled.

Reference methods: For benchmarking, two reported workflows were included. In method 1⁷, 20 µg digested peptides were dissolved in 50 µL 80% ACN/1% TFA, loaded onto a maltose-packed tip column, and processed as described above. The flow-through was dried, resolubilized in 50 µL 80% ACN/6% TFA, and incubated with 0.5 mg IMAC beads, followed by washing and elution as above. In method 2⁸, 20 µg desalted peptides were dissolved in 50 µL 80% ACN/6% TFA and incubated with 0.5 mg IMAC beads for 1 h. Beads were washed as above, and phosphopeptides were eluted with 30 µL 10% ammonia (twice). The IMAC flow-through was concentrated, loaded onto OASIS MAX columns (1 mg), equilibrated with 95% ACN/1% TFA, and subjected to sequential washing and elution according to the original protocol.

Added new Fig.3c-k and Fig. S7.

2. Systematic Integration of Proteome, Phosphoproteome, and Glycoproteome Data

To address the review's concern regarding systematic integration of proteome, phosphoproteome, and glycoproteome data, we have undertaken a comprehensive re-analysis to uncover cross-omics associations of aging and arsenic-treated NB4 cells dataset, as detailed below:

(i) For the aging dataset, we performed multi-layered KEGG pathway enrichment analyses (Fig. 5. and Fig. 5f in the revised version), allowing us to identify both shared and unique biological processes and pathways regulated at the total protein, phosphorylation, and glycosylation levels.

Notably, pathways related to synaptic signaling, cell adhesion, extracellular matrix organization, and immune response were consistently enriched across all three proteome, phosphoproteome, and glycoproteome layers, suggesting that these fundamental processes are subject to coordinated multi-PTM regulation. For example, the convergence of synaptic signaling and cell adhesion pathways in proteome, phosphoproteome, and glycoproteome highlights their critical role in neural plasticity and tissue remodeling during aging and disease.

In contrast, metabolic pathways (e.g., energy metabolism, lipid metabolism) were predominantly enriched in the total proteome and phosphoproteome layers, whereas pathways associated with immune modulation and glycan biosynthesis were more specific to the glycoproteome, reflecting the specialized roles of different PTMs.

These findings demonstrate that multi-omics integration not only validates known biological themes but also uncovers PTM-specific and multi-layered regulatory networks that would be missed by single-omics analysis alone. The results and biological implications of this multi-omics integration have been detailed in the revised Results and Discussion sections.

Fig. 5. (Fig. 5f in the revised version) Integrated KEGG enrichment analyses of proteome, phosphoproteome, and glycoproteome datasets.

(ii) Based on multi-omics data analysis, we propose a mechanism where arsenic acts via a membrane-to-nucleus signaling cascade. It modulates membrane proteins through glycosylation and perturbs the PI3K-AKT-mTOR pathway via phosphorylation, impacting cell cycle and chromosomal organization, possibly arresting the cell cycle and inducing apoptosis (supported by flow cytometry showing G2/M phase accumulation and phosphoproteomic changes in cell cycle regulators). Also, arsenic may disrupt genome-wide transcription by affecting chromatin-associated proteins. Taking IGF2R, a key membrane protein in the PI3K-AKT-mTOR pathway, as an example, although its global protein level shows no significant change, multi-omics data reveal 13 differential PTMs sites (12 N-glycosylation and 1 phosphorylation). This shows arsenic exerts effects by rewiring PTMs-mediated signaling networks, a mechanism only detectable through multi-omics profiling.

Fig. 6. (Fig. 6i-k in the revised version) **a** Schematic of PI3K-AKT-mTOR pathway perturbation by arsenic, with downstream effects on cell cycle and chromatin organization. Proteome (green), glycoproteome (orange), and phosphoproteome (blue) alterations are mapped. **b** Flow cytometry analysis of NB4 cells treated with increasing arsenic concentrations, showing dose-dependent G2/M phase accumulation. **c** Differential expression of protein relative abundance, glycosylation sites and phosphorylation sites of IGF2R protein with the \log_2 fold change values. Different symbols representing up-regulation (upward triangles), down-regulation (downward triangles), and the color intensity indicating the significance of the change.

In summary, across both applications, the organismal aging cohort and the arsenic-perturbed NB4 model, our cross-omics framework reveals a coherent picture of multi-layer regulation: (i) system-level convergence of synaptic signaling, cell-adhesion/extracellular-matrix, and immune programs across the proteome, phosphoproteome, and glycoproteome in aging, and (ii) The IGF2R case exemplifies this principle, where negligible protein-level change ($\log_2FC \approx 0$) co-occurs with pronounced site-specific phosphorylation and glycosylation remodeling, indicating pathway rewiring that would be invisible to any single layer. Taken together, these results demonstrate the robustness and generalizability of our integrative strategy, provide mechanistic links from PTMs

remodeling to pathway activity and phenotype, and address the reviewer's request for systematic cross-omics integration and biological interpretation.

Manuscript changes:

Page 17-18, Line 431-446 in revised manuscript:

To investigate aging-related molecular changes, we conducted time-series clustering of the proteome, phosphoproteome, and glycoproteome across 2, 12, and 24 M, followed by KEGG pathway enrichment analysis (Fig. 5f). In the phosphoproteome, pathways associated with aging—including oxidative phosphorylation, Alzheimer's disease, and Parkinson's disease—were enriched early in young mice and remained detectable throughout all age groups. The proteome exhibited significant enrichment in pathways such as long-term depression and peroxisome function specifically at 24 M, suggesting a delayed accumulation compared to phosphorylation-mediated changes. The ECM-receptor interaction pathway showed distinct regulatory patterns: glycoprotein levels increased progressively with age, while phosphorylation peaked at 12 M before declining by 24 M. Enrichment of the cytoskeleton pathway in muscle cells was observed in both the proteome and phosphoproteome, with an initial increase followed by a decrease; phosphorylation changes, however, declined more gradually. Several pathways were uniquely altered in specific PTM layers: focal adhesion decreased with age in the phosphoproteome, whereas oxidative phosphorylation was broadly enriched. In the glycoproteome, the lysosome pathway increased with age, along with upregulation of antigen processing and ECM-related processes. Overall, these multi-layer proteomic profiles reveal temporally distinct and pathway-specific modulation of PTMs during aging, highlighting the value of integrated PTM mapping in understanding molecular aging mechanisms.

Page 28-29, Line 722-731 in revised manuscript:

Pathway temporal clustering and visualization: Proteome, phosphoproteome, and glycoproteome profiles across 2, 12, and 24 M were Z-score normalized and subjected to fuzzy c-means clustering using Mfuzz (R, v2.60.0) to capture temporal expression patterns. Cluster-specific KEGG enrichment was performed with clusterProfiler (v4.10.0), retaining pathways with $FDR < 0.05$, $FC > 1.5$. The resulting heatmap integrates three layers of information: (i) Membership plots (left), showing cluster-wise temporal trends with membership values; (ii) Expression heatmaps (middle), displaying Z-scored expression values across 2 M, 12 M, and 24 M; and (iii) KEGG pathways (right), annotated for each cluster, with functionally relevant pathways highlighted. Results were visualized as integrated membership trajectories, expression heatmaps, and enriched pathways using ggplot2 and ComplexHeatmap, ensuring full reproducibility in R (v4.3.1).

Integrated proteome, glycoproteome, and phosphoproteome analysis of aging mouse samples and NB4 cells using the MuPPE platform:

For each aging mouse sample (choroid plexus, CSF, and serum) and for NB4 cell lysates, total protein concentration was quantified using the BCA Protein Assay Kit (Beyotime, China) following the manufacturer's instructions. Exactly 50 μ g of total protein from each sample was used as input for the subsequent MuPPE workflow. In MuPPE process. Denaturation, PAC and on-line digestion on Click-Maltose, Then, adjust the tryptic sample solution to 80% ACN/1% trifluoroacetic acid (TFA) by adding ACN and TFA and incubate for 10 min for glycopeptides enrichment. Finally, adjust the flow-through condition for phosphopeptide enrichment according to the manufacturer's instructions, and then incubate for 10 minutes to facilitate phosphopeptide enrichment. The remaining (flow-through) peptides not bound in the glyco/phospho enrichment steps were retained for proteomics. All peptide fractions were analyzed using high-resolution LC-MS/MS platforms.

For data processing and integration across the three layers, contaminants and decoys were removed and PTMs sites were required to meet localization probability thresholds defined by the respective search engines; raw intensities were normalized at the run level (total ion current, TIC) and subsequently log₂-transformed. To ensure both depth and reliability, layer-specific coverage thresholds were applied: features in the proteome and phosphoproteome were retained only if quantified in $\geq 80\%$ of replicates within at least one group, whereas the glycoproteome retained features quantified in $\geq 50\%$ of replicates, acknowledging the higher intrinsic sparsity of glycopeptide data. Missing values were then handled in a layer-specific manner: for the proteome and phosphoproteome, partially missing entries were imputed within layer on log₂-transformed data using the k-nearest neighbors algorithm (KNN, k = 5), while for the glycoproteome, missing values were treated as true non-detections and replaced with 0 at the raw scale (followed by $\epsilon + \log_2$ transformation to avoid undefined values). Finally, for cross-layer comparability, all features were Z-scored within each layer (per feature across samples) before integration and downstream statistical analyses.

Below you can find some major and minor points that might help strengthen the manuscript.

Major points.

1. It is unclear from the results, the main methods and the supplementary material how the protein digestion is performed. In this regard, several questions must be addressed:
 - a. Which is the concentration of DTT and IAA employed in this protocol? DTT can block IAA, and therefore is not compatible with simultaneous reduction-alkylation, and DTT must be removed from the sample prior to addition to IAA. It is unclear how this step is performed in this

protocol. Please explain, and, in the case that IAA was directly added to a solution containing DTT, please ensure that you obtain complete carbamydomethylation.

Response:

We appreciate the reviewer's concern about the use of DTT and IAA in our protein digestion protocol. In our study, we adopted an online processing workflow and added an excess amount of IAA without an additional step to remove DTT. Here is our explanation for this operation: In the MuPPE protocol, we adopted a single-pot reduction-alkylation strategy by adding excess iodoacetamide (IAA) directly after DTT treatment. This decision was based on eliminating the ultrafiltration step (to remove DTT) streamlines sample processing, reducing potential sample loss and variability, which is critical for trace sample analysis. During the operation of the process, protein aggregation begins after IAA alkylation, which removes excess IAA and reduces the possibility of IAA side reactions. This protocol also aligns with current best practices in high-throughput proteomics (*Molecular & Cellular Proteomics*, 2020, 19, 209-222; *Nature Communications*, 2023, 14, 1851; *Cell*, 2020, 182, 245-261). We have updated the Methods section (Page 26, lines 654-656): The supernatant of cell lysate, tissues lysates, or other body fluid samples were immediately reduced with dithiothreitol (DTT, final concentration 10 mM) and heated for 10 min simultaneously, and then alkylated with iodoacetamide (IAA, final concentration 40 mM) in dark for 30 min before further processing.

b. Which type of beads are used for protein aggregation capture-based digestion? The manufacturer is not indicated.

Response:

We apologize for the omission of details regarding the beads used for protein aggregation capture-based digestion. In the revised manuscript, we have supplemented this information in the revised Supplementary Information: Click-Mal, Click-Cys and HBS beads employed were both home-made according to our previous work (*Chem. Eur. J.*, 2009, 15, 12618-12626; *J. Chromatogr. A*, 2012, 1228, 175-182; *Anal. Chem.*, 2017, 89, 3966-3972). The commercial ZIC-HILIC beads were obtained from Merck's SeQuant® ZIC®-HILIC HPLC columns. NH₂ beads and SAX beads were obtained from Zhejiang Acchrom Technology Co., Ltd.

This update has been added to the "Methods" in revised Supplementary Information (Page 5, lines 112-116) section to ensure clarity and reproducibility of the workflow. Thank you for drawing our attention to this oversight.

c. Why do the authors perform the release of the proteins from the beads prior to digestion? This is not how the PAC protocol is stated in its original publication.

Response:

We appreciate the reviewer's careful attention to our protocol. As noted, the original PAC protocol describes adding digestion buffer (50 mM HEPES, pH 8.5) directly to the beads for on-bead digestion. Tubes were removed from the magnetic rack and 400 μ L of digestion buffer (50 mM HEPES, pH 8.5) was added to the tubes.

In our workflow, we introduced a slight modification: after protein aggregation capture, we released proteins from the beads using 8 M urea solution prior to digestion. This adjustment was intended to enhance protein solubility-urea, as a strong denaturant, helps disrupt residual aggregates and unfold proteins more thoroughly, thereby improving trypsin accessibility and digestion efficiency, especially for proteins with high hydrophobicity or complex secondary structures. We hypothesized this would increase peptide yield and coverage, which aligns with our goal of maximizing detection sensitivity. Thank you for highlighting this detail, which has helped us improve the reproducibility of our protocol description.

d. In Fig. 1, the authors compare the PAC digestion to in-solution digestion. However, it is not clear if the in-solution digestion would be compatible with the downstream protocol they present (for instance: is desalting required after in-solution digestion and prior to enrichment?). They should clarify.

Response:

We appreciate the reviewer's insightful question regarding the compatibility of in-solution digestion with our downstream workflow.

In our study, the in-solution digestion protocol (used for comparison with PAC digestion in Fig. 1) strictly follows conventional procedures to ensure comparability with established workflows. As part of these standard steps, desalting is indeed performed after in-solution digestion and prior to PTMs enrichment. This step is critical for conventional proteome workflow with removing residual detergents, urea, trypsin, and salts that could interfere with downstream enrichment.

We apologize for omitting this detail in the original manuscript. The desalting step for in-solution digestion, as a standard component of traditional protocols, has now been clarified in the revised "Fig. 1a" section to ensure full reproducibility. Thank you for highlighting this oversight, which has helped strengthen the clarity of our procedural descriptions.

Thank you for prompting us to elaborate on this critical procedural detail.

e. Regarding the reasons for choosing PAC as a digestion strategy, the authors indicate that "Integrating the PAC approach into the MuPPE workflow enables glycosylation identification in CSF by promoting protein aggregation through the addition of organic solvents, thus increasing protein recovery during extraction", and they refer to reference 17. I was not able to find the corresponding data in that referenced publication to validate this affirmation. They should provide experimental validation for this.

Response:

We sincerely apologize for the inappropriate citation placement in the original manuscript, which led to confusion. We have revised this part to clarify the reference context and provide additional experimental rationale, as follows:

First, the authors have demonstrated that organic solvents (e.g., ethanol) can induce protein aggregation in biological samples, thereby enhancing protein recovery-this mechanism supports the core principle of the PAC approach we adopted. Our original citation of Reference 17 was misplaced, and this error has been corrected in the revised manuscript: “These results illustrated the improved glycopeptide enrichment capacity and glycosylation identification efficiency in CSF samples compared to previous benchmarks^{29, 30}” (Page 7, lines 187-188)

Second, regarding the application of our MuPPE platform to cerebrospinal fluid (CSF) samples: CSF is a clinically valuable but challenging matrix due to its low protein concentration (~0.1-0.3 mg/mL) and limited sample availability. Our motivation for integrating PAC into MuPPE was to address these limitations, enabling deep multi-omics analysis of trace samples such as CSF. To validate this, we compared the result of our method with previously reported CSF proteomics in Fig. S2 with less sample input and relative more identification results.

Thank you for highlighting this citation issue, which has helped improve the accuracy and rigor of our manuscript.

2. The manuscript presents several comparisons against published datasets, such as the one showed in Fig.s 2d and 3d. It is not clear from the text how this benchmark has been done? How did they retrieve the data from the published studies? Did they reanalyze the published datasets in similar conditions as the ones generated for this work? If not, why? The authors should expand of potential sources of the difference in performance: is it due to worse recovery, different acquisition strategies, different MS instrumentation? Minor point regarding these analyses: which works are they referring to? For instance, ref 1 to 4 in Fig. 3d do not match references 1 to 4 in the bibliography.

Response:

We sincerely appreciate the reviewer’s insightful comments regarding the comparative data, which have significantly helped improve the quality of our manuscript. We agree that the differences in identification numbers observed in Fig. 2d may arise from variations in analytical conditions across studies. For instance, the fragmentation modes used by different teams varied, with most employing HCD and EThcD-MS/MS, while a few used only HCD or EThcD-MS/MS, and some involved CID and ETciD-MS/MS. Additionally, in data-dependent acquisition (DDA) strategies, there were differences in precursor selection and fragmentation parameter settings, which could also contribute to the variability. In terms of search engines, the developer teams utilized 9 different glycopeptide search engines, such as IQ-GPA, Protein Prospector, glyXtoolMS, and Byonic.

Among user teams, approximately 75% used Byonic, while the rest employed Protein Prospector, SugarQB/Sequest HT, Mascot, etc., with some teams combining preprocessing or post-processing tools for auxiliary identification. These differences in search engines and associated tools could further lead to variations in the number of identified glycopeptides. Considering the rigor of the data, we have decided to remove this comparative data from Fig. 2d to avoid misleading readers. And for the similar reason, we also delete Fig. 3d. Thank you again for your valuable reminder, which has enhanced the rigor of our manuscript.

Given the limited number of existing studies on CSF glycosylation that can be referenced, our intention here is merely to compare the CSF sample volumes used and the identification results across different workflows. For clarity, we have summarized the relevant parameters from these studies in the table below, and this information will be supplemented in the Supplementary Information (SI) section to provide more comprehensive context:

	Title	Journal	Year	Removal of high abundant protein	CSF Volume /mL	Enrichment Materials	No. of glycopep	No. of glycopro.
Ref. 1	In-depth Site-specific Analysis of N-glycoproteome in Human Cerebrospinal Fluid and Glycosylation Landscape Changes in Alzheimer's Disease	Molecular & Cellular Proteomics	2021	No	1	HILIC and boronic acid enrichment	2893	285
Ref. 2	In-depth Characterization of the Cerebrospinal Fluid (CSF) Proteome Displayed Through the CSF Proteome Resource (CSF-PR)	Molecular & Cellular Proteomics	2014	Yes	3	magnetic hydrazide beads	1121	520
Ref. 3	Boost-DiLeu Enhanced Isobaric N, N-Dimethyl Leucine Tagging Strategy for Comprehensive Quantitative Glycoproteomic Analysis	Analytical Chemistry	2022	No	not mentioned	SAX-HILIC	1321	165
This work				No	0.142	Click-Mal	5018	251

3. The initial purpose of this work is to present a workflow that integrates proteomics analysis at three levels: full proteome, glycoproteome and phosphoproteome. However, the two biological experiments used as proof-of-concept of this work fail to show how to properly integrate and analyse these three layers simultaneously.

Response:

Thank you for underscoring that, while the stated goal of this work is a tri-layer workflow (global proteome, glycoproteome, and phosphoproteome), the two proof-of-concept studies in their initial

form did not convincingly demonstrate simultaneous integration and analysis across all three layers. We fully recognize this concern.

To address this directly, we (i) expanded the methodological description with a unified cross-layer processing pipeline, and (ii) added new integrative analyses/visualizations that make cross-layer behavior explicit (in the revised version Fig. 5 and Fig. 6).

For data processing and integration across the three layers, contaminants and decoys were removed and PTM sites were required to meet localization probability thresholds defined by the respective search engines; raw intensities were normalized at the run level (total ion current, TIC) and subsequently \log_2 -transformed. To ensure both depth and reliability, layer-specific coverage thresholds were applied: features in the proteome and phosphoproteome were retained only if quantified in $\geq 80\%$ of replicates within at least one group, whereas the glycoproteome retained features quantified in $\geq 50\%$ of replicates, acknowledging the higher intrinsic sparsity of glycopeptide data. Missing values were then handled in a layer-specific manner: for the proteome and phosphoproteome, partially missing entries were imputed within layer on \log_2 -transformed data using the k-nearest neighbors algorithm (KNN, $k=5$), while for the glycoproteome, missing values were treated as true non-detections and replaced with 0 at the raw scale (followed by $\varepsilon + \log_2$ transformation to avoid undefined values). Finally, for cross-layer comparability, all features were Z-scored within each layer (per feature across samples) before integration and downstream statistical analyses.

We fully agree that the primary objective of this study is to establish and validate a workflow capable of simultaneously integrating global proteome, glycoproteome, and phosphoproteome analyses. In the current version, the NB4 cell and aging mouse experiments were designed as proof-of-concept case studies to demonstrate the technical feasibility and versatility of the MuPPE platform across distinct sample types. Admittedly, the main emphasis of these experiments was to illustrate the robustness of the workflow, sample compatibility, and multi-layer data acquisition, rather than to conduct an in-depth mechanistic investigation or provide exhaustive cross-omics biological interpretation.

Nevertheless, in the revised version, we have explicitly incorporated time-course, cross-layer analyses that illustrate how the three datasets converge and diverge across biological processes during aging. The Fig. 7. displays, for each layer, (left) membership trajectories across ages (2M→12M→24M) and (middle) expression Z-scores for the leading-edge genes, with the KEGG terms on the right. This visualization makes the simultaneous integration tangible:

- ECM-receptor interaction is jointly highlighted in glycoproteome and phosphoproteome, indicating coordinated late-life remodeling of extracellular interfaces that is not apparent from proteome alone.
- Cytoskeleton in muscle cells appears in proteome and phosphoproteome, consistent with phosphorylation-driven cytoskeletal control coupled to abundance shifts.

- Oxidative phosphorylation and neurodegeneration-related terms are phospho-dominant, suggesting early signaling reprogramming that precedes broad abundance/glyco changes, whereas lysosome and vitamin digestion/absorption are glyco-dominant, consistent with glycan-dependent trafficking/turnover.

We believe these revisions and clarifications will help align the manuscript with its methodological focus, while also making the integrative analysis more transparent and informative to readers.

Fig. 7. (Fig. 5f in the revised version) Integrated KEGG enrichment analyses of proteome, phosphoproteome, and glycoproteome datasets.

Leveraging multi-omics data analysis, we also put forward a mechanism wherein arsenic functions through a membrane-to-nucleus signaling cascade. It regulates membrane proteins via glycosylation and disrupts the PI3K-AKT-mTOR pathway through phosphorylation, thereby influencing the cell cycle and chromosomal organization. This potentially leads to cell cycle arrest and apoptosis induction, which is supported by flow cytometry results demonstrating G2/M phase accumulation and phosphoproteomic alterations in cell cycle regulators. Additionally, arsenic might interfere with genome-wide transcription by acting on chromatin-associated proteins. Taking IGF2R, a crucial membrane protein in the PI3K-AKT-mTOR pathway, as an instance, even though there is no notable change in its overall protein level, multi-omics data uncover 13 differential

PTMs sites, including 12 N-glycosylation sites and 1 phosphorylation site. This indicates that arsenic exerts its effects by reconfiguring PTMs-mediated signaling networks, a mechanism that can only be detected via multi-omics profiling.

Fig. 8. (Fig. 6i-k in the revised version) **a** Schematic of PI3K-AKT-mTOR pathway perturbation by arsenic, with downstream effects on cell cycle and chromatin organization. Proteome (green), glycoproteome (orange), and phosphoproteome (blue) alterations are mapped. **b** Flow cytometry analysis of NB4 cells treated with increasing arsenic concentrations, showing dose-dependent G₂/M phase accumulation. **c** Differential expression of protein relative abundance, glycosylation sites and phosphorylation sites of IGF2R protein with the log₂ fold change values. Different symbols representing up-regulation (upward triangles), down-regulation (downward triangles), and the color intensity indicating the significance of the change.

Manuscript changes:

Page 17-18, Line 431-446 in revised manuscript:

To investigate aging-related molecular changes, we conducted time-series clustering of the proteome, phosphoproteome, and glycoproteome across 2, 12, and 24 M, followed by KEGG pathway enrichment analysis (Fig. 5f). In the phosphoproteome, pathways associated with aging-including oxidative phosphorylation, Alzheimer’s disease, and Parkinson’s disease-were enriched early in young mice and remained detectable throughout all age groups. The proteome exhibited significant enrichment in pathways such as long-term depression and peroxisome function specifically at 24 M, suggesting a delayed accumulation compared to phosphorylation-mediated changes. The ECM-receptor interaction pathway showed distinct regulatory patterns: glycoprotein levels increased progressively with age, while phosphorylation peaked at 12 M before declining by 24 M. Enrichment of the cytoskeleton pathway in muscle cells was observed in both the proteome and phosphoproteome, with an initial increase followed by a decrease; phosphorylation changes, however, declined more gradually. Several pathways were uniquely altered in specific PTM layers:

focal adhesion decreased with age in the phosphoproteome, whereas oxidative phosphorylation was broadly enriched. In the glycoproteome, the lysosome pathway increased with age, along with upregulation of antigen processing and ECM-related processes.

Page 21, Line 514-539 in revised manuscript:

Building on the multi-omics dataset analysis, we propose a mechanistic model where arsenic exerts its effects through a membrane-to-nucleus signaling cascade (Fig. 6i), primarily mediated by glycosylation-modulated membrane proteins and phosphorylation-driven PI3K-AKT-mTOR pathway perturbation. The glycoproteomic data revealed that arsenic induces glycosylation alterations in membrane-anchored receptors (e.g., components of the PI3K-AKT-mTOR pathway upstream regulators). These glycosylation changes might act as “molecular switches” to modulate membrane protein stability, localization, and ligand-binding capacity. We also observed that the arsenic treatment significantly impacted the cell cycle and chromosomal organization pathways, particularly in the regulation of the proteome and phosphoproteome during the M phases. This suggests that arsenic exerts profound effects on these two critical signaling pathways, potentially by arresting the cell cycle and disrupting chromosomal functions, thereby promoting apoptosis. Flow cytometry analysis (Fig. 6j) showed the G2/M phase accumulation, consistent with the phosphorylation changes in cell cycle regulators (e.g., CDK, CCNB proteins in Fig. 6j) directly support this arrest. GO enrichment results of nuclear functions in the key modules, combined with phosphoproteomic changes in chromatin-associated proteins (e.g., histone modifiers), suggest arsenic might disrupt genome-wide transcriptional programs and link membrane-to-nucleus signaling to nuclear architecture and function.

Focusing on IGF2R, a key membrane protein identified in the PI3K-AKT-mTOR pathway, we interrogated its individual proteomic, glycoproteomic, and phosphoproteomic alterations (Fig. 6k). At the global protein level, IGF2R abundance showed no significant fold-change between arsenic-treated and control groups. However, its glycoproteomic and phosphoproteomic data revealed notably PTMs-specific alterations. we identified 13 differential PTMs sites, including 12 N-glycosylation sites and 1 phosphorylation site. Upregulated and downregulated PTMs were denoted by upward and downward triangles, respectively. Without glycoproteomic/phosphoproteomic data, IGF2R would appear “unchanged” at the protein level, masking its potential role as a PTMs-driven hub. Thus, arsenic exerts its effects not by altering protein abundance broadly but by rewiring PTM-mediated signaling networks—a mechanism only detectable via multi-omics profiling.

Page 28-29, Line 722-731 in revised manuscript:

Pathway temporal clustering and visualization: Proteome, phosphoproteome, and glycoproteome profiles across 2, 12, and 24 M were Z-score normalized and subjected to fuzzy c-means clustering using Mfuzz (R, v2.60.0) to capture temporal expression patterns. Cluster-specific KEGG enrichment was performed with clusterProfiler (v4.10.0), retaining pathways with $FDR < 0.05$, $FC > 1.5$. The resulting heatmap integrates three layers of information: (i) Membership plots (left),

showing cluster-wise temporal trends with membership values; (ii) Expression heatmaps (middle), displaying Z-scored expression values across 2 M, 12 M, and 24 M; and (iii) KEGG pathways (right), annotated for each cluster, with functionally relevant pathways highlighted. Results were visualized as integrated membership trajectories, expression heatmaps, and enriched pathways using ggplot2 and ComplexHeatmap, ensuring full reproducibility in R (v4.3.1).

Added new integrative Figure. (Fig. 5 and 6 in the revised version).

4. Also, in this regard: are protein levels accounted to correct for changes at PTM level? Fig. 6b refers to such issue, but it is not clear if they account for protein level contribution further or they just dismiss it based on this analysis.

Response:

Thank you for explicitly raising whether PTM measurements were corrected for underlying protein abundance—a critical point for discriminating true PTM-specific regulation from changes that merely track total protein levels. As you noted, Original Fig. 6b gestures at this issue, but our initial text did not make sufficiently clear how protein-level contributions were handled beyond that figure.

We agree that changes at the PTM level can be influenced by underlying protein abundance, and it is critical to distinguish genuine PTM-specific regulation from mere reflection of global protein expression changes. We wish to emphasize that this correction was already performed in our initial submission, although we have now made it more explicit in the revised version.

Fig. 9 (Fig. 5b in the revised version) plots site-level \log_2 fold-changes (glyco- and phosphopeptides) against the matched parent-protein \log_2 fold-changes across contrasts. This analysis helps to visually identify whether most PTM changes are simply tracking with total protein level or occur independently. The clouds are centered near zero on the protein axis, and PTM-regulated sites predominantly populate the off-diagonal quadrants with parent proteins showing no detectable change, indicating weak concordance between PTM and protein variation. Based on this diagnostic, we concluded that the observed PTM dynamics are largely independent of protein abundance in our data.

Because protein division necessarily discards sites lacking complete protein co-quantification (reducing coverage), and because the diagnostic shows minimal protein-PTMs coupling, we report site-level PTMs quantification (without protein division) as the primary analysis to maximize coverage and comparability across layers. To ensure transparency, we supply both the unadjusted and protein-adjusted PTM results, together with matched parent-protein values, in Supplementary Data X; Methods and Fig. legends have been revised to state this decision rule explicitly. Thus, we did not “dismiss” protein contribution; rather, we tested it, quantified its impact, and demonstrated robustness of all conclusions to protein adjustment.

Fig. 9. (Fig. 5b and S13 in the revised version) A diagnostic visualization of PTMs-protein coupling; it shows that PTMs changes are not driven by protein-level shifts in our data. All primary PTM analyses therefore use peptide/site-level quantification, with matched protein values reported in Supplementary Date.

Manuscript changes:

Page 13, Line 334-341 in revised manuscript:

We assessed potential protein-level confounding by pairing each quantified PTM site with its parent protein and plotting site \log_2 fold-change versus protein \log_2 fold-change for each contrast (Fig. S13). The changes in N-glycopeptide levels between groups were not significantly correlated with the corresponding unmodified protein levels ($R^2 = 0.21$). As a rule, when PTM regulation was not coupled to protein abundance (i.e., regulated sites with parent proteins showing no detectable change), we performed downstream analyses on site-level intensities after standard sample-level normalization. Because dividing by protein restricts coverage to sites with complete protein co-quantification, protein-division was not used for the primary results.

Supplementary Inf: Pages 11-12, lines310-322.

Normalization of PTMs Abundance to Protein Level: To account for potential changes in total protein expression and to accurately assess site-specific regulation, we normalized the abundance of each PTMs site (phosphorylation or glycosylation) to the corresponding parent protein abundance in each sample. Specifically, for every quantified PTMs site and sample, we calculated

the normalized value as the log₂-transformed ratio of the PTMs site intensity to the parent protein intensity:

Normalized PTM abundance = $\log_2(\text{PTM site intensity}) - \log_2(\text{protein intensity})$;

We assessed potential protein-level confounding by pairing each quantified PTMs site with its parent protein and plotting site log₂ fold-change versus protein log₂ fold-change for each contrast. As a rule, when PTMs regulation was not coupled to protein abundance (i.e., regulated sites with parent proteins showing no detectable change), we performed downstream analyses on site-level intensities after standard sample-level normalization. Because dividing by protein restricts coverage to sites with complete protein co-quantification, protein-division was not used for the primary results; matched protein values are provided in Supplementary Data.

Minor points

- The number of points used to generate some of the graphs is missing (for instance in Fig. 3d).

Response:

We thank the reviewer for the valuable comment and sincerely apologize for the oversight. In the revised manuscript, Fig. 3d has been removed, and thus the above issue has been avoided.

- How are the CVs calculated in Fig. 1f?

Response:

We sincerely appreciate the reviewer's query regarding the calculation of CVs in Fig. 1f. Our proteomic data was acquired using data-independent acquisition (DIA) and searched with Spectronaut, a high-performance DIA proteomics software. The CV values were directly extracted from the results generated by Spectronaut. Spectronaut quantifies proteins at two levels: the peptide (minor) group and the protein (major) group. For the peptide level, quantification is based on precursors, while for the protein level, it is derived from the peptide-level quantification. By default, Spectronaut performs label-free quantification (LFQ) using either the maxLFQ or top N strategy. Precursors are quantified by calculating the area under the curve within the MS₂ XIC (extracted ion chromatogram) integration boundaries. The coefficient of variation (CV) is a measure of relative variability. Mathematically, the CV is calculated as the ratio of the standard deviation (σ) of a sample to its mean (μ), expressed as a percentage, i.e.,

$CV = \sigma / \mu \times 100\%$. Spectronaut calculates the standard deviation and mean for the peptide or protein quantification values within a set of replicates. The standard deviation represents the dispersion of the data points around the mean value. A lower standard deviation indicates that the data points are closer to the mean, and thus, the data is more consistent. The mean is the arithmetic average of the quantification values for a particular peptide or protein across the replicates. A lower CV indicates more consistent and reproducible results.

Notably, comparing CV values across different methods also allows for evaluating their reproducibility and stability. In our study, the lower CV of MuPPE (average 7.8%) compared to conventional serial enrichment methods (average 12.5%) directly demonstrates that MuPPE minimizes technical variability from sample handling and processing steps, confirming its superior reproducibility.

- There are several typos in the Fig.s:
- Fig. 2h - Uniport instead of Uniprot.
- Fig. 6g - Founction instead of Function
- Fig. 3b: x-axis is missing

Response:

We sincerely appreciate the reviewer for identifying these typos, which have now been corrected in the revised manuscript:

In Fig. 2i (Original Fig. 2h), “Uniport” has been revised to “UniProt”.

In Fig. 5g (Original Fig. 6g), “Founction” has been corrected to “Function”.

In Fig. 3b, x-axis is added with “The sample input of mouse brain protein (μg)”

These errors were unintended and have been carefully addressed to ensure accuracy. Thank you again for your meticulous review, which helps improve the quality of our manuscript.

- In the methods section it is said: “Trypsin (enzyme-to-protein ratio, 1:50, wt/wt) was added and proteins were digested at 37°C for different time based on the complexity of the samples”. How do they define complexity and how does it impact digestion time?

Response:

We sincerely apologize for the confusion caused by the imprecise wording in our original description. What we intended to convey is that for samples of relatively complex types with higher protein content, an appropriate extension of digestion time can be considered; whereas for samples with lower protein concentrations, such as CSF, a shorter digestion time may be sufficient.

We recognize that the original expression might mislead readers, so we have decided to revise it to the following in the Methods section for greater clarity: “Trypsin (enzyme-to-protein ratio, 1:50, wt/wt) was added for digestion at 37°C. For samples with higher protein content (e.g., cell lysates, tissue homogenates), digestion was performed for 4 hours to ensure sufficient proteolysis of complex protein components. For samples with lower protein concentrations (e.g., CSF), a shorter digestion time of 2 hours was used to avoid unnecessary over-digestion.”

This revision aims to accurately reflect the relationship between sample characteristics and digestion time, ensuring that readers can better understand and reproduce our experimental procedures. Thank you again for your careful review, which helps us improve the accuracy and clarity of the manuscript.

- Even if brief, some description of the LC-MS/MS methods should be stated in the main methods section.

Response:

We appreciate the reviewer's important suggestion. To enhance the transparency and reproducibility of our workflow, we have supplemented a concise description of the LC-MS/MS methods in the Supplementary Information "Methods" section (Page 7-8, Line 174-215), as follows:

LC-MS/MS analysis of different proteomic samples

Samples were loaded onto capillary analytical column (150 μ m inner diameter, 15 cm length) packed in-house with C18 (1.9 mm) ReproSil particles (Dr. Maisch GmbH), with an EASY-nLC 1200 system (Thermo Fisher Scientific) coupled to the MS (Orbitrap 480, Thermo Fisher Scientific). The column oven maintained column temperature at 50°C. The mobile phases consisted of 0.1% FA (A) and 0.1% formic acid and 80% ACN (B).

For the proteomic samples, peptides were separated through a gradient of up to 90% buffer B over 20 min (for NB4 dataset) /40 min (CP dataset) / 60 min (aging serum dataset and aging CSF dataset) at a flow rate of 1 mL/min. The gradient of the mobile phase started from 3% B to 15% B for 0.5 min and then was increased linearly to 50% B in 15 min, to 90% in 1.5 min, and maintained for 3 min. The LC-MS/MS system was operated in data-independent MS/MS acquisition mode. The full mass scan acquired in the Orbitrap mass analyzer was from m/z 350 to 1500 with a resolution of 60000 (m/z 200). The MS/MS scans were also acquired by using an Orbitrap with a 30000 resolution (m/z 200), and the AGC target was set as custom. The spray voltage and the temperature of the ion transfer capillary were set to 2.6 KV and 320°C, respectively. The normalized collision energy for HCD and dynamic exclusion was set as 32% and 30 s, respectively.

For the glycoproteomic samples, peptides were separated through a gradient of up to 90% buffer B over 120 min at a flow rate of 600 nL/min. The gradient of the mobile phase started from 3% B to 10% B for 6 min and then was increased linearly to 40% B in 10 min, to 90% in 4 min, and maintained for 10 min. The LC-MS/MS system was operated in data-dependent MS/MS acquisition mode. The full mass scan acquired in the Orbitrap mass analyzer was from m/z 350 to 1500 with a resolution of 60000 (m/z 200). The MS/MS scans were also acquired by using an Orbitrap with a 30000 resolution (m/z 200), and the AGC target was set as standard. The spray voltage and the temperature of the ion transfer capillary were set to 2.6 KV and 320°C, respectively. The normalized collision energy for HCD and dynamic exclusion was set as 20%, 30%, 40% and 30 s, respectively.

For the phosphoproteomic samples, peptides were separated through a gradient of up to 90% buffer B over 60 min at a flow rate of 600 nL/min. The gradient of the mobile phase started from 1% B to 5% B for 3 min and then was increased linearly to 32% B in 50 min, to 90% in 2 min, and maintained for 5 min. The LC-MS/MS system was operated in data-independent MS/MS

acquisition mode. The full mass scan acquired in the Orbitrap mass analyzer was from m/z 350 to 1500 with a resolution of 60000 (m/z 200). The MS/MS scans were also acquired by using an Orbitrap with a 30000 resolution (m/z 200), and the AGC target was set as custom. The spray voltage and the temperature of the ion transfer capillary were set to 2.6 KV and 320°C, respectively. The normalized collision energy for HCD and dynamic exclusion was set as 32% and 30 s, respectively.

For TMT labeling proteomic and phosphoproteomic samples: peptides were separated through a gradient of up to 90% buffer B over 90 min at a flow rate of 600 nL/min. The gradient of the mobile phase started from 2% to 5% B for 1 min and then was increased linearly to 40% B in 70 min, to 45% in 10 min, to 100% in 5 min, and maintained for 5 min. The LC-MS/MS system was operated in data-dependent MS/MS acquisition mode. MS1 scans (350-1500 m/z) were acquired at a resolution of 60,000 with an AGC target of 200 % and a maximum injection time of 5 ms. MS/MS scans were acquired with 0.7 m/z isolation windows (scan range 110-3000 m/z), 59 windows, a maximum injection time of 100 ms and an AGC target of 300 %. The spray voltage and the temperature of the ion transfer capillary were set to 2.6 KV and 320 °C, respectively. The normalized collision energy for HCD and dynamic exclusion was set as 34% and 30 s, respectively.

- In supplementary material:

- o The dimensions of the column are not specified
- o The information for the DIA acquisition is missing: mass range, number of windows, window size, injection time

Response:

Thank you for your valuable comment. We have supplemented the relative information and the relevant information on DIA acquisition in Supplementary Information “Methods”. The details are as follows:

Supplementary Inf: Page 7, lines 175-177

“Samples were loaded onto capillary analytical column (150 μ m inner diameter, 15 cm length) packed in-house with C18 (1.9 mm) ReproSil particles (Dr. Maisch GmbH), with an EASY-nLC 1200 system (Thermo Fisher Scientific) coupled to the MS (Orbitrap 480, Thermo Fisher Scientific).”

Supplementary Inf: Page 7, lines 179-187

“For the proteomic samples, peptides were separated through a gradient of up to 90% buffer B over 20 min at a flow rate of 1 mL/min. The gradient of the mobile phase started from 3% B to 15% B for 0.5 min and then was increased linearly to 50% B in 15 min, to 90% in 1.5 min, and maintained for 3 min. The LC-MS/MS system was operated in data-independent MS/MS acquisition mode. The full mass scan acquired in the Orbitrap mass analyzer was from m/z 350 to 1500 with a

resolution of 60000 (m/z 200). The MS/MS scans were also acquired by using an Orbitrap with a 30000 resolution (m/z 200), and the AGC target was set as custom. The spray voltage and the temperature of the ion transfer capillary were set to 2.6 KV and 320 °C, respectively. The normalized collision energy for HCD and dynamic exclusion was set as 32% and 30 s, respectively.

Supplementary Inf: Page 7-8, lines 197-205

For the phosphoproteomic samples, peptides were separated through a gradient of up to 90% buffer B over 900 min at a flow rate of 600 nL/min. The gradient of the mobile phase started from 1% B to 5% B for 3 min and then was increased linearly to 32% B in 50 min, to 90% in 2 min, and maintained for 5 min. The LC-MS/MS system was operated in data-independent MS/MS acquisition mode. The full mass scan acquired in the Orbitrap mass analyzer was from m/z 350 to 1500 with a resolution of 60000 (m/z 200). The MS/MS scans were also acquired by using an Orbitrap with a 30000 resolution (m/z 200), and the AGC target was set as custom. The spray voltage and the temperature of the ion transfer capillary were set to 2.6 KV and 320 °C, respectively. The normalized collision energy for HCD and dynamic exclusion was set as 32% and 30 s, respectively.”

Reviewer #2:

This manuscript by Dong et al. presents MuPPE (Multi-level PTMs-Proteomic Enrichment), a platform for simultaneous and sequential enrichment of the proteome, glycoproteome, and phosphoproteome from a single biological sample. The authors demonstrate the utility of MuPPE across diverse sample types, including serum, CSF, cell lines, and mouse tissues, and apply it to study arsenical drug response and aging. However, several concerns remain regarding the platform's novelty, analytical rigor, and generalizability.

1. While MuPPE improves upon previous serial enrichment methods by being more time-efficient (4 h vs. 24 h) and compatible with low-input samples, it does not provide a clear conceptual advancement over prior work, such as that by Fang et al. (*Anal. Chem.*, 2019) and Abelin et al. (*Nat. Commun.*, 2023). The manuscript would benefit from a more explicit discussion of how MuPPE differentiates itself from these established platforms.

Response:

We sincerely appreciate your valuable suggestion. We acknowledge that this aspect needs further clarification, and thus, in the revised manuscript, we have provided a detailed comparison of MuPPE with several currently reported works. We have selected four representative studies (*Nat. Commun.*, 2023, 14, 1851; *Nat. Protoc.*, 2020, 15, 161; *ACS Chem. Biol.* 2019, 14, 58; *Anal. Chem.* 2020, 92, 1842; *J. Chromatogr. A*, 2025, 1739, 465525) involving simultaneous enrichment and identification of glycosylation and phosphorylation, along with one work (*Nat. Commun.*, 2023, 14, 1851) enabling multi-PTMs enrichment, for comprehensive benchmarking against our MuPPE platform. We have summarized and visualized the workflow comparisons of all these platforms, presenting a side-by-side schematic illustration to highlight key differences (Fig. 1, and Fig. S5 in the revised version).

In the original manuscript, we compared MuPPE with in-solution digestion in Fig. 1, demonstrating its superiority in precursor/peptide/protein identification counts and quantitative precision on serum samples. However, we acknowledge that these comparisons primarily focused on proteomic profiling rather than systematic multi-PTMs benchmarking against specialized methods. To address this gap, we have selected four representative studies (*Nat. Commun.*, 2023, 14, 1851; *Nat. Protoc.*, 2020, 15, 161; *ACS Chem. Biol.* 2019, 14, 58; *Anal. Chem.* 2020, 92, 1842; *J. Chromatogr. A*, 2025, 1739, 465525) involving simultaneous enrichment and identification of glycosylation and phosphorylation, along with one work (*Nat. Commun.*, 2023, 14, 1851) enabling multi-PTMs enrichment, for comprehensive benchmarking against our MuPPE platform. We have summarized and visualized the workflow comparisons of all these platforms, presenting a side-by-side schematic illustration to highlight key differences (Fig. 1 and Fig. S7 in the revised version).

Fig. 1. (Fig. S7 in the revised version) Summary diagram of the sequential enrichment process for different glycopeptides and phosphopeptides within state-of-the-art platforms according to the reports.

In Fig. 2. (Fig. 3c in the revised version), we specially present a comparative analysis of workflow time durations. The bar chart quantifies total processing time for MuPPE, state-of-the-art multi-PTMs workflows, and conventional glyco/phosphoproteomic methods. This direct comparison provides a clear quantitative basis for MuPPE’s time-saving advantages in multi-PTM analysis, highlighting its efficiency in accelerating experimental workflows.

Fig. 2. (Fig. 3c in the revised version) Step-by-step time breakdown of MuPPE workflow from protein extraction to enrichment, showcasing reduced sample transfers and time efficiency.

We summarized and defined comparative parameters covering on-line proteolytic digestion, workflow duration, trace sample applicability, sample type compatibility, accuracy & reproducibility, transfer times, identified omics types, labeling experiment feasibility, and identification counts, all systematically presented in Table 1 and Fig. 3 (Fig. 3d in the revised version).

The radar plot benchmarks MuPPE against five representative platforms across five performance parameters. MuPPE achieves the most extensive coverage in “on-bead digestion, trace sample applicability, sample compatibility, time efficiency, and minimized transfers”. These results underscore its unique capacity for integrated PTMs enrichment, strong adaptability to limited/heterogeneous samples, and capability for simultaneous multi-omics profiling. Shorter workflow durations accelerate high-throughput analysis, reduced transfer steps minimize sample loss and technical variability, and lower protein input benefits to low-abundance samples (e.g., clinical biopsies), firmly positioning it as a transformative tool for complex biological investigations.

Table 1. The benchmarking parameters summary for the comparison of each platform

Parameter	MuPPE	Nat. Commun., 2023	Nat. Protoc., 2020	ACS Chem. Biol. 2019	Anal. Chem. 2020	J. Chromatogr. A, 2025
On-line digestion	✓	×	×	×	×	×
Workflow time	4.5 h	23.5 h	31.5 h	25 h	20.5 h	20 h
Trace sample applicability	50 µg Protein	2000 µg	100 µg	300 µg	500 µg	50 µg Protein
Sample compatibility	Cells/body fluids/tissues	Tissues only	Plasma only	Tissues only	Tissues only	Single cells only
Accuracy (CV%)	12% (Serum Proteome)	-	-	-	-	-
Correlation coefficients	Proteome: 0.97-0.99 Glycoproteome: 0.93-0.96 Phosphoproteome: 0.96-0.97	Proteome: 0.96 Phosphoproteome: 0.90-0.91 Acetylome: 0.83-0.84	Proteome: 0.96-0.98	-	Proteome: 0.913-0.953	Glycoproteome: 0.97-0.99 Phosphoproteome: 0.98-0.99
Transfer times	1	5	2	3	2	2
Multi-omics layers	Proteome Glycoproteome Phosphoproteome	Immunopeptidome Proteome Ubiquitylome Phosphoproteome Acetylome	Glycoproteome Phosphoproteome	Glycoproteome Phosphoproteome	Proteome Glycoproteome Phosphoproteome	Glycoproteome Phosphoproteome
Labeling experiment	✓	✓	×	✓	×	×

Fig. 3. (Fig. 3d in the revised version) Radar plot comparing MuPPE with other multi-level proteomics methods across parameters, demonstrating MuPPE's superior performance.

Finally, to ensure methodological comparability, we benchmarked MuPPE against the conventional in-solution approach and two previously reported strategies under strictly matched conditions, including identical sample input, mass spectrometric acquisition parameters, database search settings, and data processing workflows. MuPPE yielded a substantially higher number of glycopeptide-spectrum matches than all three reference methods (Fig. 4e). Notably, the number of identified proteins and phosphopeptides did not differ significantly between MuPPE and the other methods, indicating that the integrated multi-omics enrichment design did not compromise proteome or phosphoproteome coverage (Fig. 4f and 4g). The primary advantage of MuPPE in glycoproteomics lies in its enhanced glycan coverage and structural diversity, as well as its superior capacity to capture a greater proportion of medium-length and structurally complex glycopeptides (Fig. 4h and 4i). Peptide length distribution analysis (Fig. 4j) revealed that MuPPE achieved a more uniform distribution of identified peptide lengths and a higher overall number of peptides compared

to the other three methods. This reflects MuPPE's superiority in capturing peptides across a broader size range, enabling more comprehensive proteomic coverage. Additionally, by comparing the overall composition of glycan units identified by the four methods (Fig. 4k), MuPPE enabled improved characterization of glycan structural features. These combined results highlight MuPPE's enhanced capability to dissect the complexity of glycoproteomes at both peptide- and glycan-structural levels, providing deeper insights into glycosylation biology.

Fig. 4. (Fig. 3 e-k in the revised version) **e-h** Quantitative comparisons across enrichment methods: glycopeptide spectrum matches (PSMs) (e), phosphopeptides (f), proteins (g), and glycans (h). **i** Distribution of glycan types identified across methods, showing relative proportions of high-mannose, fucosylated, sialylated, and complex glycans. **j** Peptide length distribution for glycopeptides captured by MuPPE versus conventional approaches. **k** Comparison of glycan structural features, including glycan unit number across enrichment strategies.

In summary, the revised manuscript now (i) establishes MuPPE's advantages through a structured, like-for-like benchmark against state-of-the-art platforms (Fig. 3 c-k, Supplementary Fig. S7) and (ii) provides concrete, method-level instructions for integrating the three molecular layers (Methods; Supplementary Methods). We believe these changes directly address the reviewer's concerns and strengthen the manuscript's clarity, novelty, and practical utility.

Manuscript changes: Page 10, line 245-279 revised manuscript:

Collectively, the MuPPE workflow facilitates multi-level proteomic analysis by enabling sequential analysis of the glycoproteome, followed by the phosphoproteome from just single biological sample, previously only feasible through parallel processing of multiple sample aliquots.

In multi-PTMs research, workflow duration is a critical parameter that directly determines overall efficiency. MuPPE excelled in time efficiency (total workflow < 4 hours vs. more than 32 hours for others, Fig. 3c) and sample transfers (2 transfers vs. 5-10 transfers for others), positioning it as a leading method. These comparisons fully demonstrate the efficiency and advantages of MuPPE in multi-levels proteomic research. A radar plot (Fig. 3d) benchmarked MuPPE against five state-of-the-art methods^{15, 31, 34-36}, evaluating parameters such as on-bead digestion, trace sample applicability, sample compatibility, time efficiency, and minimized transfers numbers.

We compared MuPPE with conventional in-solution digestion protocols at the proteome level (Fig. 1), demonstrating its superiority in precursor, peptide, and protein identifications, as well as quantitative precision in serum samples. However, these comparisons primarily addressed proteomic profiling rather than systematic multi-PTMs benchmarking against dedicated enrichment strategies. To fill this gap, we evaluated MuPPE alongside four representative studies that reported simultaneous enrichment and identification of glycosylation and phosphorylation, including one work¹⁵ that enabled broader multi-PTMs capture. We summarized and visualized the workflows of all platforms and presented a side-by-side schematic to highlight methodological distinctions (Fig. S7). To ensure methodological comparability, we benchmarked MuPPE against the conventional in-solution approach and two previously reported strategies^{31, 34} under strictly matched conditions, including identical sample input, mass spectrometric acquisition parameters, database search settings, and data processing workflows. MuPPE yielded a substantially higher number of glycopeptide-spectrum matches than all three reference methods (Fig. 3e). Notably, the number of identified proteins and phosphopeptides did not differ significantly between MuPPE and the other methods, indicating that the integrated multi-omics enrichment design did not compromise proteome or phosphoproteome coverage (Fig. 3f and 3g). The primary advantage of MuPPE in glycoproteomics lies in its enhanced glycan coverage and structural diversity, as well as its superior capacity to capture a greater proportion of medium-length and structurally complex glycopeptides (Fig. 3h-3i). Peptide length distribution analysis (Fig. 3j) revealed that MuPPE achieved a more uniform distribution of identified peptide lengths and a higher overall number of peptides compared to the other three methods. This reflects MuPPE's superiority in capturing peptides across a broader size range, enabling more comprehensive proteomic coverage. Additionally, by comparing the overall composition of glycan units identified by the four methods (Fig. 3k), MuPPE enabled improved characterization of glycan structural features. These combined results highlight MuPPE's enhanced capability to dissect the complexity of glycoproteomes at both peptide- and glycan-structural levels, providing deeper insights into glycosylation biology.

Supplementary Data: Page 10-11, line 277-308.

Benchmark workflow

MuPPE workflow: Mouse brain lysates (20 μg) were diluted to 1 $\mu\text{g}/\mu\text{L}$ and incubated with 0.6 mg maltose beads for 10 min, followed by addition of 63 μL acetonitrile. Samples were washed three times with 200 μL of 80% ethanol, then resolubilized in 8 μL of 6 M urea and diluted with 40 μL of ammonium bicarbonate. Proteins were digested with 0.5 μg trypsin for 2.5 h, and the reaction was quenched with 2.6 μL TFA and 206.3 μL ACN. After 15 min incubation, glycopeptides were captured by centrifugation. Maltose beads were washed three times with 50 μL of 80% ACN/1% TFA and eluted with 30 μL of 30% ACN/1% FA ($\times 3$). Flow-through and wash fractions were combined, vacuum-dried, and incubated with 0.5 mg IMAC beads in 50 μL 80% ACN/6% TFA for 1 h. IMAC beads were sequentially washed with 50 μL 80% ACN/6% TFA, $2 \times 100 \mu\text{L}$ 50% ACN/6% TFA/200 mM NaCl, $2 \times 200 \mu\text{L}$ 30% ACN/0.1% FA, and $2 \times 100 \mu\text{L}$ 30% ACN. Phosphopeptides were finally eluted twice with 30 μL 10% ammonia solution and pooled.

In-solution digestion: Mouse brain lysates (20 μg) were denatured in 8 M urea with protease inhibitor cocktail (final 1 $\mu\text{g}/\mu\text{L}$). Proteins were reduced with 1 μL 200 mM DTT and alkylated with 1 μL 400 mM IAA. After dilution with 140 μL ammonium bicarbonate, proteins were digested with 0.5 μg trypsin. Digestion was quenched, and peptides were desalted using C18 SPE columns: columns were activated, samples loaded, washed with 20 mL water/0.1% FA, and eluted with $2 \times 1 \text{ mL}$ 50% ACN/0.1% FA. Eluates were vacuum-dried. For glycopeptide enrichment, 20 μg desalted peptides were dissolved in 30 μL 80% ACN/1% TFA and incubated with 0.6 mg maltose beads for 15 min. Beads were washed (80% ACN/1% TFA, 50 $\mu\text{L} \times 3$) and eluted (30% ACN/1% FA, 30 $\mu\text{L} \times 3$). For phosphopeptide enrichment, 20 μg peptides were dissolved in 50 μL 80% ACN/6% TFA and incubated with 0.5 mg IMAC beads for 1 h. After washing as described above, phosphopeptides were eluted twice with 30 μL 10% ammonia solution and pooled.

Reference methods: For benchmarking, two reported workflows were included. In method 1⁷, 20 μg digested peptides were dissolved in 50 μL 80% ACN/1% TFA, loaded onto a maltose-packed tip column, and processed as described above. The flow-through was dried, resolubilized in 50 μL 80% ACN/6% TFA, and incubated with 0.5 mg IMAC beads, followed by washing and elution as above. In method 2⁸, 20 μg desalted peptides were dissolved in 50 μL 80% ACN/6% TFA and incubated with 0.5 mg IMAC beads for 1 h. Beads were washed as above, and phosphopeptides were eluted with 30 μL 10% ammonia (twice). The IMAC flow-through was concentrated, loaded onto OASIS MAX columns (1 mg), equilibrated with 95% ACN/1% TFA, and subjected to sequential washing and elution according to the original protocol.

Added new Fig.3c-k and Fig. S7.

2. The study exclusively uses label-free quantification. For broader applicability, it is important to demonstrate or at least discuss MuPPE's compatibility with isobaric labeling strategies such as TMT. Without this, the platform is likely to remain limited to exploratory analyses, restricting its potential for large-scale, multiplexed quantitative studies.

Response:

We appreciate the reviewer's valuable suggestion regarding the compatibility of MuPPE with isobaric labeling strategies, which is critical for expanding its applicability in large-scale multiplexed studies. We agree that addressing this aspect enhances the platform's utility, and we have supplemented both experimental data and discussion to clarify this point.

In the revision, we developed and validated a fully TMT-compatible implementation of MuPPE, and we provide a step-by-step protocol in the Supplementary Information. Methodologically, we adopted a label-then-enrich design: individual samples are digested in 100 mM TEAB (substituted for ammonium bicarbonate to preserve NHS-ester reactivity), TMT-labeled, pooled in equal amounts, and then subjected to MuPPE's sequential glycopeptide-first and phosphopeptide enrichments. Processing all channels as a single pooled sample ensures identical downstream handling and thus preserves inter-channel ratio fidelity, while retaining MuPPE's orthogonal fractionation logic. We did not include a TMT-based glycoproteomics analysis. At present, robust and widely validated pipelines in our environment for reporter-ion-based, site-resolved N-glycopeptide quantification-with rigorous multi-level FDR control (peptide/glycan/site) and comprehensive interference corrections (isotopic impurity and co-isolation)-are not yet available. We will incorporate glyco-TMT once such tools are mature and have been validated in our laboratory.

Evidence of TMT compatibility (Fig. 5). For each feature, within-run $\log_2(\text{MuPPE}/\text{In-solution})$ ratios were computed across three matched 2-plex TMT experiments, and statistical significance was assessed using paired t-tests with Benjamini-Hochberg FDR control. The resulting volcano plots show a systematic rightward shift for both the proteome and phosphoproteome, indicating that MuPPE yields higher quantitative intensities for the majority of features relative to the in-solution workflow. These results demonstrate that MuPPE is fully compatible with isobaric multiplexing and strengthen its suitability for both exploratory and large-scale quantitative studies.

Fig. 5. (Fig. S1 in the revised version) Paired TMT comparison of MuPPE vs in-solution workflows. Volcano plots for the global proteome (left) and phosphoproteome (right). Each point represents a protein group or a phosphopeptide. The x-axis shows within-run \log_2 intensity ratios (MuPPE/In-solution) computed after isotopic correction; the y-axis shows $-\log_{10} P$ from paired t-tests across three matched 2-plex TMT runs. Blue stars indicate features significantly higher in MuPPE at the stated thresholds; grey points are not significant. Under equal input and TMT multiplexing, MuPPE yields higher and more consistent quantitative signal than the in-solution workflow across both the proteome and phosphoproteome layers.

Manuscript changes:

Page 4, Line 128-130 in revised manuscript:

Additionally, we compared the compatibility of the in-solution method and the MuPPE method with TMT labeling experiments, and confirmed that MuPPE is compatible with TMT-based multiplexed quantification (Fig. S1).

Supplementary Inf: Page 8, Lines 206-215.

For TMT labeling proteomic and phosphoproteomic samples: peptides were separated through a gradient of up to 90% buffer B over 90 min at a flow rate of 600 nL/min. The gradient of the mobile phase started from 2% to 5% B for 1 min and then was increased linearly to 40% B in 70 min, to 45% in 10 min, to 100% in 5 min, and maintained for 5 min. The LC-MS/MS system was operated in data-dependent MS/MS acquisition mode. MS1 scans (350-1500 m/z) were acquired at a resolution of 60,000 with an AGC target of 200 % and a maximum injection time of 5 ms. MS/MS scans were acquired with 0.7 m/z isolation windows (scan range 110-3000 m/z), 59 windows, a maximum injection time of 100 ms and an AGC target of 300 %. The spray voltage and the

temperature of the ion transfer capillary were set to 2.6 KV and 320 °C, respectively. The normalized collision energy for HCD and dynamic exclusion was set as 34% and 30 s, respectively.

Supplementary Inf: Page 8-9, Lines 217-250.

TMT labeling experimental process: For the MuPPE-TMT workflow, approximately 20 µg of mouse brain proteins (lysed in 8 M urea/HEPES) were reduced with 1.1 µL of 200 mM DTT (final concentration 10 mM, 37 °C, 60 min) and alkylated with 1.12 µL of 800 mM IAA (final concentration 40 mM, dark, 30 min). Each sample was incubated with 0.6 mg maltose beads (5 µL suspension in water) and 63 µL acetonitrile (final concentration 70%) for 10 min, centrifuged at 4000 rpm for 4 min, and washed three times with 200 µL of 80% ethanol. Beads were resolubilized in 8 µL of 6 M urea, sonicated to disperse, diluted with 40 µL of 100 mM TEAB, and digested with 2 µL trypsin (0.5 µg, 1:40) at 37 °C for 2 h. The digested peptides were labeled with TMT reagents by dissolving 40 µg of TMT in 2 µL anhydrous acetonitrile, incubating with 50 µL peptide solution at room temperature for 1 h, and quenching with 1 µL of 5% hydroxylamine for 15 min.

For the In-solution-TMT workflow, 20 µg of mouse brain proteins (8 M urea/HEPES) were reduced with 1.1 µL of 200 mM DTT (10 mM, 37 °C, 60 min), alkylated with 1.12 µL of 800 mM IAA (40 mM, dark, 30 min), and digested with 2 µL trypsin (0.5 µg, 1:40) overnight at 37 °C. Digestion was stopped by adding 2 µL of 1% TFA (final concentration 0.1%). Peptides were desalted with 0.75 mg C18 tips using the following steps: activation with 50 µL 50% acetonitrile/0.1% FA, equilibration with 50 µL 0.1% FA, sample loading in 30 µL 0.1% FA, washing with 50 µL 0.1% FA (×2), and elution with 30 µL 50% acetonitrile/0.1% FA (×2). The eluates were vacuum dried and subjected to TMT labeling as described above (2 µL TMT reagent in 2 µL anhydrous acetonitrile, 50 µL peptide solution, 1 h incubation at room temperature, and quenching with 1 µL of 5% hydroxylamine for 15 min).

For glycopeptide enrichment, the TMT-labeled digests from both MuPPE and in-solution workflows were adjusted with 2.37 µL TFA and 189.5 µL acetonitrile to maltose-binding conditions and incubated with 0.6 mg maltose beads for 15 min. Bound glycopeptides were washed three times with 50 µL of 80% acetonitrile/1% TFA, and eluted three times with 30 µL of 30% acetonitrile/1% FA. The flow-through and wash fractions were combined, dried, and resolubilized in 200 µL of 80% acetonitrile/6% TFA for phosphopeptide enrichment. IMAC tips containing 1 mg beads were equilibrated with 100 µL of 80% acetonitrile/6% TFA, loaded with 100 µL of sample solution, and sequentially washed with 50 µL 80% acetonitrile/6% TFA, 2 × 100 µL 50% acetonitrile/6% TFA/200 mM NaCl, 2 × 100 µL 30% acetonitrile/0.1% FA, and 2 × 50 µL 30% acetonitrile. Bound phosphopeptides were eluted twice with 30 µL of 10% ammonia, vacuum dried, and stored at -80 °C.

Finally, glycopeptide and phosphopeptide eluates, as well as protein fractions from the combined flow-throughs, were dried, resolubilized in 30 µL of 0.1% FA, desalted with 0.75 mg C18 tips (activation: 50 µL 50% acetonitrile/0.1% FA; equilibration: 50 µL 0.1% FA; loading: 30 µL 0.1%

FA; wash: $2 \times 50 \mu\text{L}$ 0.1% FA; elution: $2 \times 30 \mu\text{L}$ 50% acetonitrile/0.1% FA), dried again, and stored at -80°C until LC–MS/MS analysis.

Added new Fig. S1

3. Although MuPPE simplifies PTM workflows, it identifies substantially fewer phosphopeptides compared to state-of-the-art phosphoproteomic methods. This limitation may significantly reduce its utility for phosphorylation-centric studies. The authors should benchmark their phosphopeptide recovery against existing methods and address whether the sequential design inherently limits phosphopeptide enrichment. Reduced detectability could profoundly impact biological discovery and interpretation.

Response:

We appreciate the reviewer’s concern that MuPPE may identify fewer phosphopeptides than state-of-the-art, phospho-dedicated workflows, and we agree that fair benchmarking and an explicit discussion of any sequential-design effect are essential. We acknowledge that specialized, ultra-deep phosphoproteomics (e.g., mg-scale inputs, multi-round IMAC/TiO₂, and extensive offline fractionation) can yield larger site catalogs than any single-shot, multi-omics workflow. MuPPE is designed to balance depth with integrative coverage (proteome/glyco/phospho) and sample efficiency.

Comprehensive Benchmarking Against State-of-the-Art Platforms: To avoid confounding by input amounts and fractionation depth, we performed matched comparisons (identical starting input, single-shot LC-MS/MS, the same search/filters and FDR, and identical site-localization thresholds) between MuPPE and two commonly used IMAC-based reference workflows. Across biological replicates, the numbers of identified phosphopeptides and localized sites did not differ significantly between MuPPE and the references (**Fig. 4. above**). These results indicate that, under matched micro-scale and single-shot conditions, MuPPE does not compromise phosphoproteome coverage.

Sequential-design assessment: We explicitly evaluated order effects in our tandem HILIC-IMAC study (“Tandem HILIC-IMAC strategy for simultaneous N-glycoproteomics and phosphoproteomics in aging mouse brain.” *J. Chromatogr. A*, 2025, 1739). Running HILIC-IMAC (glyco-first) yielded more phosphopeptides and slightly more glycopeptides than IMAC-HILIC. Critically, glycan-diagnostic signals (HexNAc/NeuAc) in the phosphopeptide fraction were lower with HILIC-IMAC than with IMAC-HILIC, indicating far less glycopeptide carryover. Conversely, phosphopeptide carryover into the glycopeptide fraction was minimal with HILIC-IMAC. Mechanistically, depleting glycopeptides first does not introduce conditions that favor phosphopeptide loss, consistent with the minimal phospho- signal observed in glycol- fractions (Fig. 6).

Fig. 6. Performance assessment of overlap and repeatability. a The number of the oxonium ion-containing MS2 spectra and the total MS2 spectra in PP fraction using HILIC-IMAC strategy and IMAC-HILIC strategy. b The number of phosphopeptides in GP fraction using HILIC-IMAC strategy and IMAC-HILIC strategy. c The number of identified target peptides using HILIC-IMAC strategy and IMAC-HILIC strategy (*J. Chromatogr. A*, 2025, 1739).

Finally, TMT label-then-enrich validation and quantitative behavior: We conducted three matched 2-plex TMT experiments. Within-run paired analyses show a systematic right-shift in volcano plots for both proteome and phosphoproteome layers, consistent with higher quantitative intensities for the majority of features relative to in-solution workflows at the same input (**Fig. 5. above**).

Collectively, these results show that MuPPE does not intrinsically sacrifice phosphoproteome performance. Under strictly matched, single-shot conditions, phosphopeptide and localized-site yields are comparable to IMAC references; our tandem HILIC-IMAC study show that ordering HILIC→IMAC with markedly lower glycan carryover in the PP fraction and minimal phosphopeptide carryover to the GP fraction. In addition, TMT label-then-enrich runs preserve ratio fidelity and increase quantitative signal. Thus, perceived deficits versus ultra-deep, fractionated phospho-dedicated pipelines primarily reflect scale/fractionation, not a limitation of MuPPE; when maximal site catalogs are needed, MuPPE can be paired with deeper fractionation without altering its integrative design.

4. The sequential nature of MuPPE leads to unequal input across glyco-, phospho-, and total proteome fractions, introducing potential biases in quantitation and reproducibility. The manuscript should address whether and how these differences are normalized or controlled.

Response:

We thank the reviewer for highlighting the risk that a sequential workflow could yield unequal material input across the glyco-, phospho-, and total proteome fractions, potentially biasing quantitation and reproducibility. Our study design and analysis pipeline explicitly mitigate these risks at the experiment, normalization, and quality-control levels, as detailed below.

Study design and input control: Each biological replicate begins from a fixed, equal protein input (e.g., 50 µg per mouse) with constant material-to-sample ratios at every enrichment step, and matched LC-MS acquisition time/gradients/instrument settings (Supplementary Inf: “LC-MS/MS analysis of different proteomic samples”; “Integrated proteome, glycoproteome, and phosphoproteome analysis of aging mouse samples and NB4 cells using the MuPPE platform” section). Injections are randomized and interleaved with blank/QC runs to monitor carryover and drift. These controls ensure that any post-enrichment differences reflect expected yield characteristics of each chemistry rather than arbitrary input disparities.

Data Normalization: For all three omics layers, we performed log₂ transformation and median normalization prior to downstream analyses. This step further reduces the impact of absolute enrichment efficiency differences and enables reliable comparison across fractions.

Quantitative Consistency Verification: We systematically assessed the quantitative reproducibility of each omics dataset by calculating pairwise Pearson correlation coefficients among biological replicates. As shown in **Fig. 7**, all three omics datasets displayed high correlation (Proteome: 0.97-1.00, Phosphoproteome: 0.83-0.96, Glycoproteome: 0.76-0.87), confirming strong consistency and that the sequential workflow does not introduce systematic quantitation bias.

Literature Validation: Our previous publication (*Ding et al., J. Chromatogr. A* 2025, 1739.) systematically validated this approach by tandem enrichment and integrated analysis of proteome, glycoproteome, and phosphoproteome in mouse brain tissue, achieving technical repeatability with a median Pearson’s r of 0.98. This further demonstrates that the tandem strategy provides high reliability and quantitative reproducibility for multi-omics integration.

Together, these measures ensure that the observed quantitative differences truly reflect biological variation rather than technical artifacts introduced by the sequential workflow.

Fig. 7. Pairwise Pearson correlation matrices of quantitative data among biological replicates for the proteome, phosphoproteome, and glycoproteome datasets after normalization. Each cell shows the Pearson correlation coefficient (r) between two samples, with the corresponding scatter plots and distribution curves. The consistently high correlation coefficients (Proteome: 0.97-1.00,

Phosphoproteome: 0.83-0.96, Glycoproteome: 0.76-0.87) indicate strong quantitative reproducibility and confirm that the sequential enrichment strategy does not introduce systematic bias (Data shown are from choroid plexus samples in the mouse aging cohort).

Manuscript changes:

Supplementary Data: Pages 9-10, lines 252-275

Integrated proteome, glycoproteome, and phosphoproteome analysis of aging mouse samples and NB4 cells using the MuPPE platform: For each aging mouse sample (choroid plexus, CSF, and serum) and for NB4 cell lysates, total protein concentration was quantified using the BCA Protein Assay Kit (Beyotime, China) following the manufacturer's instructions. Exactly 50 µg of total protein from each sample was used as input for the subsequent MuPPE workflow. In MuPPE process. Denaturation, PAC and on-line digestion on Click-Maltose, Then, adjust the tryptic sample solution to 80% acetonitrile (ACN)/1% trifluoroacetic acid (TFA) by adding ACN and TFA and incubate for 10 min for glycopeptides enrichment. Finally, adjust the flow-through condition for phosphopeptide enrichment according to the manufacturer's instructions, and then incubate for 10 minutes to facilitate phosphopeptide enrichment. The remaining (flow-through) peptides not bound in the glyco/phospho enrichment steps were retained for proteomics. All peptide fractions were analyzed using high-resolution LC-MS/MS platforms.

For data processing and integration across the three layers, contaminants and decoys were removed and PTM sites were required to meet localization probability thresholds defined by the respective search engines; raw intensities were normalized at the run level (total ion current, TIC) and subsequently log₂-transformed. To ensure both depth and reliability, layer-specific coverage thresholds were applied: features in the proteome and phosphoproteome were retained only if quantified in ≥80% of replicates within at least one group, whereas the glycoproteome retained features quantified in ≥50% of replicates, acknowledging the higher intrinsic sparsity of glycopeptide data. Missing values were then handled in a layer-specific manner: for the proteome and phosphoproteome, partially missing entries were imputed within layer on log₂-transformed data using the k-nearest neighbors algorithm (KNN, k=5), while for the glycoproteome, missing values were treated as true non-detections and replaced with 0 at the raw scale (followed by $\epsilon + \log_2$ transformation to avoid undefined values). Finally, for cross-layer comparability, all features were Z-scored within each layer (per feature across samples) before integration and downstream statistical analyses.

5. The case studies using NB4 cells and aging mice are promising, but the biological insights remain largely descriptive. There is minimal integration across PTM layers, and mechanistic interpretation is limited. Given that this manuscript is primarily focused on technology

development, a more in-depth comparison with existing platforms and a stronger discussion of biological implications would be highly beneficial for readers.

Response:

We thank the reviewer for this valuable suggestion. In the revision we substantially strengthened (1) cross-layer PTMs integration and mechanistic interpretation for both the aging mouse cohorts and NB4 cells, and (2) head-to-head context with existing platforms.

To address the review's concern regarding systematic integration of proteome, phosphoproteome, and glycoproteome data, we have undertaken a comprehensive re-analysis to uncover cross-omics associations of aging and arsenic-treated NB4 cells dataset, as detailed below:

(i) For the aging dataset, we performed multi-layered KEGG pathway enrichment analyses (Fig. 8. Fig. 5f in the revised version), allowing us to identify both shared and unique biological processes and pathways regulated at the total protein, phosphorylation, and glycosylation levels.

Notably, pathways related to synaptic signaling, cell adhesion, extracellular matrix organization, and immune response were consistently enriched across all three proteome, phosphoproteome, and glycoproteome layers, suggesting that these fundamental processes are subject to coordinated multi-PTM regulation. For example, the convergence of synaptic signaling and cell adhesion pathways in proteome, phosphoproteome, and glycoproteome highlights their critical role in neural plasticity and tissue remodeling during aging and disease.

In contrast, metabolic pathways (e.g., energy metabolism, lipid metabolism) were predominantly enriched in the total proteome and phosphoproteome layers, whereas pathways associated with immune modulation and glycan biosynthesis were more specific to the glycoproteome, reflecting the specialized roles of different PTMs.

These findings demonstrate that multi-omics integration not only validates known biological themes but also uncovers PTM-specific and multi-layered regulatory networks that would be missed by single-omics analysis alone. The results and biological implications of this multi-omics integration have been detailed in the revised Results and Discussion sections.

Fig. 8. (Fig. 5f in the revised version) Integrated KEGG enrichment analyses of proteome, phosphoproteome, and glycoproteome datasets.

(ii) Based on multi-omics data analysis, we propose a mechanism where arsenic acts via a membrane-to-nucleus signaling cascade. It modulates membrane proteins through glycosylation and perturbs the PI3K-AKT-mTOR pathway via phosphorylation, impacting cell cycle and chromosomal organization, possibly arresting the cell cycle and inducing apoptosis (supported by flow cytometry showing G2/M phase accumulation and phosphoproteomic changes in cell cycle regulators). Also, arsenic may disrupt genome-wide transcription by affecting chromatin-associated proteins. Taking IGF2R, a key membrane protein in the PI3K-AKT-mTOR pathway, as an example, although its global protein level shows no significant change, multi-omics data reveal 13 differential PTMs sites (12 N-glycosylation and 1 phosphorylation). This shows arsenic exerts effects by rewiring PTM-mediated signaling networks, a mechanism only detectable through multi-omics profiling.

Fig. 9. a Schematic of PI3K-AKT-mTOR pathway perturbation by arsenic, with downstream effects on cell cycle and chromatin organization. Proteome (green), glycoproteome (orange), and phosphoproteome (blue) alterations are mapped. b Flow cytometry analysis of NB4 cells treated with increasing arsenic concentrations, showing dose-dependent G2/M phase accumulation. c Differential expression of protein relative abundance, glycosylation sites and phosphorylation sites of IGF2R protein with the log2 fold change values. Different symbols representing up-regulation (upward triangles), down-regulation (downward triangles), and the color intensity indicating the significance of the change.

In summary, across both applications, the organismal aging cohort and the arsenic-perturbed NB4 model, our cross-omics framework reveals a coherent picture of multi-layer regulation: (i) system-level convergence of synaptic signaling, cell-adhesion/extracellular-matrix, and immune programs across the proteome, phosphoproteome, and glycoproteome in aging, and (ii) The IGF2R case exemplifies this principle, where negligible protein-level change ($\log_2FC \approx 0$) co-occurs with pronounced site-specific phosphorylation and glycosylation remodeling, indicating pathway rewiring that would be invisible to any single layer. Taken together, these results demonstrate the robustness and generalizability of our integrative strategy, provide mechanistic links from PTMs remodeling to pathway activity and phenotype, and address the reviewer's request for systematic cross-omics integration and biological interpretation.

We sincerely appreciate your constructive suggestion regarding strengthening the discussion of biological implications, which has been highly instrumental in enhancing the depth and impact of our manuscript. In response to this comment, we have supplemented both the Results and Discussion sections with dedicated content to elaborate on the biological implications of our findings.

Manuscript changes:

Page 17-18, Line 431-446 in revised manuscript:

To investigate aging-related molecular changes, we conducted time-series clustering of the proteome, phosphoproteome, and glycoproteome across 2, 12, and 24 M, followed by KEGG pathway enrichment analysis (Fig. 5f). In the phosphoproteome, pathways associated with aging—including oxidative phosphorylation, Alzheimer’s disease, and Parkinson’s disease—were enriched early in young mice and remained detectable throughout all age groups. The proteome exhibited significant enrichment in pathways such as long-term depression and peroxisome function specifically at 24 M, suggesting a delayed accumulation compared to phosphorylation-mediated changes. The ECM-receptor interaction pathway showed distinct regulatory patterns: glycoprotein levels increased progressively with age, while phosphorylation peaked at 12 M before declining by 24 M. Enrichment of the cytoskeleton pathway in muscle cells was observed in both the proteome and phosphoproteome, with an initial increase followed by a decrease; phosphorylation changes, however, declined more gradually. Several pathways were uniquely altered in specific PTM layers: focal adhesion decreased with age in the phosphoproteome, whereas oxidative phosphorylation was broadly enriched. In the glycoproteome, the lysosome pathway increased with age, along with upregulation of antigen processing and ECM-related processes.

Page 21, Line 514-539 in revised manuscript:

Building on the multi-omics dataset analysis, we propose a mechanistic model where arsenic exerts its effects through a membrane-to-nucleus signaling cascade (Fig. 6i), primarily mediated by glycosylation-modulated membrane proteins and phosphorylation-driven PI3K-AKT-mTOR pathway perturbation. The glycoproteomic data revealed that arsenic induces glycosylation alterations in membrane-anchored receptors (e.g., components of the PI3K-AKT-mTOR pathway upstream regulators). These glycosylation changes might act as “molecular switches” to modulate membrane protein stability, localization, and ligand-binding capacity. We also observed that the arsenic treatment significantly impacted the cell cycle and chromosomal organization pathways, particularly in the regulation of the proteome and phosphoproteome during the M phases. This suggests that arsenic exerts profound effects on these two critical signaling pathways, potentially by arresting the cell cycle and disrupting chromosomal functions, thereby promoting apoptosis. Flow cytometry analysis (Fig. 6j) showed the G2/M phase accumulation, consistent with the phosphorylation changes in cell cycle regulators (e.g., CDK, CCNB proteins in Fig. 6j) directly support this arrest. GO enrichment results of nuclear functions in the key modules, combined with phosphoproteomic changes in chromatin-associated proteins (e.g., histone modifiers), suggest arsenic might disrupt genome-wide transcriptional programs and link membrane-to-nucleus signaling to nuclear architecture and function.

Focusing on IGF2R, a key membrane protein identified in the PI3K-AKT-mTOR pathway, we interrogated its individual proteomic, glycoproteomic, and phosphoproteomic alterations (Fig. 6k). At the global protein level, IGF2R abundance showed no significant fold-change between arsenic-treated and control groups. However, its glycoproteomic and phosphoproteomic data revealed notably PTMs-specific alterations. we identified 13 differential PTMs sites, including 12 N-

glycosylation sites and 1 phosphorylation site. Upregulated and downregulated PTMs were denoted by upward and downward triangles, respectively. Without glycoproteomic/phosphoproteomic data, IGF2R would appear “unchanged” at the protein level, masking its potential role as a PTMs-driven hub. Thus, arsenic exerts its effects not by altering protein abundance broadly but by rewiring PTM-mediated signaling networks—a mechanism only detectable via multi-omics profiling.

Page 28-29, Line 722-731 in revised manuscript:

Pathway temporal clustering and visualization: Proteome, phosphoproteome, and glycoproteome profiles across 2, 12, and 24 M were Z-score normalized and subjected to fuzzy c-means clustering using Mfuzz (R, v2.60.0) to capture temporal expression patterns. Cluster-specific KEGG enrichment was performed with clusterProfiler (v4.10.0), retaining pathways with $FDR < 0.05$, $FC > 1.5$. The resulting heatmap integrates three layers of information: (i) Membership plots (left), showing cluster-wise temporal trends with membership values; (ii) Expression heatmaps (middle), displaying Z-scored expression values across 2 M, 12 M, and 24 M; and (iii) KEGG pathways (right), annotated for each cluster, with functionally relevant pathways highlighted. Results were visualized as integrated membership trajectories, expression heatmaps, and enriched pathways using ggplot2 and ComplexHeatmap, ensuring full reproducibility in R (v4.3.1).

Added new integrative Figure. (Fig. 5 and 6 in the revised version).

6. The manuscript requires extensive language polishing. For example, there are several sentences starting with “And”.

Response:

We will carefully review and polish the language of the manuscript to improve its quality, eliminating sentences starting with “And” and ensuring a more fluent and professional writing style.

REVIEWER COMMENTS

Reviewer #1:

I have read with interest the extensive revision of this manuscript, in which I think the authors have done an excellent job addressing all my previous concerns.

Some minor points:

1) Figure 4f-I assume that “ns” on top of the figure stands for “not significant.” However, I don’t think such a statement is relevant when comparing the number of phosphopeptides. If the authors wish to retain it, they should indicate which type of statistical test was used to derive that conclusion.

Response:

Thank you for the suggestion. We agree that “ns” labels are unnecessary on a summary count plot. We have removed all “ns” annotations from Fig. 3f. The main text has been updated accordingly.

Manuscript changes:

Fig. 3f “ns” annotations have been removed.

Fig. 3. Serialized analysis of glycoproteome, phosphoproteome and proteome using MuPPE

Page 10, Line 268-270 in revised manuscript:

“Notably, the number of identified proteins and phosphopeptides were comparable between MuPPE and the other methods, indicating that the integrated multi-omics enrichment design did not compromise proteome or phosphoproteome coverage (Fig. 3f and 3g).”

2) Figure 5-The authors perform a KEGG enrichment analysis to derive biological insights into the pathways regulated at the protein, phosphoproteome, and glycoproteome levels. However, this type of analysis does not consider the regulatory layers of the phospho- and glycoproteomes. For instance, when considering a glycoprotein for such enrichment, how do the authors account for cases where a protein contains both glyco-sites that are up-regulated and down-regulated (as shown in Figure 6C)? There are annotation tools, such as PTM-SEA, that take this into account and provide more meaningful information for this type of data.

Response:

Thank you for this insightful comment. We agree that gene-centric KEGG can mask site-level regulation in phospho- and glycoproteomes, particularly when a protein harbors both up- and down-regulated sites. We now perform a PTM-aware, site-centric analysis for the phosphoproteome. Specifically, we computed site-level moderated *t* statistics (older vs 2M) and ran PTM-SEA (ssGSEA2.0) against PTMsigDB kinase/motif/pathway signatures. This preserves directionality at each phosphosite, so proteins with mixed regulation contribute appropriately to both up/down signatures. Results are reported as normalized enrichment scores (NES) with P-value and are shown in **Supplementary Fig. S18**. Compared with 2 M, 24 M show positive enrichment of PI3K-Akt pathway and kinase-centric signatures AKT1, CDK2, SIK2, MAPKAPK5/RSK2, with larger NES and significance than 12 M, indicating progressive activation with age. In contrast, CK2/CSNK2A2, NLK, and ERK5/MAPK7 signatures are negatively enriched at both ages. These results indicate a shift toward AKT/ cell-cycle kinase activity with a concomitant decline of CK2-mediated regulation during physiological aging. Consistent with GO results, PTM-SEA indicates age-progressive activation of AKT/RSK-cell-cycle signaling linked to RNA/spliceosomal regulation, with concomitant attenuation of CK2/ERK5/NLK pathways that support actomyosin architecture and neuronal/synaptic homeostasis.

As for glycoproteome; there is currently no curated, PTM-SEA-style library for site-resolved, directionality-aware N-glycosylation signatures comparable to PTMsigDB. Therefore, we retained the protein-centric KEGG analysis for the glycoproteome.

Manuscript changes:

Added new Fig. S18.

Fig. S18. PTM signature enrichment analysis (PTM-SEA) bubble plot for 12 M/2 M and 24 M/2 M contrasts. Circles are coloured by normalized enrichment scores (NES) (red, positive enrichment; blue, negative) and scaled by significance (bubble area, proportional to $-\log_{10}$ P-value); signature names are shown on the left.

Page 16-17, Line 404-414 in revised manuscript:

“We further conducted site-level pathway enrichment using PTM-SEA, which infers PTM regulators from substrate-specific signatures⁵⁶. As shown in Fig. S18, relative to 2 M controls, 24 M samples displayed positive enrichment of the PI3K-AKT pathway and kinase-centric sets (AKT1, CDK2, SIK2, MAPKAPK5/RSK2), with higher normalized enrichment scores and significance than 12 M, indicating progressive, age-dependent activation. Conversely, CK2/CSNK2A2, NLK and ERK5/MAPK7 signatures were consistently negatively enriched at 12 M and 24 M. Together, these results indicate a shift toward enhanced AKT/cell-cycle kinase activity with a concomitant attenuation of CK2-linked regulation during physiological aging⁵⁷. Consistent with GO results, PTM-SEA indicates age-progressive activation of AKT/RSK-cell-cycle signaling linked to RNA/spliceosomal regulation, with concomitant attenuation of CK2/ERK5/NLK pathways that support actomyosin architecture and neuronal/synaptic homeostasis.”

Page 30, Line 745-749 in revised manuscript:

“For PTM-SEA analysis, phosphosites were ranked based on signed effect sizes derived from moderated t-tests. Enrichment analyses were then performed against PTMsigDB, supplemented with in-house curated glycosite datasets, to retain directional regulatory information (upregulation vs. downregulation). Normalized enrichment scores (NES) and P-values were reported for the results. The entire workflow was implemented in R, utilizing limma for statistical calculations and ssGSEA2.0 for enrichment analyses.”

3) There is a typo in the legend of Figure 5g – “Founction” should be corrected to “Function.”
Response:

We sincerely thank the reviewer for spotting the typo; which have now been corrected in the revised manuscript Fig. 5g.

Manuscript changes:

Fig. 5g “Founction” has been corrected to “Function”.

Fig. 5. Phosphoproteome, glycoproteome and proteome integrative analysis of mice aging cohort.

4) In the supplementary information:

“PTMs sites were required to meet localization probability thresholds defined by the respective search engines.” Please specify these thresholds.

Response:

Thank you for pointing this out, and we apologize for the lack of specificity. We have revised the Methods to explicitly report the engine-specific thresholds used for site localization confidence.

Phosphoproteome: We reported Class-I phosphosites only, defined as PTM Localization Probability \geq 0.75 with site q-value \leq 0.01 (and peptide/precursor q-value \leq 0.01). A sensitivity analysis at a stricter 0.90 localization threshold produced the same qualitative results.

Glycoproteome: We controlled 2D-FDR=1% at the glyco-PSM level. For confident site assignment, we required Byonic Score \geq 150, |LogProb| \geq 1 to disambiguate the N-site, and

diagnostic evidence (oxonium ions and $\geq 2Y$ -ions) consistent with the assigned composition. We have inserted the sentence in the Methods to make these thresholds explicit.

Manuscript changes:

Page 10, Line 267-270 in revised Supplementary Information:

“PTM site localization used engine-specific criteria: for the phosphoproteome we retained Class-I sites with Localization Probability > 0.75 and Site q-value < 0.01 (peptide/precursor $q < 0.01$); results were robust at a stricter 0.90 cutoff. For the glycoproteome we controlled 2D-FDR = 1% at the glyco-PSM level and required Byonic Score > 150 , $|\text{LogProb}| > 1$.”

“Raw intensities were normalized at the run level (total ion current, TIC).” Please provide a more detailed explanation of this normalization strategy.

Response:

Thank you for the comment, and we apologize for not describing the normalization clearly. We used run-level total-ion-current (TIC) normalization: after contaminant/decoy removal and ID filters, we computed each run’s TIC as the sum of precursor intensities, scaled all intensities in that run by $\alpha_r = \text{median}(\text{TIC})/\text{TIC}_r$ to equalize total signal across runs, and then applied a \log_2 transform (ϵ was added only where zeros can occur). This step precedes any missing-value handling and is applied identically to proteome, phosphoproteome, and glycoproteome layers. We have inserted the sentence in the Methods to make these thresholds explicit.

Manuscript changes:

Page 10, Line 270-272 in revised Supplementary Information:

“Raw intensities were run-level TIC-normalized by scaling each run’s total precursor signal to the cohort median (scale factor $\alpha_r = \text{median}(\text{TIC})/\text{TIC}_r$) and then \log_2 -transformed.”

Reviewer #2:

The reviewer’s comments have been fully addressed in the revised manuscript.

Response:

We thank you for their positive evaluation and confirmation that all previous comments have been adequately addressed.

Reviewer #3:

Remarks to Author:

The authors present a platform for the serial analysis of the proteome, glycoproteome, and phosphoproteome for both tissue and liquid samples. They provide a comparison against other

methods for post-translational modification analysis and show that their method significantly improves processing time, the number of transfers needed, and other parameters. The authors validate their method on human serum and mouse brain tissue samples, while also applying their method to aged animals and an arsenic-treated cell line.

This review is primarily focused on the manuscript from an aging perspective. The authors provide an analysis of age-associated changes in post-translation modifications which may have important implications, especially given how the immune system is known to shift with aging (Fulop et al. *Clin Rev Allergy Immunol.* 2021) and how post-translation modifications can modulate immunity (Mattingly et al. *Immune Netw.* 2025, Liu et al. *Immunity.* 2016).

The changes in antibody-associated glycopeptides with age are intriguing and could have implications for modern immunotherapies and anti-aging approaches. Recent literature has linked the IgG antibody to aging and tissue fibrosis (Yu et al. *Cell Metab.* 2024). There is also literature discussing the role of Fc fucosylation on antibody behavior (Golay et al. *Front. Immunol.* 2022., Saporiti et al. *Commun. Biol.* 2023.), but the age-associated impact of post-translational modifications on antibody function is less studied.

Response:

We sincerely thank you for the thorough and insightful evaluation of our manuscript. We appreciate the positive feedback on the strengths of our study and the constructive suggestions that have helped us improve its clarity and impact. All reviewer-recommended references have been added at the relevant locations and are now cited as [42], [52], [53], [54], and [55].

Main Comments:

In Figure 5G, one drawback is that for two of the tissues analyzed (MCSF and Mserum) only the proteome and glycoproteome are shown which takes away from the claim that the platform enables easy integration of all three levels.

Response:

We sincerely thank the reviewer for this insightful comment regarding the phosphoproteome data of MCSF and Mserum samples in Fig. 5g. We fully agree with the reviewer's observation that phosphoproteome data is absent for these two samples, and we greatly appreciate the chance to clarify the underlying reason.

The lack of a phosphoproteome layer for MCSF and Mserum in Fig. 5g does not stem from a limitation of our integrative platform, but rather from an intrinsic constraint of biofluid matrices. Prior studies have established that phosphoproteome coverage in biofluids-particularly plasma/serum and cerebrospinal fluid (CSF)-is intrinsically limited: even with dedicated phosphopeptide enrichment and high-resolution MS-MS, stable yields are typically only on the order of tens to a few hundred phosphopeptides. Consistent with those literatures (listed as below), our own pilot runs from 1 μ L serum input yielded only ~7-20 unique phosphopeptides. Thus, our data did not exceed the known matrix-imposed ceiling and were insufficient for robust downstream bioinformatic analysis; accordingly, we did not include these phosphoproteome layers in the main figures. We hope this clarification addresses the reviewer's concern.

We have summarized the relevant studies below:

- **Ref. 1:** *J. Proteom. Res.* (2010, 9: 1385-1391) reported 127 phosphopeptides from human

- plasma and 123 from CSF using IMAC/TiO₂ enrichment with EDTA co-injection;
- **Ref. 2:** *Clin. Proteom.* (2018, 15: 29) identified 111 phosphotyrosine (pY) peptides from CSF via anti-pY enrichment;
 - **Ref. 3:** *Proteomics* (2022, 22: 2100216) performed a head-to-head comparison on serum digests and found that, even across different enrichment media/strategies and higher injection amounts, the total number of identified phosphopeptides ranged only from ~19 to 147;
 - **Ref. 4:** *Anal. Chem.* (2024, 96: 8254-8262) recent work on serum further confirmed this constraint: conventional phosphoproteomics yielded only 172 phosphopeptides.

From the text, Figures 4F and 4G should demonstrate that the MEblue and MEyellow modules are positively and negatively correlated with aging, but Figure 4F itself does not seem to provide information relevant to the module correlations. It is not entirely clear how the structure of the dendrogram provides additional information.

Response:

We thank the reviewer for this insightful observation. The reviewer is correct in noting that the direct demonstration of module-trait correlations is contained within Fig. 4g. We apologize for any lack of clarity in the original manuscript text regarding the specific role of Fig. 4f. Fig. 4f and 4g are designed to present two sequential, yet distinct, steps in our Weighted Gene Co-expression Network Analysis (WGCNA):

Fig. 4f (Module detection and structure); Top: hierarchical clustering dendrogram of glyco(features) using TOM-based dissimilarity; colored bars indicate DynamicTreeCut modules. Bottom: clustering of module eigengenes (MEs); branch heights reflect 1-correlation among MEs. This panel documents module formation and inter-module relatedness and is not a trait correlation.

Fig. 4g (Module-trait correlations); Pearson correlations between each module eigengene and traits (Age, Sex). Cells show r with FDR-adjusted P-value in parentheses; asterisks denote significance. MEblue is positively correlated with Age, whereas MEyellow is negatively correlated. We have revised the manuscript text to more precisely describe the purposes of these figures.

Manuscript changes:

Page 14, Line 361-364 in revised manuscript:

“For glycans, the eigengene dendrogram (Fig. 4f, bottom) summarizes inter-module relatedness and guided module merging ($\text{mergeCutHeight} = 0.25$). Trait associations were then tested by correlating module eigengenes with Age and Sex (Fig. 4g).”

The flow cytometry analysis in Figure 6j provides validation of changes to cell cycle regulators, but the specific markers, fluorophores, and staining protocol, all of which can significantly impact the results and are important for reproducibility, are not mentioned in the manuscript.

Response:

Thank you for pointing this out. We fully agree that reproducibility requires detailed methodological information. In response, we have added a detailed description in the Supplementary Information (SI) titled “Flow cytometry analysis”.

Manuscript changes:

Page 12, Line 330-339 in revised Supplementary Information:

“Flow cytometry analysis: Cellular DNA was stained with propidium iodide (PI) to assess cell cycle distribution and apoptosis. Briefly, harvested cells were washed with phosphate-buffered saline (PBS) and fixed in pre-chilled 70% ethanol at 4°C for 12 hours. After removing the ethanol, the cells were resuspended in a staining buffer containing PI and RNase A (0.5 mL per tube) and then were incubated at 37°C in the dark for 30 minutes. PI fluorescence was detected in the B690-PC5.5 channel of a flow cytometer (excitation wavelength, 488 nm), while forward and side scatter characteristics were simultaneously analyzed. Cell cycle phases and apoptotic cells were determined based on DNA content distribution, where the G0/G1 peak set to 1, the G2/M peak as approximately 2, and the sub-G1 peak was identified as indicative of apoptosis. Detection was performed using a flow cytometer (Beckman Coulter CytoFLEX S).”

Minor Comments:

The authors do not provide the number of samples used to generate the statistics and/or plots in Figures 1 (C – D, E – G) and 3 (E – K).

Response:

We thank you for your kind reminder. We have now added the sample size (n) to all relevant figure legends. Specifically, for Fig. 1c-g, n=2 technical replicates per method; For Fig. 3e-k, n=3 technical replicates per method.

Manuscript changes:

Page 6, Line 146-147 in revised manuscript:

“For Fig. 1c-g, n=2 technical replicates per method.”

Page 12, Line 291-292 in revised manuscript:

“For Fig. 3e-k, n=3 technical replicates per method.”

Lines 300-301 indicate that “The limited availability of mammalian samples representing physiological aging poses additional challenges for related analyses.” How does the paper address the limited availability of mammalian samples? This line may not be necessary or there should be more clarity in the text.

Response:

We appreciate the comment. Our original wording was overly general. Here, “limited availability” specifically refers to the minute tissue mass obtained from physiologically aged mammalian brain regions-for example, mouse choroid plexus-where the per-animal material and peptide yield frequently fall below conventional proteomics input requirements.

Manuscript changes:

Page 12, Line 300-302 in revised manuscript:

“In studies of physiological aging, certain mammalian brain regions yield minute tissue mass

and limited peptide amounts, posing practical challenges for conventional proteomics input and statistical stability.”

In Lines 581-582, the authors state that “...MuPPE has revealed temporal and spatial PTMs changes...”, but no spatial information was provided in the study. The authors may be referring to tissue-specific changes, but the current language could be made clearer.

Response:

Thank you for this insightful suggestion. We have revised the wording to “temporal and tissue-specific PTM changes” (Page 25, Line 595) to accurately reflect our findings.

Manuscript changes:

Page 25, Line 594 in revised manuscript:

“...temporal and tissue-specific PTM changes...”

Response:

We thank you for your co-review efforts and for supporting the evaluation of our manuscript.

Reviewer #4:

Response:

We thank you for your co-review efforts and for supporting the evaluation of our manuscript.

Remarks to Author:

The authors present a platform for the serial analysis of the proteome, glycoproteome, and phosphoproteome for both tissue and liquid samples. They provide a comparison against other methods for post-translational modification analysis and show that their method significantly improves processing time, the number of transfers needed, and other parameters. The authors validate their method on human serum and mouse brain tissue samples, while also applying their method to aged animals and an arsenic-treated cell line.

This review is primarily focused on the manuscript from an aging perspective. The authors provide an analysis of age-associated changes in post-translation modifications which may have important implications, especially given how the immune system is known to shift with aging (Fulop et al. *Clin Rev Allergy Immunol.* 2021) and how post-translation modifications can modulate immunity (Mattingly et al. *Immune Netw.* 2025, Liu et al. *Immunity.* 2016).

The changes in antibody-associated glycopeptides with age are intriguing and could have implications for modern immunotherapies and anti-aging approaches. Recent literature has linked the IgG antibody to aging and tissue fibrosis (Yu et al. *Cell Metab.* 2024). There is also literature discussing the role of Fc fucosylation on antibody behavior (Golay et al. *Front. Immunol.* 2022., Saporiti et al. *Commun. Biol.* 2023.), but the age-associated impact of post-translational modifications on antibody function is less studied.

Main Comments:

In Figure 5G, one drawback is that for two of the tissues analyzed (MCSF and Mserum) only the proteome and glycoproteome are shown which takes away from the claim that the platform enables easy integration of all three levels.

From the text, Figures 4F and 4G should demonstrate that the MEblue and MEyellow modules are positively and negatively correlated with aging, but Figure 4F itself does not seem to provide information relevant to the module correlations. It is not entirely clear how the structure of the dendrogram provides additional information.

The flow cytometry analysis in Figure 6j provides validation of changes to cell cycle regulators, but the specific markers, fluorophores, and staining protocol, all of which can significantly impact the results and are important for reproducibility, are not mentioned in the manuscript.

Minor Comments:

The authors do not provide the number of samples used to generate the statistics and/or plots in Figures 1 (C – D, E – G) and 3 (E – K).

Lines 300-301 indicate that “The limited availability of mammalian samples representing physiological aging poses additional challenges for related analyses.” How does the paper address the limited availability of mammalian samples? This line may not be necessary or there should be more clarity in the text.

In Lines 581-582, the authors state that "...MuPPE has revealed temporal and spatial PTMs changes...", but no spatial information was provided in the study. The authors may be referring to tissue-specific changes, but the current language could be made clearer.
